

# Local optimization on pure Gaussian state manifolds

**Bennet Windt[1,2*], Alexander Jahn[3†], Jens Eisert[3,4,5‡] and Lucas Hackl[6,7,2∘]**

**1** Blackett Laboratory, Imperial College London, Prince Consort Road, SW7 2AZ, UK
**2** Max-Planck-Institut für Quantenoptik, Hans-Kopfermann-Str. 1, 85748 Garching, Germany
**3** Dahlem Center for Complex Quantum Systems,
Freie Universität Berlin,14195 Berlin, Germany
**4** Mathematics and Computer Science, Takustraße 9,
Freie Universität Berlin 14195 Berlin, Germany
**5** Helmholtz-Zentrum Berlin für Materialien und Energie,
Hahn-Meitner-Platz 1, 14109 Berlin, Germany
**6** School of Mathematics and Statistics & School of Physics,
The University of Melbourne, Parkville, VIC 3010, Australia
**7** QMATH, Department of Mathematical Sciences, University of Copenhagen,
Universitetsparken 5, 2100 Copenhagen, Denmark
**8** Munich Center for Quantum Science and Technology,
Schellingstr. 4, 80799 München, Germany

★ bennet.windt17@imperial.ac.uk, † a.jahn@fu-berlin.de,
‡ jense@zedat.fu-berlin.de, ∘ lucas.hackl@unimelb.edu.au

## Abstract

We exploit insights into the geometry of bosonic and fermionic Gaussian states to develop an efficient local optimization algorithm to extremize arbitrary functions on these families of states. The method is based on notions of gradient descent attuned to the local geometry which also allows for the implementation of local constraints. The natural group action of the symplectic and orthogonal group enables us to compute the geometric gradient efficiently. While our parametrization of states is based on covariance matrices and linear complex structures, we provide compact formulas to easily convert from and to other parametrization of Gaussian states, such as wave functions for pure Gaussian states, quasiprobability distributions and Bogoliubov transformations. We review applications ranging from approximating ground states to computing circuit complexity and the entanglement of purification that have both been employed in the context of holography. Finally, we use the presented methods to collect numerical and analytical evidence for the conjecture that Gaussian purifications are sufficient to compute the entanglement of purification of arbitrary mixed Gaussian states.

# 1 Introduction

Gaussian states form one of the most prominently used and best understood families of quantum states. The standard definition covers bosonic [1–3] and fermionic [4] Gaussian states both pure and mixed. They naturally appear as ground and thermal states of quadratic Hamiltonians in physical systems and are hence ubiquitous in non-interacting quantum many-body systems in the condensed matter context and as vacua in free field theories. Bosonic Gaussian states are heavily used in the study of bosonic systems with negligible interactions, such as Bose-Einstein condensates [5], instances of systems of cold atoms in optical lattices [6] and to a very good approximation photonic systems [7]. Their fermionic counterparts are equally important for the study of fermionic quantum many-body systems, including systems captured by the Bardeen–Cooper–Schrieffer (BCS) theory [8] or the Hartree-Fock framework [9] that can be seen as a variational principle over Gaussian fermionic states. Other applications range from field theories [10], continuous variable quantum information [1–3], relativistic quantum information [11] and quantum fields in curved spacetime [12].

Mathematically speaking, pure Gaussian states can be seen as forming Kähler sub-manifolds of the projective Hilbert space, *i.e.*, they have a natural notion of distance (Riemannian manifold with metric) and the structure of a classical phase space (symplectic manifold with symplectic form). This mathematical structure will be heavily relied on in this work, where pure bosonic and fermionic Gaussian states are the focus of attention. For systems constituted of $N$ modes, the manifold of pure bosonic states $\mathcal{M}_b$ and of pure fermionic states $\mathcal{M}_f$ can be constructed as a symmetric space, *i.e.*, as a quotient of two Lie groups, namely

$$\mathcal{M}_b = \mathrm{Sp}(2N,\mathbb{R})/\mathrm{U}(N), \tag{1}$$
$$\mathcal{M}_f = \mathrm{O}(2N)/\mathrm{U}(N), \tag{2}$$

where $\mathrm{Sp}(2N,\mathbb{R})$ is the symplectic group, $\mathrm{O}(2N)$ the orthogonal group and $\mathrm{U}(N)$ the unitary group. When restricting to one superselection sector of the parity of the fermion number, we can restrict to the special orthogonal group $\mathrm{SO}(2N)$. Gaussian manifolds come with a natural group action of the respective groups, which we can exploit when performing local optimization.

We optimize over bosonic and fermionic Gaussian manifolds by taking the natural geometry into account, *i.e.*, the notion of distance between different quantum states as measured by the Fubini-Study metric [13, 14]. Given a Riemannian manifold $\mathcal{M}$ with local coordinates $x = (x^\mu)$ and positive-definite metric $\boldsymbol{g} = (g_{\mu\nu})$, such that $v \cdot u = g_{\mu\nu} v^\mu u^\nu$, the gradient descent vector field $\widetilde{\mathcal{F}}^\mu$ of a function $f$ is

$$\mathcal{F}^\mu = -G^{\mu\nu}\frac{\partial f}{\partial x^\nu}, \tag{3}$$

where $G^{\mu\nu}$ is the inverse of $g_{\mu\nu}$ with $G^{\mu\sigma}g_{\sigma\nu} = \delta^\mu{}_\nu$. Typically, the inverse metric $G^{\mu\nu}$ needs to be re-evaluated at every point of the manifold, but for Gaussian states we can explicitly construct a basis in which the matrix representation of $G^{\mu\nu}$ is constant. This provides a crucial speedup of the underlying algorithm.

The goal of this manuscript is two-fold: First, we demonstrate how the rich geometry of pure bosonic and fermionic Gaussian states can be exploited to find extremal points of arbitrary real functions without dealing with redundant directions or parametrizations. Second, we use a unified framework to describe pure bosonic and fermionic states and carefully review how to convert between other representations of Gaussian states. This ensures that a reader can seamlessly apply our methods to their problem of choice. The present manuscript thereby complements [15], where the geometry of quantum states is discussed, and [16], where the unified mathematical formalism for Gaussian states is rigorously introduced.

*A crucial motivation for our work stems from the goal to compare entanglement of purification (EoP) and complexity of purification (CoP) in free quantum fields, which recently attracted increasing interest in the context of applying quantum information methods to holography and field theory. We provide the* GaussianOptimization.m *Mathematica package as a simple implementation of our methods, which has already been used successfully in [17] to study EoP and CoP in quantum field theory. The package can be downloaded from our arXiv submission.*

This manuscript is structured as follows: In section 2, we review a unified formalism to treat pure bosonic and fermionic Gaussian states and compute the resulting Kähler geometry (positive-definite metric, symplectic form) on the resulting state manifold. In section 3, we provide a comprehensive treatment of the most commonly used parametrizations of pure Gaussian states and how to convert between them. In section 4, we use the geometry of the pure Gaussian state manifold to develop a gradient descent algorithm with an efficient evaluation of $G^{\mu\nu}$ which avoids over-parametrization of tangent directions. The following section 5 is devoted to applications, including the well-known problem of finding approximate ground states, computing Gaussian entanglement of purification (EoP) and Gaussian complexity of purification (CoP), for which a given function $f$ is optimized over all Gaussian purifications of a given mixed Gaussian state. In section 6, we use our methods to collect numerical and analytical evidence for two conjectures stating that for mixed Gaussian states, the Gaussian EoP is actually optimal (and thus coincides with regular EoP), as well as stating which system decompositions are necessary to reach this optimum. Finally, we conclude with a discussion of our results in section 7.

## 2 Review of Gaussian states

We introduce bosonic and fermionic Gaussian states, both pure and mixed, with a particular emphasis on the geometry of the state manifold. While standard reviews of Gaussian states include ref. [1] based on covariance matrices, we follow the conventions of [15, 16, 18] based on linear complex structures that provides a basis-independent and unified treatment of bosons and fermions.

### 2.1 Quadrature operators and Majorana modes

Bosonic and fermionic quantum systems with $N$ modes can be constructed from $N$ creation or annihilation operators

$$\hat{\xi} \overset{a,a\dagger}{\equiv} (\hat{a}_1, \cdots, \hat{a}_N, \hat{a}_1^\dagger, \cdots, \hat{a}_N^\dagger), \tag{4}$$

which satisfy commutation relations $[\hat{a}_i, \hat{a}_j] = [\hat{a}_i^\dagger, \hat{a}_j^\dagger] = 0$, $[\hat{a}_i, \hat{a}_j^\dagger] = \delta_{ij}$ for bosons or anti-commutation relations $\{\hat{a}_i, \hat{a}_j\} = \{\hat{a}_i^\dagger, \hat{a}_j^\dagger\} = 0$, $\{\hat{a}_i, \hat{a}_j^\dagger\} = \delta_{ij}$ for fermions. Instead of (4), we can choose a basis of $2N$ Hermitian operators

$$\hat{\xi} \overset{q,p}{\equiv} (\hat{q}_1, \cdots, \hat{q}_N, \hat{p}_1, \cdots, \hat{p}_N), \tag{5}$$

which are related to the first by the equations

$$\hat{a}_i = \frac{\hat{q}_i + i\hat{p}_i}{\sqrt{2}} \quad \text{and} \quad \hat{a}_i^\dagger = \frac{\hat{q}_i - i\hat{p}_i}{\sqrt{2}} \tag{6}$$

with $[\hat{q}_i, \hat{q}_j] = [\hat{p}_i, \hat{p}_j] = 0$, $[\hat{q}_i, \hat{p}_j] = i\delta_{ij}$ for bosons and $\{\hat{q}_i, \hat{q}_j\} = \{\hat{p}_i, \hat{p}_j\} = \delta_{ij}$, $\{\hat{q}_i, \hat{p}_j\} = 0$ for fermions. Most readers will be familiar with this notation for bosonic systems, where the Hermitian

basis operators in (5) are called *quadratures*. However, we will use the same naming convention for Hermitian fermionic operators, which often go by the name of *Majorana modes*, just as our creation and annihilation operators from (4) referred to both bosonic or fermionic variables. The goal of these conventions is to treat bosons and fermions in a unified framework. All of our formulas containing indices $a, b, c$ will be manifestly independent from the chosen basis $\hat{\xi}$, but when giving concrete examples, we will typically provide the explicit matrix representations for the two bases from (4) and (5). The position of the index indicates if the corresponding matrix row or column refers to the classical phase space (upper index) or its dual (lower index). We use Einstein's summation convention where we implicitly assume to sum over repeated indices, where we are only allowed to pair an upper and a lower index.[1] This formalism is heavily used in the general relativity and high energy physics literature, but is particularly suitable for the unified treatment of bosonic and fermionic Gaussian states. *We will use the symbols $\overset{q,p}{\equiv}$ and $\overset{a,a\dagger}{\equiv}$ to indicate that the RHS of the equation gives the explicit matrix representations in the basis* (5) *and* (4)*, respectively.*

As we will see, Gaussian states are uniquely specified by their two-point correlation functions in the fundamental operators $\hat{\xi}$. For any state $\rho$, we denote the expectation value of an operator $\hat{O}$ as $\langle\hat{O}\rangle = \text{Tr}(\rho\hat{O})$. We may separately consider the symmetrized and antisymmetrized part of these correlations, given by the two real bilinear forms

$$G^{ab} = \langle \hat{\xi}^a \hat{\xi}^b + \hat{\xi}^b \hat{\xi}^a \rangle \tag{7}$$

$$\Omega^{ab} = -i\langle \hat{\xi}^a \hat{\xi}^b - \hat{\xi}^b \hat{\xi}^a \rangle, \tag{8}$$

$$(\text{Requirement: } z^a = \langle \hat{\xi}^a \rangle = 0), \tag{9}$$

where we restrict to $z^a = 0$ for the purpose of this manuscript to present bosons and fermions in parallel.[2]

For bosons, the *symplectic form* $\Omega$ is fixed by canonical commutation relations (CCR), while the positive-definite *metric* $G$ contains the physical correlations; for fermions, the situation is reversed, with $G$ fixed by canonical anti-commutation relations (CAR) and $\Omega$ describing the physical correlations. In summary, we have

$$\begin{aligned} [\hat{\xi}^a, \hat{\xi}^b] &= i\Omega^{ab}, \quad \textbf{(bosons)} \\ \{\hat{\xi}^a, \hat{\xi}^b\} &= G^{ab}. \quad \textbf{(fermions)} \end{aligned} \tag{10}$$

With respect to our bases, we have the state-independent expressions

$$\begin{aligned} \Omega &\overset{q,p}{\equiv} \begin{pmatrix} 0 & \mathbb{1} \\ -\mathbb{1} & 0 \end{pmatrix} \overset{a,a\dagger}{\equiv} \begin{pmatrix} 0 & -i\mathbb{1} \\ i\mathbb{1} & 0 \end{pmatrix}, \quad \textbf{(bosons)} \\ G &\overset{q,p}{\equiv} \begin{pmatrix} \mathbb{1} & 0 \\ 0 & \mathbb{1} \end{pmatrix} \overset{a,a\dagger}{\equiv} \begin{pmatrix} 0 & \mathbb{1} \\ \mathbb{1} & 0 \end{pmatrix}. \quad \textbf{(fermions)} \end{aligned} \tag{11}$$

When having chosen a set of creation and annihilation operators, our Hilbert space $\mathcal{H}$ is spanned by the orthonormal basis of number eigenvectors $|n_1, \ldots, n_N\rangle$ with $\hat{n}_i |n_1, \ldots, n_N\rangle = n_i |n_1, \ldots, n_N\rangle$, where $\hat{n}_i = \hat{a}_i^\dagger \hat{a}_i$ and $n_i \in \mathbb{N}_{\geq 0}$ for bosons and $n_i \in \{0, 1\}$ for fermions. We

---

[1]Readers familiar with Penrose' abstract index notation [19] can also read such as equations as tensor identities.

[2]For bosonic states with $z^a \neq 0$, it is easy to adjust the definition of $G^{ab}$ to be given by $G^{ab} = \langle \hat{\xi}^a \hat{\xi}^b + \hat{\xi}^b \hat{\xi}^a \rangle - 2z^a z^b$. While there exist fermionic states with $z^a \neq 0$, they will either not be Gaussian or they are unphysical (as $z^a$ would need to consist of Grassmann variables). Note that also the fermionic superselection rule forbids $z^a \neq 0$ for genuine fermionic systems, but we could have $z^a \neq 0$ for spin states mapped to fermions via Jordan-Wigner transformation. As we focus on physical Gaussian states, we do not consider either of these cases.

Table 1: *Overview of notations for operator bases.* Listed are real (self-adjoint) and complex operator bases for bosons and fermions, as well as a unified notation used throughout this work. For an $N$-mode quantum system, indices are in the range $j, k \in \{1, \ldots, N\}$ or $a, b \in \{1, \ldots, 2N\}$. The creation and annihilation operators in a complex basis satisfy *canonical commutation/anti-commutation relations* (CCR/CAR). Commonly used alternative notations are also listed, some omitting the hat notation for Hilbert space operators.

|  | **Real basis** | **Complex basis** |
| --- | --- | --- |
| **Bosons** | Phase space $(\hat{q}_j, \hat{p}_k)$<br>*Also:* $(\hat{x}_j, \hat{p}_k)$ | CCR operators $(\hat{b}_j, \hat{b}_k^\dagger)$<br>*Also:* $(\hat{a}_j, \hat{a}_k^\dagger)$ |
| **Fermions** | Majorana modes $\hat{m}_a$<br>*Also:* $\gamma_a$, $c_a$, $(c_j, \tilde{c}_k)$ | CAR operators $(\hat{f}_j, \hat{f}_k^\dagger)$<br>*Also:* $(\hat{c}_j, \hat{c}_k^\dagger)$ |
| **Unified** | $\hat{\xi} \overset{q,p}{=} (\hat{q}_j, \hat{p}_k)$ | $\hat{\xi} \overset{a,a\dagger}{=} (\hat{a}_j, \hat{a}_k^\dagger)$ |

now consider the *state-dependent* bilinear forms containing the physical correlations. These are contained in the *covariance matrix* $\Gamma^{ab}$, defined as[3]

$$\Gamma^{ab} = \begin{cases} G^{ab} & \textbf{(bosons)} \\ \Omega^{ab} & \textbf{(fermions)} \end{cases} . \tag{12}$$

We can combine the state-dependent (12) and state-independent parts (11) into a single object $J$, defined below. Due to the fact that $G^{ab}$ is always positive-definite, we can invert it to define its inverse $g_{ab} = (G^{-1})_{ab}$ with $G^{ac} g_{cb} = \delta^a{}_b$. Similarly, we define $\omega_{ab} = (\Omega^{-1})_{ab}$ satisfying $\Omega^{ac} \omega_{cb} = \delta^a{}_b$. Note that $\Omega$ may not be invertible, in which case $\omega$ refers to the pseudo-inverse with respect to $G$. This enables us to define the linear map

$$J^a{}_b = \begin{cases} -G^{ac} \omega_{cb} & \textbf{(bosons)} \\ \Omega^{ac} g_{cb} & \textbf{(fermions)} \end{cases} , \tag{13}$$

which depends on the state under consideration. *We will see in (15) that for Gaussian states the two formulas in (13) coincide, completely specifying all correlations for both bosons and fermions.*

## 2.2 Definition of pure Gaussian states

Up to this point, the quantum state $\rho$ has been assumed to be an arbitrary quantum state in the Hilbert space (with $z^a = 0$). In what follows, we put a specific emphasis on the set of *pure Gaussian states*. There are many equivalent definitions in the literature: One may define Gaussian states as those satisfying Wick's theorem, as ground states of non-interacting (*i.e.*, quadratic), non-degenerate Hamiltonians, or as states vanishing under a full set of specific annihilation operators. Here, we use yet another equivalent, though very compact definition [16] based on (13), which states that for a state $\rho$

$$\rho \text{ is a pure Gaussian state} \quad \Longleftrightarrow \quad J^2 = -\mathbb{1} . \tag{14}$$

---

[3]Depending on convention, $\Gamma^{ab}$ is sometimes defined with additional prefactors, *e.g.*, as $\Gamma^{ab} = -\Omega^{ab}$ for fermions.

If this holds, both formulas in (13) coincide[4], such that

$$\rho \text{ is a pure Gaussian} \quad \Leftrightarrow \quad -G^{ac}\omega_{cb} = \Omega^{ac}g_{cb}. \tag{15}$$

As a pure state, we can write $\rho = |\psi\rangle\langle\psi|$ for a normalized state vector $|\psi\rangle$. One can show that this state vector $|\psi\rangle$ is uniquely determined (up to a complex phase) by either the covariance matrix $\Gamma^{ab}$ or equivalently by the complex structure $J$, which we use as a label to write $|\psi\rangle = |J\rangle$.[5]

An alternative and completely equivalent definition of pure Gaussian states can be phrased directly in terms of $J$, where $|J\rangle$ is the solution of the equations

$$\tfrac{1}{2}(\delta^a{}_b + iJ^a{}_b)\hat{\xi}^b |J\rangle = 0. \tag{16}$$

This definition is based on the observation that the eigenvectors $\hat{\xi}^a_\pm$ of $J$ with eigenvalues $\pm i$ are given by[6]

$$\hat{\xi}^a_\pm = \tfrac{1}{2}(\delta^a{}_b \mp iJ^a{}_b)\hat{\xi}^b \quad \text{with} \quad \hat{\xi}^a_- |J\rangle = 0. \tag{17}$$

The variables $\hat{\xi}^a_\pm$ behave in many ways as creation and annihilation operators, but do not require a specific basis in phase space, which enables a compact covariant proof of Wick's theorem [16]. Moreover, $\hat{\xi}^a_\pm$ spans the $N$-dimensional complex eigenspaces $V_{\mathbb{C}}^\pm$ of $J$, which are the spaces of creation or annihilation operators associated to $|J\rangle$, respectively. We refer to $\hat{\xi}^a_\pm$ as phase space covariant creation and annihilation operators, which satisfy the following commutation (bosons) or anti-commutation (fermions) relations:

$$\begin{aligned}
[\hat{\xi}^a_\pm, \hat{\xi}^b_\pm] = 0, \quad [\hat{\xi}^a_-, \hat{\xi}^b_+] = C_2^{ab}, \quad &\textbf{(bosons)}\\
\{\hat{\xi}^a_\pm, \hat{\xi}^b_\pm\} = 0, \quad \{\hat{\xi}^a_-, \hat{\xi}^b_+\} = C_2^{ab}, \quad &\textbf{(fermions)}
\end{aligned} \tag{18}$$

where we introduced the 2-point function

$$C_2^{ab} = \langle \hat{\xi}^a \hat{\xi}^b \rangle = \frac{1}{2}(G^{ab} + i\Omega^{ab}). \tag{19}$$

For a given state vector $|J\rangle$, we can always choose a basis

$$\hat{\xi} \overset{q,p}{\equiv} (\hat{q}_1, \ldots, \hat{q}_N, \hat{p}_1, \ldots, \hat{p}_N) \overset{a,a^\dagger}{\equiv} \left(\hat{a}_1, \ldots, \hat{a}_N, \hat{a}_1^\dagger, \ldots, \hat{a}_N^\dagger\right), \tag{20}$$

in which $\Omega$ and $G$ simultaneously take the standard forms

$$\Omega \overset{q,p}{\equiv} \begin{pmatrix} 0 & \mathbb{1} \\ -\mathbb{1} & 0 \end{pmatrix} \overset{a,a^\dagger}{\equiv} \begin{pmatrix} 0 & -i\mathbb{1} \\ i\mathbb{1} & 0 \end{pmatrix}, \quad G \overset{q,p}{\equiv} \begin{pmatrix} \mathbb{1} & 0 \\ 0 & \mathbb{1} \end{pmatrix} \overset{a,a^\dagger}{\equiv} \begin{pmatrix} 0 & \mathbb{1} \\ \mathbb{1} & 0 \end{pmatrix}. \tag{21}$$

In contrast to (10), where only one of the respective background structure ($\Omega$ for bosons *or* $G$ for fermions) takes this form, while the other may take any allowed form, we have now chosen the basis $\{\hat{\xi}^a\}$ attuned to $|J\rangle$, so that also $\Gamma$ ($G$ for bosons, $\Omega$ for fermions) takes the above standard form. In this basis, we find[7]

---

[4]We can prove this by computing $(G\omega)^{-1} = \omega^{-1}G^{-1} = \Omega g$ and vice versa. Moreover, (14) implies $J^{-1} = -J$. The two relations together imply (15).

[5]If we allow for bosons $z^a = \langle\xi\rangle^a \neq 0$, we would need to include this in our label of the state vector to write $|\psi\rangle = |J, z\rangle$.

[6]Here, $\hat{\xi}^a_\pm$ is a vector whose components are operators. It is easy to verify $J^a{}_b\hat{\xi}^b_\pm = \pm i\hat{\xi}^a_\pm$ from (17).

[7]Complex conjugation of the basis $\hat{\xi}^a$ satisfies $\hat{\xi}^{\dagger a} = C^a{}_b\hat{\xi}^b$ implying $\hat{\xi}^{\dagger a}_\pm = C^a{}_b\hat{\xi}^b_\mp$. We have the conjugation matrix

$$C \overset{q,p}{\equiv} \begin{pmatrix} \mathbb{1} & 0 \\ 0 & \mathbb{1} \end{pmatrix} \overset{a,a^\dagger}{\equiv} \begin{pmatrix} 0 & \mathbb{1} \\ \mathbb{1} & 0 \end{pmatrix}.$$

$$\hat{\xi}_- \overset{q,p}{\equiv} \left( \tfrac{\hat{a}_1}{\sqrt{2}}, \ldots, \tfrac{\hat{a}_N}{\sqrt{2}}, \tfrac{-\mathrm{i}\hat{a}_1}{\sqrt{2}}, \ldots, \tfrac{-\mathrm{i}\hat{a}_N}{\sqrt{2}} \right) \overset{a,a^\dagger}{\equiv} (\hat{a}_1, \ldots, \hat{a}_N, 0, \ldots, 0) \,,$$

$$\hat{\xi}_+ \overset{q,p}{\equiv} \left( \tfrac{\hat{a}_1^\dagger}{\sqrt{2}}, \ldots, \tfrac{\hat{a}_N^\dagger}{\sqrt{2}}, \tfrac{\mathrm{i}\hat{a}_1^\dagger}{\sqrt{2}}, \ldots, \tfrac{\mathrm{i}\hat{a}_N^\dagger}{\sqrt{2}} \right) \overset{a,a^\dagger}{\equiv} \left( 0, \ldots, 0, \hat{a}_1^\dagger, \ldots, \hat{a}_N^\dagger \right) \,. \tag{22}$$

Most of the relevant intuition for Gaussian states for $N$ modes comes from considering one bosonic or two fermionic modes, as reviewed in the following examples, as Gaussian states for a single fermionic mode is *almost* trivial. We will further see explicitly that the families of fermionic Gaussian states consist of two disconnected components.

**Example 1** (Single mode pure Gaussian bosonic states). *We consider a single bosonic mode with $\hat{\xi} \overset{q,p}{\equiv} (\hat{q}, \hat{p}) \overset{a,a^\dagger}{\equiv} (\hat{a}, \hat{a}^\dagger)$. With respect to the number eigenvectors $|n\rangle$, the most general Gaussian state vector with $z^a = 0$ is*

$$|J\rangle = \frac{1}{\sqrt{\cosh \tfrac{\rho}{2}}} \sum_{n=0}^{\infty} \frac{\sqrt{(2n)!}}{2^n n!} \left( -e^{\mathrm{i}\phi} \tanh \tfrac{\rho}{2} \right)^n |2n\rangle \,, \tag{23}$$

*where $\phi \in [0, 2\pi]$ and $\rho \in [0, \infty)$. With respect to above bases, one finds*

$$G \overset{q,p}{\equiv} \begin{pmatrix} \cosh \rho + \cos \phi \sinh \rho & \sin \phi \sinh \rho \\ \sin \phi \sinh \rho & \cosh \rho - \cos \phi \sinh \rho \end{pmatrix} \overset{a,a^\dagger}{\equiv} \begin{pmatrix} e^{\mathrm{i}\phi} \sinh \rho & \cosh \rho \\ \cosh \rho & -e^{-\mathrm{i}\phi} \sinh \rho \end{pmatrix}, \tag{24}$$

$$J \overset{q,p}{\equiv} \begin{pmatrix} -\sin \phi \sinh \rho & \cos \phi \sinh \rho + \cosh \rho \\ \cos \phi \sinh \rho - \cosh \rho & \sin \phi \sinh \rho \end{pmatrix} \overset{a,a^\dagger}{\equiv} \begin{pmatrix} -\mathrm{i} \cosh \rho & \mathrm{i} e^{\mathrm{i}\phi} \sinh \rho \\ -\mathrm{i} e^{-\mathrm{i}\phi} \sinh \rho & \mathrm{i} \cosh \rho \end{pmatrix}. \tag{25}$$

*In summary, Gaussian states of a single bosonic mode form a two-dimensional plane parametrized by polar coordinates $(\rho, \phi)$.*

**Example 2** (Single and two mode pure Gaussian fermionic states). *We consider a single fermionic mode with $\hat{\xi} \overset{q,p}{\equiv} (\hat{q}, \hat{p}) \overset{a,a^\dagger}{\equiv} (\hat{a}, \hat{a}^\dagger)$. There are only two distinct pure Gaussian states, which are characterized by the state vectors*

$$\left\{ \begin{array}{l} |J_+\rangle = |0\rangle \\ |J_-\rangle = |1\rangle \end{array} \right\} , \tag{26}$$

*whose covariance matrix and complex structures are*

$$\Omega_\pm \overset{q,p}{\equiv} \begin{pmatrix} 0 & \pm 1 \\ \mp 1 & 0 \end{pmatrix} \overset{a,a^\dagger}{\equiv} \begin{pmatrix} 0 & \mp \mathrm{i} \\ \pm \mathrm{i} & 0 \end{pmatrix}, \tag{27}$$

$$J_\pm \overset{q,p}{\equiv} \begin{pmatrix} 0 & \pm 1 \\ \mp 1 & 0 \end{pmatrix} \overset{a,a^\dagger}{\equiv} \begin{pmatrix} \mp \mathrm{i} & 0 \\ 0 & \pm \mathrm{i} \end{pmatrix}. \tag{28}$$

*In summary, there are only two distinct Gaussian pure states for a single fermionic mode rather than a family of states. We therefore consider also two fermionic modes with $\hat{\xi} \overset{q,p}{\equiv} (\hat{q}_1, \hat{q}_2, \hat{p}_1, \hat{p}_2) \overset{a,a^\dagger}{\equiv} (\hat{a}_1, \hat{a}_2, \hat{a}_1^\dagger, \hat{a}_2^\dagger)$, where the most general Gaussian state vectors are*

$$\left\{ \begin{array}{l} |J_+\rangle = \cos \tfrac{\theta}{2} |0,0\rangle + e^{\mathrm{i}\phi} \sin \tfrac{\theta}{2} |1,1\rangle \\ |J_-\rangle = \cos \tfrac{\theta}{2} |1,0\rangle + e^{\mathrm{i}\phi} \sin \tfrac{\theta}{2} |0,1\rangle \end{array} \right\} , \tag{29}$$

*with $\theta \in [0, \pi]$ and $\phi \in [0, 2\pi]$. Their covariance matrix and complex structure are*

$$
\Omega_\pm \overset{q,p}{\equiv} \begin{pmatrix} 0 & \mp \sin\theta \sin\phi & \pm \cos\theta & \pm \sin\theta \cos\phi \\ \pm \sin\theta \sin\phi & 0 & -\sin\theta \cos\phi & \cos\theta \\ \mp \cos\theta & \sin\theta \cos\phi & 0 & \sin\theta \sin\phi \\ \mp \sin\theta \cos\phi & -\cos\theta & -\sin\theta \sin\phi & 0 \end{pmatrix}
$$

$$
\overset{a,a^\dagger}{\equiv} \begin{pmatrix} 0 & i e^{i\phi} \sin\theta & -i\cos\theta & 0 \\ -i e^{i\phi} \sin\theta & 0 & 0 & -i\cos\theta \\ i\cos\theta & 0 & 0 & -i e^{-i\phi} \sin\theta \\ 0 & i\cos\theta & i e^{-i\phi} \sin\theta & 0 \end{pmatrix} , \tag{30}
$$

$$
J_\pm \overset{q,p}{\equiv} \begin{pmatrix} 0 & \mp \sin\theta \sin\phi & \pm \cos\theta & \pm \sin\theta \cos\phi \\ \pm \sin\theta \sin\phi & 0 & -\sin\theta \cos\phi & \cos\theta \\ \mp \cos\theta & \sin\theta \cos\phi & 0 & \sin\theta \sin\phi \\ \mp \sin\theta \cos\phi & -\cos\theta & -\sin\theta \sin\phi & 0 \end{pmatrix}
$$

$$
\overset{a,a^\dagger}{\equiv} \begin{pmatrix} \mp i\cos\theta & i\delta_\mp e^{-i\phi}\sin\theta & 0 & i\delta_\pm e^{i\phi}\sin\theta \\ i\delta_\mp e^{i\phi}\sin\theta & -i\cos\theta & -i\delta_\pm e^{i\phi}\sin\theta & 0 \\ 0 & -i\delta_\pm e^{-i\phi}\sin\theta & \pm i\cos\theta & -i\delta_\mp e^{-i\phi}\sin\theta \\ i\delta_\pm e^{-i\phi}\sin\theta & 0 & -i\delta_\mp e^{i\phi}\sin\theta & i\cos\theta \end{pmatrix} , \tag{31}
$$

*with $\delta_\pm = \frac{1\pm 1}{2}$, i.e., $\delta_+ = 1$ and $\delta_- = 0$. In summary, Gaussian states of two fermionic modes form two disconnected spheres parametrized by angles $(\theta, \phi)$, where we further distinguish the Gaussian state vectors of type $|J_+\rangle$ and $|J_-\rangle$. The two sets are distinguished by the parity operator $\hat{P} = \exp(i\pi\hat{N})$, as the total number operator $\hat{N} = \sum_i \hat{a}_i^\dagger \hat{a}_i$ is even for $|J_+\rangle$ and odd for $|J_-\rangle$*

### 2.3 Gaussian transformations

In this section, we will introduce a special set of unitary transformations that map Gaussian states into Gaussian states. They are generated by operators that are quadratic in $\hat{\xi}^a$. We define the Lie group $\mathcal{G}$ as linear transformations on the classical phase space $V$ that preserve the symplectic form $\Omega^{ab}$ for bosons or the metric $G^{ab}$ for fermions

$$
\mathcal{G} = \begin{cases} \mathrm{Sp}(2N, \mathbb{R}) & \textbf{(bosons)} \\ \mathrm{O}(2N, \mathbb{R}) & \textbf{(fermions)} \end{cases} , \tag{32}
$$

which we represent as matrices $M : V \to V$ with

$$
\begin{aligned}
\mathrm{Sp}(2N, \mathbb{R}) &= \left\{ M^a{}_b \in \mathrm{GL}(2N, \mathbb{R}) \,\middle|\, M\Omega M^\intercal = \Omega \right\} , \\
\mathrm{O}(2N, \mathbb{R}) &= \left\{ M^a{}_b \in \mathrm{GL}(2N, \mathbb{R}) \,\middle|\, MGM^\intercal = G \right\} .
\end{aligned} \tag{33}
$$

The associated Lie algebras $\mathfrak{g}$ are then defined as[8]

$$
\begin{aligned}
\mathfrak{sp}(2N, \mathbb{R}) &= \left\{ K^a{}_b \in \mathfrak{gl}(2N, \mathbb{R}) \,\middle|\, K\Omega + \Omega K^\intercal = 0 \right\} , \\
\mathfrak{so}(2N, \mathbb{R}) &= \left\{ K^a{}_b \in \mathfrak{gl}(2N, \mathbb{R}) \,\middle|\, KG + GK^\intercal = 0 \right\} .
\end{aligned} \tag{34}
$$

We can construct a (projective) representation of these Lie groups as unitary operators $\mathcal{S}(M)$ on Hilbert space by exponentiating quadratic operators. For this, we first define an identification between Lie algebra elements $K \in \mathfrak{g}$ and anti-Hermitian quadratic operators $\widehat{K}$ with

$$
K^a{}_b \quad \Longleftrightarrow \quad \widehat{K} = \begin{cases} -\frac{i}{2}\omega_{ac} K^c{}_b \hat{\xi}^a \hat{\xi}^b & \textbf{(bosons)} \\ \frac{1}{2} g_{ac} K^c{}_b \hat{\xi}^a \hat{\xi}^b & \textbf{(fermions)} \end{cases} , \tag{35}
$$

---

[8]Note that the Lie algebra of $\mathrm{O}(2N, \mathbb{R})$ and $\mathrm{SO}(2N, \mathbb{R})$ are the same, commonly referred to as $\mathfrak{so}(2N, \mathbb{R})$.

which is uniquely fixed by the requirement

$$\widehat{[K_1, K_2]} = [\widehat{K}_1, \widehat{K}_2]. \tag{36}$$

For any $M = e^K$, we define the squeezing operator

$$\mathcal{S}(e^K) \cong e^{\widehat{K}}, \tag{37}$$

where $\cong$ implies equality up to a complex phase. For fermions, products of $M = e^K$ for $K \in \mathfrak{so}(2N, \mathbb{R})$ will only generate the subgroup $\mathrm{SO}(2N, \mathbb{R})$, whose group elements satisfy $\det M = 1$. To generate other group elements $M \in \mathrm{O}(2N, \mathbb{R})$ with $\det M = -1$, we can take any dual vector $v_a \in V^*$ satisfying $v_a G^{ab} v_b = 2$ to define

$$\mathcal{S}(M_v) = v_a \hat{\xi}^a, \quad \textbf{(fermions)} \tag{38}$$

representing

$$(M_v)^a{}_b = v_c G^{ca} v_b - \delta^a{}_b \in \mathrm{O}(2N, \mathbb{R}) \tag{39}$$

with $\det M_v = -1$. We can further check that $\mathcal{S}(M_v)$ is unitary. Moreover, we have $\mathcal{S}^\dagger(M_v)\hat{\xi}^a \mathcal{S}(M_v) = (M_v)^a{}_b \hat{\xi}^b$. Consequently, together $\mathcal{S}(e^K)$ and $\mathcal{S}(M_v)$ for a single chosen $v_a$ generate the full orthogonal group $\mathrm{O}(2N, \mathbb{R})$, *i.e.*, every element $M \in \mathrm{O}(2N, \mathbb{R})$ with $\det M = -1$ can be represented as a $\mathcal{S}(M) \cong \mathcal{S}(e^K)\mathcal{S}(M_v)$ for a fixed $v_a$ and $K = \log M M_v^{-1}$. This definition of $\mathcal{S}(M)$ forms a projective representation satisfying[9]

$$\mathcal{S}(M_1)\mathcal{S}(M_2) \cong \mathcal{S}(M_1 M_2). \tag{40}$$

Furthermore, we can read off the group element $M$ from $\mathcal{S}(M)$ by its action on $\hat{\xi}^a$ via the relation

$$\mathcal{S}^\dagger(M)\hat{\xi}^a \mathcal{S}(M) = M^a{}_b \hat{\xi}^b. \tag{41}$$

Every Gaussian state vector $|J\rangle$ has a stabilizer subgroup

$$\mathrm{U}(N) = \left\{ M \in \mathcal{G} \,\middle|\, M\Gamma M^\intercal = \Gamma \right\} = \left\{ M \in \mathcal{G} \,\middle|\, MJM^{-1} = J \right\}, \tag{42}$$

which preserves $\Gamma$ and $J$. Note that $\mathrm{U}(N)$ depends on $J$, so one could write $\mathrm{U}_J(N)$ to indicate this dependence. Similarly, the associated unitary transformation $\mathcal{S}(M)$ will preserve the quantum state vector $|J\rangle$ up to a complex phase, *i.e.*, we have $\mathcal{S}(M)|J\rangle \cong |J\rangle$ for all $M \in \mathrm{U}(N)$. This defines the Lie subalgebra

$$\mathfrak{u}(N) = \left\{ K \in \mathfrak{g} \,\middle|\, K\Gamma + \Gamma K^\intercal = 0 \right\} = \left\{ K \in \mathfrak{g} \,\middle|\, [K, J] = 0 \right\}. \tag{43}$$

Similarly, we have $\widehat{K}|J\rangle \propto |J\rangle$. Given a Gaussian reference state vector $|J_0\rangle$, we can reach any other Gaussian target state vector $|J\rangle$ via

$$|J\rangle \cong \mathcal{S}(M)|J_0\rangle \cong |M\Gamma_0 M^\intercal\rangle. \tag{44}$$

The solution of the equation $M\Gamma_0 M^\intercal = \Gamma$ is not unique, as we can always multiply by $u \in \mathrm{U}(N)$ associated to $|J_0\rangle$, such that $(Mu)\Gamma_0(Mu)^\intercal = Mu\Gamma_0 u^\intercal M^\intercal = M\Gamma_0 M^\intercal$. We can fix a special solution $T$ by imposing the condition $T\Gamma_0 = \Gamma_0 T^\intercal$ leading to the simpler equation

---

[9]The equality turns out to hold up to an overall sign, *i.e.*, we can choose $\mathcal{S}(M)$, such that $\mathcal{S}(M_1)\mathcal{S}(M_2) = \pm\mathcal{S}(M_1 M_2)$.

$J = TJ_0 T^{-1} = T^2 J_0$, which is solved by $T^2 = -JJ_0$. We define this as the relative complex structure[10]

$$\Delta^a{}_b = T^a{}_c T^c{}_b = -J^a{}_c (J_0)^c{}_b = \Gamma^{ac}(\Gamma_0^{-1})_{cb}. \tag{45}$$

It captures the full basis independent information about the relationship of the two Gaussian states $J$ and $J_0$. We have the following properties as proven in [16]:

- **Bosons.** The spectrum of $\Delta$ consists of pairs $(e^{2r_i}, e^{-2r_i})$ with $r_i \in [0, \infty)$, such that $T = \sqrt{\Delta}$ has eigenvalues $(e^{r_i}, e^{-r_i})$. $\Delta$ is diagonalizable and a symplectic group element.

- **Fermions.** The spectrum of $\Delta$ consists of quadruples $(e^{i2r_i}, e^{i2r_i}, e^{-i2r_i}, e^{-i2r_i})$ with $r_i \in (0, \frac{\pi}{2})$ or pairs $(1, 1)$ or $(-1, -1)$, which correspond to $r_i \in \{0, \frac{\pi}{2}\}$. If the number of pairs $(-1, -1)$ is even, *i.e.*, the eigenvalue $-1$ appears with multiplicity divisible by four, $J$ and $J_0$ lie in the same topological component of fermionic Gaussian states, *i.e.*, they can be continuously deformed into each other. Otherwise, *i.e.*, if the number of eigenvalue pairs $(-1, -1)$ is odd, $J$ and $J_0$ live in separate components. $T = \sqrt{\Delta}$ is only well defined in the former case and has quadruple eigenvalues $(e^{ir_i}, e^{ir_i}, e^{-ir_i}, e^{-ir_i})$ for $r \in (0, \frac{\pi}{2})$. If there are eigenvalue quadruples $(-1, -1, -1, -1)$, there are different, but equivalent ways[11] to define $T$ as a real linear map with $T^2 = \Delta$ in this sub block corresponding to choosing different eigenvectors for the quadruple of eigenvalues $(i, i, -i, -i)$.

We can bring $\Delta$, $T$ and $K = \log T$ into block-diagonal form. We find $2 \times 2$ one-mode squeezing blocks for bosons and $4 \times 4$ two-mode squeezing blocks for fermions. The parameters $\{r_i\}$ from above correspond to $\frac{\rho}{2}$ in our bosonic example 1 and $\frac{\theta}{2}$ in our fermionic example 2.

**Example 3** (Bosons revisited). *We reconsider Example 1 and choose the reference state vector $|J_0\rangle$ with*

$$G_0 \overset{q,p}{\equiv} \begin{pmatrix} 1 & 0 \\ 0 & 1 \end{pmatrix} \overset{a,a^\dagger}{\equiv} \begin{pmatrix} 0 & 1 \\ 1 & 0 \end{pmatrix}, \; J_0 \overset{q,p}{\equiv} \begin{pmatrix} 0 & 1 \\ -1 & 0 \end{pmatrix} \overset{a,a^\dagger}{\equiv} \begin{pmatrix} i & 0 \\ 0 & -i \end{pmatrix}. \tag{46}$$

*A general symplectic transformation $\mathcal{G} = \mathrm{Sp}(2, \mathbb{R})$ is*

$$M \overset{q,p}{\equiv} \begin{pmatrix} \cos\tau \cosh\frac{\rho}{2} - \sin\theta \sinh\frac{\rho}{2} & -\sin\tau \cosh\frac{\rho}{2} + \cos\theta \sinh\frac{\rho}{2} \\ \sin\tau \cosh\frac{\rho}{2} + \cos\theta \sinh\frac{\rho}{2} & \cos\tau \cosh\frac{\rho}{2} + \sin\theta \sinh\frac{\rho}{2} \end{pmatrix} \overset{a,a^\dagger}{\equiv} \begin{pmatrix} e^{i\tau} \cosh\frac{\rho}{2} & i e^{i\theta} \sinh\frac{\rho}{2} \\ -i e^{-i\theta} \sinh\frac{\rho}{2} & e^{-i\tau} \cosh\frac{\rho}{2} \end{pmatrix}, \tag{47}$$

*for which we have $|J\rangle \cong \mathcal{S}(M) |J_0\rangle$ with $\Gamma$ from (24), where $\phi = \tau - \theta$. The stabilizer group of $|J_0\rangle$ consists of*

$$u \overset{q,p}{\equiv} \begin{pmatrix} \cos\varphi & \sin\varphi \\ -\sin\varphi & \cos\varphi \end{pmatrix} \overset{a,a^\dagger}{\equiv} \begin{pmatrix} e^{i\varphi} & 0 \\ 0 & e^{-i\varphi} \end{pmatrix}. \tag{48}$$

*From the relative complex structure $\Delta = T^2 = -JJ_0$, we compute the generator*

$$K = \log T \overset{q,p}{\equiv} \frac{\rho}{2} \begin{pmatrix} \sin\phi & \cos\phi \\ \cos\phi & -\sin\phi \end{pmatrix} \overset{a,a^\dagger}{\equiv} \frac{\rho}{2} \begin{pmatrix} 0 & i e^{-i\phi} \\ -i e^{i\phi} & 0 \end{pmatrix}, \tag{49}$$

*such that $|J\rangle \cong e^{\widehat{K}} |J_0\rangle$. We can always change basis to reach a standard form $\phi = \frac{\pi}{2}$, where we can read off the eigenvalues $(e^\rho, e^{-\rho})$ of $\Delta$.*

---

[10]Sometimes also referred to as relative covariance matrix [20, 21].

[11]In essence, $T$ describes half way on the shortest path between $\Gamma_0$ and $\Gamma$. The eigenvalues $(-1, -1, -1, -1)$ imply that $\Gamma_0$ and $\Gamma$ are on opposite poles of spheres, in which case all the points on the equator are equivalent choices of being half-way.

**Example 4** (Fermions revisited). *We reconsider Example 2. For a single fermionic mode, we choose the reference state vector $|J_0\rangle$ with*

$$\Omega_0 \overset{q,p}{\equiv} \begin{pmatrix} 0 & 1 \\ -1 & 0 \end{pmatrix} \overset{a,a^\dagger}{\equiv} \begin{pmatrix} 0 & -\mathrm{i} \\ \mathrm{i} & 0 \end{pmatrix}, \, J_0 \overset{q,p}{\equiv} \begin{pmatrix} 0 & 1 \\ -1 & 0 \end{pmatrix} \overset{a,a^\dagger}{\equiv} \begin{pmatrix} -\mathrm{i} & 0 \\ 0 & \mathrm{i} \end{pmatrix}. \tag{50}$$

*The stabilizer subgroup* $\mathrm{U}(1)$ *consists of the same elements as in* (42), *which coincides with the group* $\mathrm{SO}(2,\mathbb{R})$. *Consequently, the only group elements that transform $|J_0\rangle = |J_+\rangle$ into $|J_-\rangle$ lie in the disconnected component. We also reconsider two fermionic modes with reference state vector $|J_0\rangle$ given by*

$$\Omega_0 \overset{q,p}{\equiv} \begin{pmatrix} 0 & \mathbb{1} \\ -\mathbb{1} & 0 \end{pmatrix} \overset{a,a^\dagger}{\equiv} \begin{pmatrix} 0 & -\mathrm{i}\,\mathbb{1} \\ \mathrm{i}\,\mathbb{1} & 0 \end{pmatrix}, \, J_0 \overset{q,p}{\equiv} \begin{pmatrix} 0 & \mathbb{1} \\ -\mathbb{1} & 0 \end{pmatrix} \overset{a,a^\dagger}{\equiv} \begin{pmatrix} -\mathrm{i}\,\mathbb{1} & 0 \\ 0 & \mathrm{i}\,\mathbb{1} \end{pmatrix}. \tag{51}$$

*There is a 4-dimensional subspace of these generators also satisfying $[K, J_0] = 0$, which generates* $\mathrm{U}(2) \subset \mathrm{O}(4,\mathbb{R})$. *We can reach the most general complex structure $J_+$ by a continuous path generated by*

$$K = \frac{1}{2} \log \Delta \overset{q,p}{\equiv} \frac{\theta}{2} \begin{pmatrix} 0 & \cos\phi & 0 & \sin\phi \\ -\cos\phi & 0 & -\sin\phi & 0 \\ 0 & \sin\phi & 0 & -\cos\phi \\ -\sin\phi & 0 & \cos\phi & 0 \end{pmatrix} \tag{52}$$

*for $\Delta = -J_+ J_0$. To reach state vectors of the form $|J_-\rangle$, we must also apply an additional transformation $\mathcal{S}(M_v)$ with $v \overset{q,p}{\equiv} (\sqrt{2}, 0, 0, 0) \overset{a,a^\dagger}{\equiv} (1, 0, 1, 0)$ to find $|J_-\rangle = \mathcal{S}(M_v)|J_+\rangle$. We can always change basis to reach a standard forms $\phi = 0$, where we can read off the eigenvalues $(e^{\mathrm{i}\theta}, e^{\mathrm{i}\theta}, e^{-\mathrm{i}\theta}, e^{-\mathrm{i}\theta})$ of $\Delta$.*

## 2.4 Geometry of pure Gaussian states

The family of pure Gaussian states forms a differentiable manifold $\mathcal{M}$. It provides a versatile tool for analytical and numerical studies of bosonic and fermionic quantum systems with applications ranging from condensed matter [5,7–9] and quantum information [1,3] to quantum optics [7] and field theory [12]. Mathematically, $\mathcal{M}$ is a symmetric space [22] (type CI for bosons and DIII for fermions) and has the properties of a so-called Kähler manifold. The latter makes Gaussian states particularly suitable for variational studies, where ground states and time evolution are approximated on a suitable subset of Hilbert space. In the following, we will discuss the rich geometry of this manifold, which plays an important role when one wishes to locally optimize a function on it. We closely follow the conventions of [15], which contains a comprehensive review of the geometry of variational families, which in turn builds upon ideas of the time-dependent variational principle [23,24].

We recall our definition of Gaussian state vectors $|J\rangle$ as normalized vectors in Hilbert space, such that their linear complex structure $J^a{}_b$ satisfies $J^2 = -\mathbb{1}$. Note that knowing $\Gamma$ does not fix the complex phase of the Hilbert space vector, *i.e.*, $\Gamma$ actually describes elements of a projective Hilbert space $\mathcal{P}(\mathcal{M})$ which we could represent as (pure) density operators $\rho_\Gamma = |J\rangle\langle J|$ rather than Hilbert space vectors $|J\rangle$. However, we often prefer to think of pure quantum states as state vectors $|\psi\rangle$ rather than density operators $\rho = |\psi\rangle\langle\psi|$ and accept that we need to keep in mind that these vectors are actually only defined up to a complex phase, *i.e.*, $|J\rangle \cong e^{\mathrm{i}\varphi}|J\rangle$.

Given a covariance matrix $\Gamma$ of a pure Gaussian state vector $|J\rangle$, we are only allowed to change it in such a way that respects symmetry (symmetric for bosons, antisymmetric for fermions) and preserves purity ($J^2 = -\mathbb{1}$). For the infinitesimal change $\delta\Gamma^{ab}$, we thus find the constraints

$$\delta\Gamma^{ab} = \delta\Gamma^{ba}, \quad \delta\Gamma J^\intercal = J\delta\Gamma, \quad \textbf{(bosons)}$$
$$\delta\Gamma^{ab} = -\delta\Gamma^{ba}, \quad \delta\Gamma J^\intercal = J\delta\Gamma. \quad \textbf{(fermions)} \tag{53}$$

Knowing the change of the covariance matrix $\Gamma$ does not uniquely fix the change of the state vector $|J\rangle$, as we could also change the complex phase. Such change would be proportional to $\mathrm{i}\,|J\rangle$. To remove such pure change of gauge, we require that the tangent vector $|\delta\Gamma\rangle = \delta\Gamma^{ab}\,|V_{ab}\rangle$ is orthogonal to $|J\rangle$ itself, *i.e.*, $\langle\Gamma|V_{ab}\rangle = 0$. Under this condition, one can derive [15]

$$|V_{ab}\rangle = \begin{cases} \frac{\mathrm{i}}{4}g_{ac}\omega_{bd}\hat{\xi}_+^c\hat{\xi}_+^d\,|J\rangle & \textbf{(bosons)} \\ \frac{1}{4}g_{ac}\omega_{bd}\hat{\xi}_+^c\hat{\xi}_+^d\,|J\rangle & \textbf{(fermions)} \end{cases}. \tag{54}$$

This allows us to compute the inner product between two different variations $\delta\Gamma$ and $\delta\tilde{\Gamma}$ as [15]

$$\langle\delta\Gamma|\delta\tilde{\Gamma}\rangle = \frac{1}{2}\Big(\boldsymbol{g}(\delta\Gamma,\delta\tilde{\Gamma}) + \mathrm{i}\,\boldsymbol{\omega}(\delta\Gamma,\delta\tilde{\Gamma})\Big), \tag{55}$$

where we introduced the real bilinear forms $\boldsymbol{g}$ and $\boldsymbol{\omega}$ on the tangent space, *i.e.*, the space of allowed variations $\delta\Gamma^{ab}$ subject to (53). Interestingly, $\boldsymbol{g}$ is a metric (symmetric, positive-definite) just as $g$ and $\boldsymbol{\omega}$ is a symplectic form (antisymmetric, non-degenerate) just as $\omega$. We can evaluate them using (55) and (54) leading to

$$\begin{aligned} \boldsymbol{g}(\delta\Gamma,\delta\tilde{\Gamma}) &= \tfrac{1}{8}\operatorname{Tr}(\delta\Gamma g\delta\tilde{\Gamma}g) = \tfrac{1}{8}\delta\Gamma^{ab}g_{bc}\delta\tilde{\Gamma}^{cd}g_{da}, \\ \boldsymbol{\omega}(\delta\Gamma,\delta\tilde{\Gamma}) &= \tfrac{1}{8}\operatorname{Tr}(\delta\Gamma g\delta\tilde{\Gamma}\omega) = \tfrac{1}{8}\delta\Gamma^{ab}g_{bc}\delta\tilde{\Gamma}^{cd}\omega_{da}, \end{aligned} \tag{56}$$

which establishes relationships between $\boldsymbol{g}$, $\boldsymbol{\omega}$, $g$ and $\omega$.

Given a Gaussian state vector $|J\rangle$ and a Lie algebra element $K \in \mathfrak{g}$, we compute the induced variation

$$\delta\Gamma_K = \frac{d}{dt}\Big|_{t=0} e^{tK}\Gamma e^{tK^{\intercal}} = K\Gamma + \Gamma K^{\intercal}, \tag{57}$$

$$\delta J_K = \frac{d}{dt}\Big|_{t=0} e^{tK}Je^{-tK} = [K,J]. \tag{58}$$

This is the linear map $\delta\Gamma_K : \mathfrak{g} \to \mathcal{T}_\Gamma\mathcal{M} : K \mapsto \delta\Gamma_K$. Its kernel consists of all Lie algebra elements that do not change the covariance matrix $\Gamma$ and is thus

$$\mathfrak{u}(N) = \left\{ K \in \mathfrak{g} \,\middle|\, [K,J] = 0 \right\} \tag{59}$$

from (43). We define its orthogonal complement[12]

$$\mathfrak{u}_\perp(N) = \left\{ K \in \mathfrak{g} \,\middle|\, \{K,J\} = 0 \right\}, \tag{60}$$

which is isomorphic to the tangent space $\mathcal{T}_\Gamma\mathcal{M}$. This will allow us to exploit the group structure of Gaussian states to compute gradient descent with respect to $\boldsymbol{g}$ and symplectic evolution with respect to $\boldsymbol{\omega}$ without needing to evaluate them at every step.

**Example 5** (Tangent space for bosons). *We reconsider a single bosonic mode from example 4 at the state vector $|J_0\rangle$ with $\Gamma_0$ and $J_0$ defined in (51). The tangent space can be parametrized as*

$$\delta G \overset{q,p}{\equiv} \begin{pmatrix} a & b \\ b & -a \end{pmatrix} \overset{a,a^\dagger}{\equiv} \begin{pmatrix} a+\mathrm{i}\,b & 0 \\ 0 & a-\mathrm{i}\,b \end{pmatrix}, \tag{61}$$

$$\delta J \overset{q,p}{\equiv} \begin{pmatrix} -b & a \\ a & b \end{pmatrix} \overset{a,a^\dagger}{\equiv} \begin{pmatrix} 0 & b+\mathrm{i}\,a \\ b-\mathrm{i}\,a & 0 \end{pmatrix}. \tag{62}$$

---

[12]It is the genuine orthogonal complement on the Lie algebra $\mathfrak{g}$ with respect to the Killing form $\mathcal{K}(K,\tilde{K}) = 2N\operatorname{Tr}(K\tilde{K})$.

The associated Hilbert space vector $|\delta\Gamma\rangle = \delta\Gamma^{ab}|V_{ab}\rangle$ is

$$|\delta\Gamma\rangle = \frac{a+\mathrm{i}\,b}{4}\hat{a}^{2\dagger}|J_0\rangle\,. \tag{63}$$

We further find $\langle\delta\Gamma|\delta\tilde{\Gamma}\rangle = \frac{a\tilde{a}+b\tilde{b}}{8} + \mathrm{i}\frac{a\tilde{b}-b\tilde{a}}{8}$ which implies

$$\boldsymbol{g}(\delta\Gamma,\delta\tilde{\Gamma}) = \frac{a\tilde{a}+b\tilde{b}}{4}\quad\text{and}\quad\boldsymbol{\omega}(\delta\Gamma,\delta\tilde{\Gamma}) = \frac{a\tilde{b}-b\tilde{a}}{4}\,. \tag{64}$$

**Example 6** (Tangent space for fermions)**.** *We reconsider Example 4. For a single fermionic mode, the tangent space is trivial,* i.e., *zero-dimensional, because the set of pure Gaussian states consists of two discrete elements. We therefore directly consider two fermionic modes with reference state vector $|J_0\rangle$ defined in (51). The tangent space is then parametrized as*

$$\delta\Omega \overset{q,p}{\equiv} \begin{pmatrix} & & a & b \\ & & b & -a \\ -a & -b & & \\ -b & a & & \end{pmatrix} \overset{a,a\dagger}{\equiv} \begin{pmatrix} & & -\mathrm{i}a & -\mathrm{i}b \\ & & -\mathrm{i}b & \mathrm{i}a \\ \mathrm{i}a & \mathrm{i}b & & \\ \mathrm{i}b & -\mathrm{i}a & & \end{pmatrix}, \tag{65}$$

$$\delta J \overset{q,p}{\equiv} \begin{pmatrix} & & a & b \\ & & b & -a \\ -a & -b & & \\ -b & a & & \end{pmatrix} \overset{a,a\dagger}{\equiv} \begin{pmatrix} -\mathrm{i}a & -\mathrm{i}b & & \\ -\mathrm{i}b & \mathrm{i}a & & \\ & & \mathrm{i}a & \mathrm{i}b \\ & & \mathrm{i}b & -\mathrm{i}a \end{pmatrix}. \tag{66}$$

*The associated Hilbert space vector $|\delta\Gamma\rangle = \delta^{ab}|V_{ab}\rangle$ is*

$$|\delta\Gamma\rangle = -\frac{\mathrm{i}}{2}(a+\mathrm{i}\,b)\hat{a}_1^\dagger\hat{a}_2^\dagger|J\rangle\,. \tag{67}$$

*We further find $\langle\delta\Gamma|\delta\tilde{\Gamma}\rangle = \frac{a\tilde{a}+b\tilde{b}}{4} + \mathrm{i}\frac{a\tilde{b}-b\tilde{a}}{4}$ which implies*

$$\boldsymbol{g}(\delta\Gamma,\delta\tilde{\Gamma}) = \frac{a\tilde{a}+b\tilde{b}}{2}\quad\text{and}\quad\boldsymbol{\omega}(\delta\Gamma,\delta\tilde{\Gamma}) = \frac{a\tilde{b}-b\tilde{a}}{2}\,. \tag{68}$$

## 2.5 Parametrization of Gaussian states

In the previous sections, we saw that a Gaussian state vector $|J\rangle$ is uniquely (up to a complex phase) characterized by its complex structure $J$. For our purpose, it is more efficient to parametrize Gaussian states by first choosing a reference complex structure $J_0$ and then label the Gaussian state vector $|J_M\rangle$ by the group transformation $M$, such that $J_M = MJ_0M^{-1}$. While $M$ is not unique for a given $J_M$, i.e., the map $M \mapsto J_M$ is not injective, it suffices for the purpose of optimization if we can efficiently compute gradients on the group manifold.

We will decompose the space of directions on the group into redundant directions (not changing the state) and non-redundant directions (that change the state). This space is called tangent space $\mathcal{T}_{J_M}\mathcal{M}$ and it can be described by the allowed variations $\delta J_M$ of the complex structure $J_M$. These variations are not completely free, because we only allow variations that respect the conditions on a complex structure $J$, i.e., $J^2 = -\mathbb{1}$ and that are compatible with symplectic form $\Omega$ or metric $G$.

For a reference $J_0$ and a group element $M$, we can associate to every Lie algebra element $K \in \mathfrak{g}$ the variation

$$\delta J_M(K) = M[K,J_0]M^{-1}\,. \tag{69}$$

This is the change induced from moving along $Me^{tK}$ away from $J_M$.

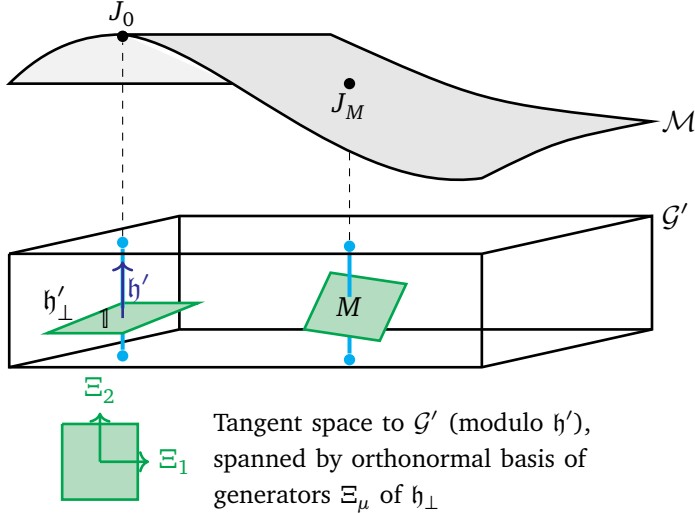

Figure 1: *Parametrization of Gaussian states.* We fix a (pure) reference Gaussian state vector $|J_0\rangle$ and then use the subgroup $\mathcal{G}' \subset \mathcal{G}$ to generate the manifold $\mathcal{M}$ described by complex structures $J_M = M J_0 M^{-1}$ for $M \in \mathcal{G}'$.

In some situations, we may not wish to parametrize the full manifold of Gaussian states, but only a subset. This applies in particular when optimizing over Gaussian purifications $|J\rangle$ of a mixed Gaussian state $\rho_A$, *i.e.*, we require $\rho_A = \mathrm{Tr}_{\mathcal{H}_{A'}} |J\rangle \langle J|$ for Hilbert spaces $\mathcal{H} = \mathcal{H}_A \otimes \mathcal{H}_{A'}$. Any other Gaussian purification $|\tilde{J}\rangle$ of $\rho_A$ is related to $|J\rangle$ by a Gaussian transformation

$$|\tilde{J}\rangle = \mathcal{S}(\mathbb{1}_A \oplus M_{A'}) |J\rangle = \mathcal{S}_{A'}(M_{A'}) |J\rangle \,, \tag{70}$$

*i.e.*, the set of purifications of $\rho_A$ is generated from $|J\rangle$ by the subgroup

$$\mathcal{G}' = \{\mathbb{1}_A \oplus M_{A'} \in \mathcal{G}\} \subset \mathcal{G} \,. \tag{71}$$

This group only affects the Hilbert space $\mathcal{H}_{A'}$, such that the reduction of $|J'\rangle$ onto $\mathcal{H}_A$ will not change and thus stay to be $\rho_A$. In summary, we will consider a subalgebra $\mathfrak{g}' \subset \mathfrak{g}$ that generates the allowed transformations, *e.g.*, the ones only changing the subsystem $\mathcal{H}_{A'}$. Analogous to the decomposition $\mathfrak{g} = \mathfrak{u}(N) \oplus \mathfrak{u}_{\perp}(N)$, we can then define

$$\begin{aligned}
\mathfrak{h}' &= \left\{ K \in \mathfrak{g}' \,\middle|\, [K, J_0] = 0 \right\} \,, \\
\mathfrak{h}'_{\perp} &= \left\{ K \in \mathfrak{g}' \,\middle|\, \{K, J_0\} = 0 \right\} \,,
\end{aligned} \tag{72}$$

such that $\mathfrak{g}' = \mathfrak{h}' \oplus \mathfrak{h}'_{\perp}$. In this case, a basis of $\mathfrak{h}'_{\perp}$ consists of a maximal set of generators $\Xi_\mu \in \mathfrak{g}'$ that lead to linearly independent changes of the state $J_0$.

In summary, our parametrization of Gaussian states or subfamilies is based on the following ingredients, which are illustrated in fig. 1.

- **Reference state vector $|J_0\rangle$.** We specify a pure Gaussian state vector $|J_0\rangle$ as reference by its complex structure $J_0$.

- **Subalgebra $\mathfrak{g}'$ of allowed transformations.** We specify a subalgebra $\mathfrak{g}' \subset \mathfrak{g}$ that we use to generate any other allowed state (potentially $\mathfrak{g}' = \mathfrak{g}$).

- **Generated subgroup $\mathcal{G}'$.** The Lie subalgebra $\mathfrak{g}'$ generates the Lie subgroup $\mathcal{G}' \subset \mathcal{G}$.

- **Manifold of certain Gaussian states $\mathcal{M}$.** The resulting subgroup $\mathcal{G}'$ generates all reachable complex structures $J_M = M J_0 M^{-1}$ and the associated state vectors $|J_M\rangle$.

- **Stabilizer $\mathfrak{h}'$ of $J_0$.** We define the subalgebra $\mathfrak{h}' = \{K \in \mathfrak{g}' \,|\, [K, J_0] = 0\} \subset \mathfrak{g}'$ generates those transformation that leave $J_0$ invariant.

- **Subspace $\mathfrak{h}'_\perp$ changing $J_0$.** We define the subspace $\mathfrak{h}'_\perp = \{K \in \mathfrak{g}' \,|\, \{K, J_0\} = 0\}$ of generators $K$ that are orthogonal to the space $\mathfrak{h}'$.

- **Tangent space $\mathcal{T}_{J_M}\mathcal{M}$.** We span the tangent space at a given Gaussian state $J_M = M J_0 M^{-1}$ as $\delta J_i = M[K_i, J_0]M^{-1}$ with $K_i \in \mathfrak{h}'_\perp$.

We therefore choose a basis $\Xi \equiv (\Xi_1, \ldots, \Xi_m)$ of $\mathfrak{h}'_\perp$ and then compute

$$g_{\mu\nu} = g(\delta\Gamma_\mu, \delta\Gamma_\nu) \quad \text{and} \quad \omega_{\mu\nu} = \omega(\delta\Gamma_\mu, \delta\Gamma_\nu), \tag{73}$$

where $\delta\Gamma_\mu = \Xi_\mu\Gamma_0 + \Gamma_0\Xi_\mu^{\mathsf{T}}$. We can simplify this to find

$$g_{\mu\nu} = \frac{1}{4}\,\mathrm{Tr}(\Xi_\mu\Xi_\nu + \Xi_\mu J_0 \Xi_\nu J_0), \tag{74}$$

$$\omega_{\mu\nu} = \frac{1}{2}\,\mathrm{Tr}(\Xi_\mu J_0 \Xi_\nu), \tag{75}$$

where we have unified the expressions for bosons and fermions from (56).

## 2.6 Purification of mixed Gaussian states

An important class of sub-manifolds of pure Gaussian states that are related by the action of some subgroup $\mathcal{G}' \subset \mathcal{G}$ are Gaussian purifications of a given mixed Gaussian state $\rho$. Various measures of quantum correlations, such as entanglement of purification (EoP) or complexity of purification (CoP), are defined as some critical value on such manifolds, which we review in section 5. Here, we discuss the properties of the underlying manifold of Gaussian purifications.

In section 2.2, we focused on pure Gaussian states, which we introduced as those states, for which the complex structure $J$ satisfies $J^2 = -\mathbb{1}$. A mixed Gaussian state $\rho$ is still fully characterized by $J$ as computed in (13), but which now satisfies the condition

$$\begin{aligned} \mathbb{1} &\le -J^2, &\textbf{(bosons)} \\ 0 &\le -J^2 \le \mathbb{1}. &\textbf{(fermions)} \end{aligned} \tag{76}$$

This implies that the eigenvalues of $J$ appear in conjugate pairs $\pm i c_i$ with $c_i \in [1, \infty)$ for bosons and $c_i \in [0, 1]$ for fermions. We do not refer to such $J$ as complex structures (unless $J^2 = -\mathbb{1}$, *i.e.*, all $c_i = 1$, in which case $\rho$ is pure), but rather as a *restricted complex structure*. As all the eigenvalues are imaginary, we can diagonalize $J$ only over the complex numbers. If we only use real transformations, we can only bring $J$ into a block-diagonal form with antisymmetric $2 \times 2$ blocks.

In contrast to pure Gaussian states, it is not sufficient to require (76) to ensure that $\rho$ is Gaussian. Instead, we need to require that

$$\rho = \begin{cases} e^{-q_{ab}\hat{\xi}^a\hat{\xi}^b - c_0} & \textbf{(bosons)} \\ e^{-iq_{ab}\hat{\xi}^a\hat{\xi}^b - c_0} & \textbf{(fermions)} \end{cases}, \tag{77}$$

where $q_{ab}$ is a positive-definite bilinear form, *i.e.*, $\rho$ is the exponential of a quadratic operator ($c_0$ fixes the normalization $\mathrm{Tr}(\rho) = 1$). We will later see in formula 8 how $J$, $q$ and $c_0$ are related.

We now consider purifications of Gaussian states. For this, we refer to the original Hilbert space as $\mathcal{H}_A$ with $2N_A$ associated operators $\hat{\xi}_A^a$ and classical phase space $A$. We consider a

mixed Gaussian state $\rho_A$ in this system with restricted complex structure $J_A$. From our previous discussion of the eigenvalues $\pm i c_i$ of $J_A$, we can find for every fixed basis $\hat{\xi}_A$ a group transformation $T_A \in \mathcal{G}_A$ (symplectic or orthogonal transformation on $A$) with

$$J_A \equiv T_A J^{\mathrm{m}}_{\mathrm{sta}} T_A^{-1}, \tag{78}$$

with respect to $\hat{\xi}_A$. We have the mixed state standard form

$$
J^{\mathrm{m}}_{\mathrm{sta}} \equiv \begin{pmatrix} \cosh(2r_1)\mathbb{A}_2 & \cdots & 0 \\ \vdots & \ddots & \vdots \\ 0 & \cdots & \cosh(2r_{N_A})\mathbb{A}_2 \end{pmatrix}, \quad \textbf{(bosons)}
$$
$$
J^{\mathrm{m}}_{\mathrm{sta}} \equiv \begin{pmatrix} \cos(2r_1)\mathbb{A}_2 & \cdots & 0 \\ \vdots & \ddots & \vdots \\ 0 & \cdots & \cos(2r_{N_A})\mathbb{A}_2 \end{pmatrix} \quad \textbf{(fermions)}
\tag{79}
$$

with squeezing parameters $r_i$ and the $2 \times 2$ matrix

$$\mathbb{A}_2 \overset{q,p}{\equiv} \begin{pmatrix} 0 & 1 \\ -1 & 0 \end{pmatrix} \overset{a,a^\dagger}{\equiv} \begin{pmatrix} -i & 0 \\ 0 & i \end{pmatrix}. \tag{80}$$

It is well-known that such a mixed Gaussian state $\rho_A$ can be purified by adding more degrees of freedom. For this, we extend the Hilbert space from $\mathcal{H}_A$ to $\mathcal{H}' = \mathcal{H}_A \otimes \mathcal{H}_{A'}$ with operators $\hat{\xi}' = (\hat{\xi}_A, \hat{\xi}_{A'})$. The purification of $\rho_A$ is then a state vector $|J\rangle$ in the larger Hilbert space $\mathcal{H}'$, such that

$$\rho_A = \mathrm{Tr}_{\mathcal{H}_{A'}} |J\rangle \langle J|. \tag{81}$$

This requires $\mathcal{H}_{A'}$ to be sufficiently large, such that all mixed modes can be purified. In light of our previous considerations involving the squeezing parameters $r_i$, system $\mathcal{H}_{A'}$ must contain at least as many modes $N_2$ as there are non-zero parameters $r_i$ associated in the standard form of $J_A$ to $\rho_A$. The Gaussian purification can then be inferred by constructing the complex structure $J$ on the larger phase space $A \oplus A$, such that the restriction $[J]_A$ to $A$ yields $J_A$. Put differently, every non-zero squeezing parameter $r_i$ corresponds to an individual bosonic or fermionic degree of freedom that can and needs to be purified by correlating it with an additional auxiliary degree of freedom. Of course, we are always free to add even more auxiliary modes that are uncorrelated.

The resulting purified form of $J$ with respect to an enlarged basis $\hat{\xi}' = (\hat{\xi}_A, \hat{\xi}_{A'})$ is given by

$$J \equiv (T_A \oplus T_{A'}) J^{\mathrm{p}}_{\mathrm{sta}} (T_A^{-1} \oplus T_{A'}^{-1}), \tag{82}$$

for arbitrary $T_{A'} \in \mathcal{G}_{A'}$, i.e., any such $T_{A'}$ will lead to a valid purification $|J\rangle$. The purified

standard form $J^{\mathrm{p}}_{\mathrm{sta}}$ has been derived in [25] as

$$
J^{\mathrm{p}}_{\mathrm{sta}} = \left(\begin{array}{ccc|ccc|ccc}
\cosh(2r_1)\mathbb{A}_2 & \cdots & 0 & \sinh(2r_1)\mathbb{S}_2 & \cdots & 0 & 0 & \cdots & 0 \\
\vdots & \ddots & \vdots & \vdots & \ddots & \vdots & \vdots & \ddots & \vdots \\
0 & \cdots & \cosh(2r_{N_A})\mathbb{A}_2 & 0 & \cdots & \sinh(2r_{N_A})\mathbb{S}_2 & 0 & \cdots & 0 \\
\hline
\sinh(2r_1)\mathbb{S}_2 & \cdots & 0 & \cosh(2r_1)\mathbb{A}_2 & \cdots & 0 & 0 & \cdots & 0 \\
\vdots & \ddots & \vdots & \vdots & \ddots & \vdots & \vdots & \ddots & \vdots \\
0 & \cdots & \sinh(2r_{N_A})\mathbb{S}_2 & 0 & \cdots & \cosh(2r_{N_A})\mathbb{A}_2 & 0 & \cdots & 0 \\
0 & \cdots & 0 & 0 & \cdots & 0 & \mathbb{A}_2 & \cdots & 0 \\
\vdots & \ddots & \vdots & \vdots & \ddots & \vdots & \vdots & \ddots & \vdots \\
0 & \cdots & 0 & 0 & \cdots & 0 & 0 & \cdots & \mathbb{A}_2
\end{array}\right), \quad \textbf{(bosons)}
$$

$$
J^{\mathrm{p}}_{\mathrm{sta}} = \left(\begin{array}{ccc|ccc|ccc}
\cos(2r_1)\mathbb{A}_2 & \cdots & 0 & \sin(2r_1)\mathbb{S}_2 & \cdots & 0 & 0 & \cdots & 0 \\
\vdots & \ddots & \vdots & \vdots & \ddots & \vdots & \vdots & \ddots & \vdots \\
0 & \cdots & \cos(2r_{N_A})\mathbb{A}_2 & 0 & \cdots & \sin(2r_{N_A})\mathbb{S}_2 & 0 & \cdots & 0 \\
\hline
-\sin(2r_1)\mathbb{S}_2 & \cdots & 0 & \cos(2r_1)\mathbb{A}_2 & \cdots & 0 & 0 & \cdots & 0 \\
\vdots & \ddots & \vdots & \vdots & \ddots & \vdots & \vdots & \ddots & \vdots \\
0 & \cdots & -\sin(2r_{N_A})\mathbb{S}_2 & 0 & \cdots & \cos(2r_{N_A})\mathbb{A}_2 & 0 & \cdots & 0 \\
0 & \cdots & 0 & 0 & \cdots & 0 & \mathbb{A}_2 & \cdots & 0 \\
\vdots & \ddots & \vdots & \vdots & \ddots & \vdots & \vdots & \ddots & \vdots \\
0 & \cdots & 0 & 0 & \cdots & 0 & 0 & \cdots & \mathbb{A}_2
\end{array}\right), \quad \textbf{(fermions)}
$$

$$ \tag{83} $$

where we used the $2 \times 2$ matrices

$$
\mathbb{A}_2 \overset{q,p}{\equiv} \begin{pmatrix} 0 & 1 \\ -1 & 0 \end{pmatrix} \overset{a,a^\dagger}{\equiv} \begin{pmatrix} -\mathrm{i} & 0 \\ 0 & \mathrm{i} \end{pmatrix}, \quad \mathbb{S}_2 \overset{q,p}{\equiv} \begin{pmatrix} 0 & 1 \\ 1 & 0 \end{pmatrix} \overset{a,a^\dagger}{\equiv} \begin{pmatrix} 0 & \mathrm{i} \\ -\mathrm{i} & 0 \end{pmatrix}. \tag{84}
$$

Let us now discuss how unique a chosen purification is. From the perspective of pure states, we can act with arbitrary unitary operators $U = \mathbb{1}_1 \otimes U_2$ on the state vector $|J\rangle$, *i.e.*, $|\psi_U\rangle = U|J\rangle$, while preserving the property $\rho_A = \mathrm{Tr}_{\mathcal{H}_{A'}}|\psi_U\rangle\langle\psi_U|$. This is well-known in the context of the Schmidt-decomposition of $|J\rangle$. However, acting with such a general $U$ will generally lead to a non-Gaussian state vector $|\psi_U\rangle$. Instead, we restrict to Gaussian unitaries of the form $\mathcal{S}(M) = \mathcal{S}_1(\mathbb{1}_A \oplus M_{A'})$ with $M_{A'} \in \mathcal{G}_{A'}$, where $\mathcal{S}(M)$ has been introduced in Section 2.3. Here, we used the representation theory of the Lie group $\mathcal{G}$, *i.e.*, the symplectic group for bosonic systems and the orthogonal group for fermionic ones. From the requirement that $\mathcal{S}(M)$ must act as the identity on $\mathcal{H}_A$, we have

$$
M = \mathbb{1}_A \oplus M_{A'}. \tag{85}
$$

We therefore recognize exactly the setup described in Section 2.5 with the subgroup $\mathcal{G}'$ from (71).

We can use the standard form of the purified $J$ to find the subalgebra $\mathfrak{h}' \subset \mathfrak{g}'$ that preserves $J$. More precisely, we have

$$
\mathfrak{h}' = \left\{ K = \mathbb{0}_1 \oplus K_2 \,\middle|\, [K, J] = 0 \right\}. \tag{86}
$$

If the restriction $J_{A'} = [J]_{A'}$ would describe a pure Gaussian state, the resulting algebra $\mathfrak{h}'$ would be isomorphic to $\mathfrak{u}(N_{A'})$. However, for a mixed state complex structure $J_A$, the algebra $\mathfrak{h}'$ will be smaller, which consequently means that its orthogonal complement $\mathfrak{h}'_\perp$ of those Lie algebra elements that change the state vector $|J\rangle\langle J|$ will be of higher dimension than $\mathfrak{u}_\perp(N_2)$.

It turns out that we have $\mathfrak{h}' = 0$ if there are no $r_i = 0$. Otherwise, we have $\mathfrak{h}' \simeq \mathfrak{u}(N_0)$ if there are $N_0$ distinct parameters $r_i = 0$, which correspond to the allowed Gaussian unitaries that change the pure Gaussian state contained in $\rho_A$. Put differently, we have

$$
\begin{aligned}
\rho_A &= \mathrm{Tr}_{\mathcal{H}_{A'}}|J\rangle\langle J| = \rho_{r_1} \otimes \cdots \otimes \rho_{r_{N_A}}, \\
\rho_{A'} &= \mathrm{Tr}_{\mathcal{H}_A}|J\rangle\langle J| = \rho_{r_1} \otimes \cdots \otimes \rho_{r_{N_A}} \otimes \underbrace{\rho_0 \otimes \cdots \otimes \rho_0}_{N_{A'} - N_A \text{ times}},
\end{aligned} \tag{87}
$$

where $\rho_r$ is generally a mixed Gaussian state of a single bosonic or fermionic mode with

$$
\rho_r = \begin{cases} \frac{1}{\sinh r \cosh r}\, e^{-\hat{n}\ln\coth r} & \textbf{(bosons)} \\ \sin r \cos r \;\, e^{-\hat{n}\ln\tan r} & \textbf{(fermions)} \end{cases}, \tag{88}
$$

where $\hat{n}$ is the number operator of a single bosonic or fermionic mode associated to the creation and annihilation operator of the corresponding block of $J_A$ or $J_{A'}$ in their block-diagonal form.

In summary, provided that all $r_i \neq 0$, we have $\mathfrak{h}' = 0$, such that the set of orthogonal generators is given by

$$
\mathfrak{h}'_\perp = 0 \oplus \mathfrak{g}_{A'} = \{0 \oplus K_{A'} \,|\, K_{A'} \in \mathfrak{g}_{A'}\}, \tag{89}
$$

where $\mathfrak{g}_{A'}$ are the generators of $\mathcal{G}_{A'}$. In this specific case, this orthogonal set forms itself a Lie algebra. Note that the the prime in $\mathfrak{h}'$ and $\mathfrak{h}'_\perp$ is not related to the prime in $A'$.

## 3 Representations of Gaussian states

The literature on quantum physics is for good reason full of Gaussian states for bosonic and fermionic systems and they appear under various names and they a multitude of different forms. In this section, we attempt to provide a comprehensive dictionary that collects the commonly used notions of characterizing Gaussian states and explains how to convert between them. Table 2 provides a summary of these notions, which we review in the following sections including compact conversion formulas. We restrict to Gaussian states with $z^a = \langle J|\hat{\xi}^a|J\rangle = 0$, but it is relatively straightforward by incorporating such displacements for bosons if necessary.

All conversions are based on standard linear algebra operations, *i.e.*, matrix addition and multiplication, evaluation of eigenvalues and so on. Our formulas will include matrix functions $f(M)$ which can be either evaluated as power series or by applying $f$ on the eigenvalues of $M$. Note that we do not require $M$ to be symmetric or Hermitian, as it suffices that either the power series of $f(x)$ converges or that $M$ is diagonalizable for $f(M)$ to be well-defined.

In our formulas, we take great care to make any additional structures explicit. For example, instead of writing a formula where we implicitly assume that the respective basis to be orthonormal with respect to some reference inner product, we will write a basis independent (covariant) formula, which explicitly includes the relevant metric. If we then express the formula with respect to an orthonormal basis, the matrix $g$ representing the metric is just the identity.

### 3.1 Covariance matrix

Given any basis $\{\hat{\xi}^a\}$ of (possibly complexified) linear observables, we compute the bosonic covariance matrix $G^{ab}$ and the fermionic covariance matrix $\Omega^{ab}$ of a Gaussian state vector $|J\rangle$ with $\langle\Gamma|\hat{\xi}^a|\Gamma\rangle = 0$ as defined in (19) based on the following formula.

**Formula 1** (Covariance matrices of pure Gaussian states). *A Gaussian state vector $|J\rangle$ is fully characterized by its bosonic or fermionic covariance matrix defined as*

$$
\Gamma^{ab} = \begin{cases} G^{ab} = \langle J|(\hat{\xi}^a\hat{\xi}^b + \hat{\xi}^b\hat{\xi}^a)|J\rangle & \textbf{(bosons)} \\ \Omega^{ab} = \langle J|(\hat{\xi}^a\hat{\xi}^b - \hat{\xi}^b\hat{\xi}^a)|J\rangle & \textbf{(fermions)} \end{cases}. \tag{90}
$$

We clearly have $G^{ba} = G^{ab}$ and $\Omega^{ba} = -\Omega^{ab}$. The covariance matrix is the (anti-)symmetrised autocorrelator of the quadrature operators and a key characteristic of Gaussian states is that they are unambiguously defined by their first and second moments (the displacement in phase space and covariance matrix) only [1–4].

Table 2: *Common representations of Gaussian states.* We list commonly used representations of bosonic and fermionic Gaussian states of both pure and mixed form.

| | Bosons | Fermions |
|---|---|---|
| **Pure state representations** | **Covariance matrix (3.1)** $$G^{ab} = \langle J|(\hat{\xi}^a\hat{\xi}^b + \hat{\xi}^b\hat{\xi}^a)|J\rangle$$ | **Covariance matrix (3.1)** $$\Omega^{ab} = \langle J|(\hat{\xi}^a\hat{\xi}^b - \hat{\xi}^b\hat{\xi}^a)|J\rangle$$ |
| | **Linear complex structure (3.2)** $$\tfrac{1}{2}(\delta^a{}_b + iJ^a{}_b)\hat{\xi}^b|J\rangle = 0$$ | **Linear complex structure (3.2)** $$\tfrac{1}{2}(\delta^a{}_b + iJ^a{}_b)\hat{\xi}^b|J\rangle = 0$$ |
| | **Characteristic function (3.3)** $$\chi_s(w) = \exp\left(-\tfrac{1}{4}w_a(G + sG_0)^{ab}w_b\right)$$ | **Characteristic function (3.3)** $$\chi_s(w) = \exp\left(-\tfrac{i}{4}w_a(\Omega - s\Omega_0)^{ab}w_b\right)$$ |
| | **Quasiprobability distribution (3.4)** $$W_s(\xi) = \frac{\exp(-\xi^a(G+sG_0)^{-1}_{ab}\xi^b)}{\sqrt{\det \pi(G+sG_0)}}$$ | **Quasiprobability distribution (3.4)** $$W_s(\xi) = \frac{\exp(-4i\,\xi^a(\Omega-s\Omega_0)^{-1}_{ab}\xi^b)}{\sqrt{\det \frac{\Omega-s\Omega_0}{2}}}$$ |
| | **Gaussian unitary (3.5)** $$|G\rangle = e^{\widehat{K}}|0\rangle = \exp(-\tfrac{i}{2}k_{ab}\hat{\xi}^a\hat{\xi}^b)|0\rangle$$ | **Gaussian unitary (3.5)** $$|\Omega\rangle = e^{\widehat{K}}|0\rangle = \exp(\tfrac{1}{2}k_{ab}\hat{\xi}^a\hat{\xi}^b)|0\rangle$$ |
| | **Squeezed vacuum (3.6)** $$|J\rangle = \sqrt[4]{\mathbb{1} - \gamma\gamma^\dagger}\,e^{\tfrac{1}{2}\gamma^{ij}\hat{a}_i^\dagger\hat{a}_j^\dagger}|0\rangle = \sqrt[8]{\det(\mathbb{1}-L^2)}\,e^{-\tfrac{i}{2}\omega_{ac}L^c{}_b\hat{\xi}_+^a\hat{\xi}_+^b}|0\rangle$$ | **Squeezed vacuum (3.6)** $$|J\rangle = \frac{e^{\tfrac{1}{2}\gamma^{ij}\hat{a}_i^\dagger\hat{a}_j^\dagger}|0\rangle}{\sqrt[4]{\mathbb{1}+\gamma\gamma^\dagger}} = \frac{e^{\tfrac{1}{2}g_{ac}L^c{}_b\hat{\xi}_+^a\hat{\xi}_+^b}|0\rangle}{\sqrt[8]{\det(\mathbb{1}-L^2)}}$$ |
| | **Bogoliubov transformation (3.7)** $$\hat{b}_i = \alpha_{ij}\hat{a}_j + \beta_{ij}\hat{a}_j^\dagger$$ | **Bogoliubov transformation (3.7)** $$\hat{b}_i = \alpha_{ij}\hat{a}_j + \beta_{ij}\hat{a}_j^\dagger$$ |
| | **Wave function (3.9)** $$\psi(q) = \sqrt[4]{\det\tfrac{A}{\pi}}\,\exp(-\tfrac{1}{2}q^\alpha(A+iB)_{\alpha\beta}q^\beta)$$ | |
| **Mixed state representations** | **Covariance matrix (3.1)** $$G^{ab} = \mathrm{Tr}\left(\rho(\hat{\xi}^a\hat{\xi}^b + \hat{\xi}^b\hat{\xi}^a)\right)$$ | **Covariance matrix (3.1)** $$\Omega^{ab} = \mathrm{Tr}\left(\rho(\hat{\xi}^a\hat{\xi}^b - \hat{\xi}^b\hat{\xi}^a)\right)$$ |
| | **Characteristic function (3.3)** $$\chi_s(w) = \exp\left(-\tfrac{1}{4}w_a(G + sG_0)^{ab}w_b\right)$$ | **Characteristic function (3.3)** $$\chi_s(w) = \exp\left(-\tfrac{i}{4}w_a(\Omega - s\Omega_0)^{ab}w_b\right)$$ |
| | **Quasiprobability distribution (3.4)** $$W_s(\xi) = \frac{\exp(-\xi^a(G+sG_0)^{-1}_{ab}\xi^b)}{\sqrt{\det \pi(G+sG_0)}}$$ | **Quasiprobability distribution (3.4)** $$W_s(\xi) = \frac{\exp(-4i\,\xi^a(\Omega-s\Omega_0)^{-1}_{ab}\xi^b)}{\sqrt{\det \frac{\Omega-s\Omega_0}{2}}}$$ |
| | **Thermal state (3.8)** $$\rho = \exp(-\tfrac{1}{2}q_{ab}\hat{\xi}^a\hat{\xi}^b + c_0)$$ | **Thermal state (3.8)** $$\rho = \exp(-\tfrac{i}{2}q_{ab}\hat{\xi}^a\hat{\xi}^b + c_0)$$ |
| | **Wave function (3.9)** $$\rho(q,\bar{q}) = \sqrt{\det\tfrac{A+C}{\pi}}\,\exp\left(-\tfrac{1}{2}\begin{pmatrix}q\\\bar{q}\end{pmatrix}^{\mathsf{T}}\begin{pmatrix}A+iB & C+iD\\ C-iD & A-iB\end{pmatrix}\begin{pmatrix}q\\\bar{q}\end{pmatrix}\right)$$ | |

## 3.2 Linear complex structure

An alternative to parametrizing Gaussian states by their covariance matrix is to use the so called linear complex structure. It is less commonly used in quantum information and condensed matter, but has been extensively studied in the context of quantum field theory in curved spacetime [12].

**Formula 2** (Linear complex structure). *Given a bosonic Gaussian state vector $|J\rangle$ with symplectic form $\Omega^{ab}$ or a fermionic Gaussian state vector $|\Omega\rangle$ with metric $G^{ab}$, the associated linear complex $J$ is*

$$J^a{}_b = -G^{ac}\Omega^{-1}_{cb} = \Omega^{ac}G^{-1}_{cb}. \tag{91}$$

We will see that the compatibility of these three structures shown in (91) imbues the manifold of pure Gaussian states with the structure of a Kähler manifold.

## 3.3 Characteristic functions

The characteristic function $\chi$ of a quasiprobability distribution $W$ on the phasespace $V$ is defined as the inverse Fourier transform

$$\chi : V^* \to \mathbb{C} : v \mapsto \chi(w) = \int_V d^{2N}\xi \, e^{-\mathrm{i}w_a\xi^a} W(\xi). \tag{92}$$

We see that $\chi$ is defined on the dual phase space $V^*$. From the perspective of probability theory, one usually first defines $W$, from which $\chi$ is derived. However, in the context of quantum theory, it is actually easier to first give explicit formulas for $\chi$ and then define $W$ as its Fourier transform. This is why we first present the results for $\chi$ in the present subsection and then discuss the respective $W$ in the next subsection. Note that for fermions, both $\chi$ and $W$ are defined as a polynomial in Grassman variables $w_a$ and $\xi^a$, which anti-commute among themselves and with each other in (92).

In the case of bosonic and fermionic quantum systems, there is a natural set of quasi probability distributions $W_s$ and their associated characteristic functions $\chi_s$ labelled by a real parameter $s \in [-1, 1]$. For $s \neq 0$, they are defined with respect to a notion of ordering creation and annihilation operators associated to a Gaussian reference state $|J_0\rangle$ with covariance matrix $\Gamma_0$. For most practical applications, only the following cases of $s \in \{-1, 0, 1\}$ are studied.

**Wigner.** For $s = 0$, the quasiprobability distribution is independent of $\Gamma_0$ and called *Wigner distribution* $\xi \mapsto W_0(\xi)$. The Wigner characteristic function $\chi_0(w)$ of an operator $\hat{\mathcal{O}}$ can be computed from the operator $\hat{\mathcal{O}}$ as

$$\chi_0(w) = \mathrm{Tr}(\hat{\mathcal{O}}e^{-\mathrm{i}w_a\hat{\xi}^a}). \tag{93}$$

**Glauber.** For $s = 1$, we have the Glauber–Sudarshan $P$ characteristic function, which is the Fourier transform of the Glauber–Sudarshan $P$ quasiprobability distribution $\xi \mapsto P(\xi) = W_{-1}$. The characteristic function is

$$\chi_1(w) = \mathrm{Tr}(\hat{\mathcal{O}}e^{-\mathrm{i}w_a\hat{\xi}^a_+}e^{-\mathrm{i}w_a\hat{\xi}^a_-}). \tag{94}$$

**Husimi.** For $s = -1$, we have the Husimi $Q$ characteristic function, which is the Fourier transform of the Glauber–Sudarshan $P$ quasiprobability distribution $\xi \mapsto P(\xi) = W_{-1}$. The characteristic function is

$$\chi_{-1}(w) = \mathrm{Tr}(\hat{\mathcal{O}}e^{-\mathrm{i}w_a\hat{\xi}^a_-}e^{-\mathrm{i}w_a\hat{\xi}^a_+}). \tag{95}$$

In all these cases, we can compute the expectation value of an arbitrary polynomial in linear observables $\{\hat{\xi}^a\}$ as

$$\langle(\hat{\xi}^{a_1}\dots\hat{\xi}^{a_n})_s\rangle = \frac{\partial}{\partial w_{a_1}}\cdots\frac{\partial}{\partial w_{a_1}}\bigg|_{w=0}\chi_s(w), \tag{96}$$

where $(\dots)_s$ refers to the respective ordering with respect to a Gaussian reference state vector $|0\rangle = |J_0\rangle$, *i.e.*, symmetric ordering for $s = 0$, normal-ordering for $s = 1$ and anti-normal ordering for $s = -1$. From this condition, the following forms of $\chi$ for Gaussian states can be derived.

**Formula 3** (Characteristic functions of Gaussian states)**.** *The general formula for the characteristic function of a pure or mixed Gaussian state with respect to the reference state vector $|0\rangle = |J_0\rangle$ is*

$$\chi_s(w) = \begin{cases} \exp\left(-\frac{1}{4}w_a(G + sG_0)^{ab}w_b\right) & \textbf{(bosons)} \\ \exp\left(-\frac{i}{4}w_a(\Omega - s\Omega_0)^{ab}w_b\right) & \textbf{(fermions)} \end{cases}. \tag{97}$$

As can be seen from (92), the characteristic function $\chi$ is defined on the dual phase space $V^*$, while the quasiprobability distribution $W$ is directly defined on the phase space $V$. We can, however, use the isomorphism $w_a \Leftrightarrow \xi^a$ given by $w_a = \omega_{ab}\xi^b$ for bosons and $w_a = g_{ab}\xi^b$ for fermions to map the characteristic function $\chi_s(w)$ into the phase space function $\widetilde{\chi}_s(\xi)$, such that

$$\widetilde{\chi}_s(\xi) = \begin{cases} \chi_s(\omega_{ab}\xi^b) & \textbf{(bosons)} \\ \chi_s(g_{ab}\xi^b) & \textbf{(fermions)} \end{cases}. \tag{98}$$

One can show that for pure Gaussian states $W(\xi) \propto \widetilde{\chi}(\xi)$ for all $\xi$.

## 3.4 Quasiprobability distributions

As foreshadowed in the previous section, bosonic and fermionic quantum states can also be represented as quasiprobability distributions on the classical phase space, *i.e.*, real valued functions $W : V \to \mathbb{R}$ satisfying $\int d^{2N}\xi \, W(\xi) = 1$. In contrast to regular probability distributions, $W(\xi)$ is allowed to also take negative values and in fact, this negativity can be directly linked to the non-classicality of the respective quantum state [26,27]. For Gaussian states, all quasiprobability distributions are themselves Gaussian functions and in particular positive, *i.e.*, classical in the sense of ref. [26].

More generally, operators $\mathcal{O}$ on Hilbert space can be related to a phase space distributions $W : V \to \mathbb{C}$, which may not be normalized. To define $W(\xi^a)$, we need to write the operator $\mathcal{O}$ as a power series

$$\mathcal{O} = t_0 + (t_1)_a \hat{\xi}^a + (t_2)_{ab} \hat{\xi}^a \hat{\xi}^b + \dots \tag{99}$$

in terms of linear observables $\hat{\xi}^a$ (or as limit of a sequence of such series). Clearly, the sequence is not unique, because we can use commutation or anti-commutation relations to change the ordering of $\hat{\xi}^a$, $\hat{\xi}^b$ and so on in (99), which will create additional terms. For example, we have $\hat{q}\hat{p} = \hat{p}\hat{q} + i$. Given a Gaussian reference state vector $|0\rangle = |J_0\rangle$, we can express everything in terms of $\hat{\xi}^a_\pm$, which are defined with respect to $|0\rangle$, and then use commutation relations to bring them into some standard ordering. The most common orderings are

| | |
|---|---|
| symmetric ordering ($s = 0$): | $\frac{1}{2}(\hat{\xi}^a_+ \hat{\xi}^b_- - \hat{\xi}^b_- \hat{\xi}^a_+) + \dots,$ |
| normal ordering ($s = 1$): | $\hat{\xi}^a_+ \hat{\xi}^b_- + \dots,$ |
| anti-normal ordering ($s = -1$): | $\hat{\xi}^a_- \hat{\xi}^b_+ + \dots,$ |

where the parameter $s \in [-1, 1]$ describes a continuum of intermediate orderings, as introduced in ref. [28]. Let us emphasize that we bring the power series (99) by using the canonical commutation or anti-commutation relations and not by just reordering the terms by force, which would change the resulting operator $\mathcal{O}$. We can then express $\xi^a_\pm$ in terms of $\hat{\xi}^a$ via (17) to find the coefficients $(t^s_n)_{a_1 \dots a_n}$ of a series expansion with ordering $s$. Plugging in the variables $\xi \overset{q,p}{\equiv} (q_1, \dots, q_N, p_1, \dots, p_N) \overset{a,a^\dagger}{\equiv} (a_1, \dots, a_N, a^\dagger_1, \dots, a^\dagger_N)$ rather than operators $\hat{\xi}$ then defines the phase space distribution

$$W_s(\xi) = \sum_{n=0}^{\infty} (t^s_n)_{a_1 \dots a_n} \xi^{a_1} \dots \xi^{a_n}. \tag{100}$$

For Hermitian operators $\mathcal{O}$, the associated $W_s$ will be real-valued on $V$. One can further show that we have $\text{Tr}\,\mathcal{O} = \int d\xi^{2N} W_s(\xi)$. For a density operator $\rho$, we thus have $\int \xi^{2N} W_s(\xi) = 1$. Given an observable $\mathcal{O}$ and a density operator $\rho$, we can compute the expectation value

$$\langle \mathcal{O} \rangle_\rho = \text{Tr}(\rho\,\mathcal{O}) = \int d\xi^{2N} W_s^\rho(\xi) W_s^{\mathcal{O}}(\xi), \tag{101}$$

*i.e.*, the trace of the product of two operators can be computed by just integrating over the pointwise product of the respective phase space functions. Note that this formula does not generalize to computing the trace of a product of more than two operators.

In practice, $W_s$ is most efficiently computed from the respective characteristic function $\chi_s$ via the regular Fourier transform

$$W_s(\xi) = \frac{1}{(2\pi)^{2N}} \int dw^{2N} \chi_s(w) e^{i w_a \xi^a}. \tag{102}$$

With this in hand, we can compute the quasi-probability distributions $W_s$ for Gaussian states as follows.

**Formula 4** (Quasiprobability distributions)**.** *For a Gaussian state with covariance matrix $\Gamma$, we have the quasiprobability distribution*

$$W_s(\xi) = \begin{cases} \dfrac{e^{-\xi^a (G+sG_0)^{-1}_{ab} \xi^b}}{\sqrt{\det \pi(G+G_0)}} & \textbf{(bosons)} \\[2ex] \dfrac{e^{i\xi^a (\Omega - s\Omega_0)^{-1}_{ab} \xi^b}}{\sqrt{\det \frac{\Omega - s\Omega_0}{2}}} & \textbf{(fermions)} \end{cases}, \tag{103}$$

*with respect to the reference state vector $|0\rangle = |J_0\rangle$.*

## 3.5 Gaussian unitaries

We can parametrize Gaussian states also by the unitary Gaussian transformation $\mathcal{S}(M)$ that takes us from a reference state (vacuum $|0\rangle = |J_0\rangle$) to the state under consideration, *i.e.*, $|G\rangle$ or $|\Omega\rangle$. This unitary is not unique, because we can always compose $U$ with some other Gaussian unitary satisfying $u\,|0\rangle = e^{i\varphi}\,|0\rangle$.

We have the reference covariance matrix $\Gamma_0$ of the state $|J_0\rangle = |0\rangle$ and a target covariance matrix $\Gamma$ of the state $|J\rangle$, such that the relative complex structure (45) is

$$\Delta^a{}_b = -J^a{}_c (J_0)^c{}_b = \Gamma^{ac}(\Gamma_0^{-1})_{cb}. \tag{104}$$

With this, we can define the group element $T = \sqrt{\Delta}$, satisfying $J = TJ_0 T^{-1}$, from which we can deduce the Lie algebra generator[13]

$$K_+ = \log T = \frac{1}{2}\log \Delta. \tag{105}$$

The unitary transformation $\mathcal{S}$ satisfying $|J\rangle = \mathcal{S}\,|0\rangle$ is

$$\mathcal{S} = e^{\widehat{K}_+} = \begin{cases} \exp\left(-\frac{i}{2}\omega_{ac}(K_+)^c{}_b\,\hat{\xi}^a \hat{\xi}^b\right) & \textbf{(bosons)} \\ \exp\left(\frac{1}{2}g_{ac}(K_+)^c{}_b\,\hat{\xi}^a \hat{\xi}^b\right) & \textbf{(fermions)} \end{cases}. \tag{106}$$

---

[13]We denote it by $K_+$ because if we define $K_\pm = \frac{1}{2}(K \pm J_0 K J_0)$ for any $K$, our choice of $K_+ = \frac{1}{2}\log \Delta$ will be of this type. They are called pure squeezing transformations as explained in [16].

Vice versa, if we know the anti-Hermitian quadratic operator $\widehat{K} = -\frac{\mathrm{i}}{2} h_{ab} \hat{\xi}^a \hat{\xi}^b$ for bosons or $\widehat{K} = \frac{1}{2} h_{ab} \hat{\xi}^a \hat{\xi}^b$ for fermions (which may not be of the type $K_+$), we can compute the associated generator

$$K^a{}_b = \begin{cases} \Omega^{ac} h_{cb} & \textbf{(bosons)} \\ G^{ac} h_{cb} & \textbf{(fermions)} \end{cases} , \tag{107}$$

from which we find the transformed covariance matrix as

$$\Gamma = M \Gamma_0 M^{\intercal} \quad \text{with} \quad M = e^K . \tag{108}$$

In summary, we have the following formulas.

**Formula 5** (Pure Gaussian state transformations). *Given a reference Gaussian state vector $|0\rangle$ with covariance matrix $\Gamma_0$, we can compute for every Gaussian state vector $|J\rangle$ the quadratic operator $\widehat{K}$, such that*

$$|J\rangle = e^{\widehat{K}} |0\rangle \quad \text{with} \quad K = \frac{1}{2} \log \Delta , \tag{109}$$

*where $\Delta^a{}_b = \Gamma^{ac} (\Gamma_0^{-1})_{cb}$. Vice versa, for the same setup (reference state vector $|0\rangle$ with covariance matrix $\Gamma_0$), we compute for every Gaussian unitary $e^{\widehat{K}}$ the covariance matrix*

$$\Gamma = M \Gamma_0 M^{\intercal} \quad \text{with} \quad M = e^K . \tag{110}$$

*Note that all equalities of quantum state vectors are only up to a global complex phase. In particular,*

## 3.6 Squeezed vacuum

Given a bosonic or fermionic Gaussian state vector $|0\rangle$ together with a complete set of annihilation operators $\hat{a}_i$ satisfying $\hat{a}_i |0\rangle = 0$, a Gaussian state vector $|J\rangle$ can be described by a squeezing matrix $\gamma$, which is a complex $N \times N$ matrix that is symmetric for bosons and antisymmetric for fermions. For bosonic systems, we can thereby reach any covariance matrix $G^{ab}$, while for fermionic systems we can reach any covariance matrix $\Omega^{ab}$ with the same parity as explained around (38). In the following, we will derive the relations between $K$, $\gamma$, $\Gamma$ and $\Gamma_0$.

We choose our standard bases such that the Kähler structures associated to reference state $|J_0\rangle$ take the standard forms from (21). We consider $|J\rangle \cong \mathcal{S}(T)|0\rangle$, where $T^2 = \Delta = \Gamma \Gamma_0^{-1}$. By construction, we have $\mathcal{S}(T) = e^{\widehat{K}}$ with $K = \frac{1}{2} \log \Delta$. Here, $K$ takes the standard forms

$$K \overset{q,p}{\cong} \left( \begin{array}{c|c} K_1 & K_2 \\ \hline K_2 & -K_1 \end{array} \right) \overset{a,a\dagger}{\cong} \left( \begin{array}{c|c} 0 & K_1 + \mathrm{i} K_2 \\ \hline K_1 - \mathrm{i} K_2 & 0 \end{array} \right) , \tag{111}$$

which both anti-commute with $J_0$. Note that the decomposition into $K_1$ and $K_2$ still has a U($N$) redundancy, *i.e.*, we would preserve the standard forms (21) of $J_0$, $\Gamma_0$ and $\Omega$ for bosons or $G$ for fermions, while $K_1$ and $K_2$ will mix with each other.

A matrix $u \in$ U($N$) satisfies $[u, J_0] = 0$ and is

$$u \overset{q,p}{\cong} \left( \begin{array}{c|c} u_1 & u_2 \\ \hline -u_2 & u_1 \end{array} \right) \overset{a,a\dagger}{\cong} \left( \begin{array}{c|c} u_1 - \mathrm{i} u_2 & 0 \\ \hline 0 & u_1 + \mathrm{i} u_2 \end{array} \right) . \tag{112}$$

Under a change of basis $K \mapsto u K u^{-1}$, we thus have the change $K_1 + \mathrm{i} K_2 \mapsto (u_1 - \mathrm{i} u_2)(K_1 + \mathrm{i} K_2)(u_1 + \mathrm{i} u_2)$. Mathematically speaking, we have the complex $N$-dimensional vector space $V_{\mathbb{C}}^-$

of annihilation operators and $V_{\mathbb{C}}^+$ of creation operators. The two spaces are embedded in the complexified phase space $V^{\mathbb{C}}$ and can be canonically identified using complex conjugation on $V^{\mathbb{C}}$.

Our goal is to find a compact expression of $|J\rangle$. We consider $K$ with $\{K, J_0\} = 0$, which satisfies

$$
\widehat{K} = \begin{cases}
-\frac{i}{2}\omega_{ac}K^c{}_b(\hat{\xi}_+^a\hat{\xi}_+^b + \hat{\xi}_-^a\hat{\xi}_-^b) & \textbf{(bosons)} \\
\frac{1}{2}g_{ac}K^c{}_b(\hat{\xi}_+^a\hat{\xi}_+^b + \hat{\xi}_-^a\hat{\xi}_-^b) & \textbf{(fermions)}
\end{cases}.
\tag{113}
$$

We can simplify $e^{\widehat{K}}$ based on the known relations

$$
\begin{aligned}
\exp\left[\frac{r}{2}(e^{i\theta}(\hat{a}^\dagger)^2 - e^{-i\theta}\hat{a}^2)\right] &= \exp\left[\frac{1}{2}e^{i\theta}(\tanh r)(\hat{a}^\dagger)^2\right] \\
&\times \exp\left[-(\ln\cosh r)(\hat{n}+\tfrac{1}{2})\right]\exp\left[-\tfrac{1}{2}(e^{-i\theta}\tanh r)\hat{a}^2\right],
\end{aligned}
\qquad \textbf{(bosons)} \tag{114}
$$

$$
\begin{aligned}
\exp\left[r(e^{i\theta}\hat{a}_1^\dagger\hat{a}_2^\dagger + e^{-i\theta}\hat{a}_1\hat{a}_2)\right] &= \exp\left[e^{i\theta}\tan r\,\hat{a}_1^\dagger\hat{a}_2^\dagger\right] \\
&\times \exp\left[-(\ln\cos r)(\hat{n}_1+\hat{n}_2-1)\right]\exp\left[e^{-i\theta}\tan r\,\hat{a}_1\hat{a}_2\right],
\end{aligned}
\qquad \textbf{(fermions)} \tag{115}
$$

which are derived in ref. [29]. Using them and the definition $L = \tanh K$, we find the covariant expressions

$$
e^{\widehat{K}} = e^{-\frac{i}{2}\omega_{ac}L^c{}_b\hat{\xi}_+^a\hat{\xi}_+^b}e^{-\frac{i}{2}\omega_{ac}\log(\mathbb{1}-L^2)^c{}_b(\hat{\xi}_+^a\hat{\xi}_-^b + \frac{i}{4}\Omega^{ba})}e^{-\frac{i}{2}\omega_{ac}L^c{}_b\hat{\xi}_-^a\hat{\xi}_-^b},
\qquad \textbf{(bosons)} \tag{116}
$$

$$
e^{\widehat{K}} = e^{\frac{1}{2}g_{ac}L^c{}_b\hat{\xi}_+^a\hat{\xi}_+^b}e^{\frac{1}{2}g_{ac}\log(\mathbb{1}-L^2)^c{}_b(\hat{\xi}_+^a\hat{\xi}_-^b - \frac{1}{4}G^{ba})}e^{\frac{1}{2}g_{ac}L^c{}_b\hat{\xi}_-^a\hat{\xi}_-^b},
\qquad \textbf{(fermions)} \tag{117}
$$

where we emphasize that they only apply to algebra elements $K \in \mathfrak{g}$ with $\{K, J_0\} = 0$. When applied to $|0\rangle$, we find

$$
|J\rangle = e^{\widehat{K}}|0\rangle = \begin{cases}
\det^{\frac{1}{8}}(\mathbb{1}-L^2)\; e^{-\frac{i}{2}\omega_{ac}L^c{}_b\hat{\xi}_+^a\hat{\xi}_+^b}|0\rangle & \textbf{(bosons)} \\
\det^{-\frac{1}{8}}(\mathbb{1}-L^2)\, e^{\frac{1}{2}g_{ac}L^c{}_b\hat{\xi}_+^a\hat{\xi}_+^b}|0\rangle & \textbf{(fermions)}
\end{cases},
\tag{118}
$$

where we used $e^{\pm\frac{1}{8}\operatorname{Tr}\log(\mathbb{1}-L^2)} = \det^{\pm\frac{1}{8}}(\mathbb{1}-L^2)$. The relevant linear map $L = \tanh K$ takes the form

$$
L \overset{q,p}{\equiv} \left(\begin{array}{c|c} L_1 & L_2 \\ \hline L_2 & -L_1 \end{array}\right) \overset{a,a^\dagger}{\equiv} \left(\begin{array}{c|c} 0 & L_1 + iL_2 \\ \hline L_1 - iL_2 & 0 \end{array}\right),
\tag{119}
$$

which is analogous to (111). We find

$$
L^2 \overset{q,p}{\equiv} \left(\begin{array}{c|c} L_1^2 + L_2^2 & L_1L_2 - L_2L_1 \\ \hline L_2L_1 - L_1L_2 & L_1^2 + L_2^2 \end{array}\right) \overset{a,a^\dagger}{\equiv} \left(\begin{array}{c|c} \gamma^*\gamma & 0 \\ \hline 0 & \gamma\gamma^* \end{array}\right),
\tag{120}
$$

where we defined in this basis the complex matrix

$$
\gamma := L_1 + iL_2.
\tag{121}
$$

The matrix representations $L_1$ and $L_2$ are symmetric for bosons and antisymmetric for fermions. This leads to

$$
\det(\mathbb{1}-L^2) = \begin{cases}
\det^2(\mathbb{1}-\gamma\gamma^\dagger) & \textbf{(bosons)} \\
\det^2(\mathbb{1}+\gamma\gamma^\dagger) & \textbf{(fermions)}
\end{cases},
\tag{122}
$$

where the sign changes due to the anti-symmetry of $L_i$ for fermions. Using the fact that the spaces of $V_{\mathbb{C}}^\pm$ of creation and annihilation operators are Hilbert spaces with Hermitian inner product, we can use $\gamma$ as bilinear form $\gamma^{ij}$, which satisfies

$$
\frac{1}{2}\gamma^{ij}\hat{a}_i^\dagger\hat{a}_j^\dagger = \begin{cases}
-\frac{i}{2}\omega_{ac}L^c{}_b\hat{\xi}_+^a\hat{\xi}_+^b & \textbf{(bosons)} \\
\frac{1}{2}g_{ac}L^c{}_b\hat{\xi}_+^a\hat{\xi}_+^b & \textbf{(fermions)}
\end{cases}.
\tag{123}
$$

This leads to our final formula as follows.

**Formula 6** (Parametrizing squeezed state vectors). *Given a Gaussian reference vacuum $|0\rangle$ with covariance matrix $\Gamma_0$ and creation operators $\hat{a}_i^\dagger$, we can parametrize the squeezed state vector $|J\rangle$ by an arbitrary symmetric complex matrix $\gamma$, such that*

$$|J\rangle = \begin{cases} \left(\det(\mathbb{1} - \gamma^\dagger \gamma)\right)^{\frac{1}{4}} e^{\frac{1}{2}\gamma^{ij}\hat{a}_i^\dagger \hat{a}_j^\dagger}|0\rangle & \textbf{(bosons)} \\[2mm] \left(\det(\mathbb{1} + \gamma^\dagger \gamma)\right)^{-\frac{1}{4}} e^{\frac{1}{2}\gamma^{ij}\hat{a}_i^\dagger \hat{a}_j^\dagger}|0\rangle & \textbf{(fermions)} \end{cases}. \tag{124}$$

*We can seamlessly convert between the matrix $\gamma$ and the covariance $\Gamma$. In particular, we have*

$$L = \tanh\left(\frac{1}{2}\log\Delta\right) \overset{q,p}{\equiv} \left(\begin{array}{c|c} \operatorname{Re}\gamma & \operatorname{Im}\gamma \\ \hline \operatorname{Im}\gamma & -\operatorname{Re}\gamma \end{array}\right) \overset{a,a\dagger}{\equiv} \left(\begin{array}{c|c} 0 & \gamma \\ \hline \gamma^* & 0 \end{array}\right), \tag{125}$$

*which can be used to compute $\gamma$ from $\Delta^a{}_b = \Gamma^{ac}(\Gamma_0^{-1})_{cb}$ or vice versa. Note that the block-decomposition from* (125) *only requires the standard forms of* (21) *of the state vector $|0\rangle$.*

For bosons, when we express $\Omega$ in the real standard basis (*i.e.*, such that $\Omega$ takes the real standard form), we have

$$G^{ab} \overset{q,p}{\equiv} \left(\begin{array}{c|c} G_1^{\alpha\beta} & G_2^{\alpha\dot{\beta}} \\ \hline G_3^{\dot{\alpha}\beta} & G_4^{\dot{\alpha}\dot{\beta}} \end{array}\right), \quad \Omega \overset{q,p}{\equiv} \left(\begin{array}{cc} 0 & \mathbb{1} \\ -\mathbb{1} & 0 \end{array}\right), \tag{126}$$

where $\alpha, \beta, \dot{\alpha}, \dot{\beta}$ are indices running over $1, \dots, N$, while $a = (\alpha, \dot{\alpha})$ and $b = (\beta, \dot{\beta})$ describe the full $2N$-by-$2N$ block. The elements of this bosonic covariance matrix can again be directly expressed in terms of $\gamma$:

$$\begin{aligned} G_1 &= \operatorname{Re}\left[(\mathbb{1}_N + 2\gamma + \gamma^\dagger\gamma)(\mathbb{1}_N - \gamma^\dagger\gamma)^{-1}\right], \\ G_2 &= \operatorname{Im}\left[(\mathbb{1}_N + 2\gamma + \gamma^\dagger\gamma)(\mathbb{1}_N - \gamma^\dagger\gamma)^{-1}\right], \\ G_3 &= \operatorname{Im}\left[(-\mathbb{1}_N + 2\gamma - \gamma^\dagger\gamma)(\mathbb{1}_N - \gamma^\dagger\gamma)^{-1}\right], \\ G_4 &= \operatorname{Re}\left[(\mathbb{1}_N - 2\gamma + \gamma^\dagger\gamma)(\mathbb{1}_N - \gamma^\dagger\gamma)^{-1}\right]. \end{aligned} \tag{127}$$

The inverse operation is given by

$$\gamma = \frac{G_1 - G_4 + \mathrm{i}(G_2 + G_3)}{2\,\mathbb{1}_N + G_1 + G_4 + \mathrm{i}(G_2 - G_3)}. \tag{128}$$

Similarly, for fermions in the real (Majorana) basis (*i.e.*, such that $G \overset{q,p}{\equiv} \mathbb{1}$), we have

$$\Omega^{ab} \overset{q,p}{\equiv} \left(\begin{array}{c|c} \Omega_1^{\alpha\beta} & \Omega_2^{\alpha\dot{\beta}} \\ \hline \Omega_3^{\dot{\alpha}\beta} & \Omega_4^{\dot{\alpha}\dot{\beta}} \end{array}\right), \quad G^{ab} \overset{q,p}{\equiv} \left(\begin{array}{cc} \mathbb{1} & 0 \\ 0 & \mathbb{1} \end{array}\right). \tag{129}$$

In this basis, we find

$$\begin{aligned} \Omega_1 &= \operatorname{Im}\left[2(-\mathbb{1}_N - \gamma)(\mathbb{1}_N + \gamma^\dagger\gamma)^{-1}\right], \\ \Omega_2 &= \operatorname{Re}\left[(\mathbb{1}_N + 2\gamma - \gamma^\dagger\gamma)(\mathbb{1}_N + \gamma^\dagger\gamma)^{-1}\right], \\ \Omega_3 &= \operatorname{Re}\left[(-\mathbb{1}_N + 2\gamma + \gamma^\dagger\gamma)(\mathbb{1}_N + \gamma^\dagger\gamma)^{-1}\right], \\ \Omega_4 &= \operatorname{Im}\left[2(-\mathbb{1}_N + \gamma)(\mathbb{1}_N + \gamma^\dagger\gamma)^{-1}\right]. \end{aligned} \tag{130}$$

Again, this relationship can be inverted, leading to

$$\gamma = \frac{\Omega_2 + \Omega_3 - \mathrm{i}(\Omega_1 - \Omega_4)}{2\,\mathbb{1}_N + \Omega_2 - \Omega_3 - \mathrm{i}(\Omega_1 + \Omega_4)}, \tag{131}$$

where the fraction $A/B$ denotes $AB^{-1}$.

### 3.7 Bogoliubov transformation

The transformation from one Gaussian state to the other is sometimes encoded in a Bogoliubov transformation. This is an indirect way to describe the transformation from a reference vacuum $|0\rangle$ annihilated by $\hat{a}_i$ to the new Gaussian state vector $|J\rangle$ annihilated by $\hat{b}_i$ with

$$\hat{b}_i = \alpha_{ij}\hat{a}_j + \beta_{ij}\hat{a}_j^\dagger, \tag{132}$$

where we sum over the repeated index $j$. We will see that the information contained in $\alpha_{ij}$ and $\beta_{ij}$ is equivalent to the one contained in a group transformation $M \in \mathcal{G}$. In fact, a Bogoliubov transformation is nothing else than a symplectic or orthogonal group transformation expressed in a complex basis $\hat{\xi}^a$.

**Formula 7** (Bogoliubov transformations). *Given a Gaussian reference state vector $|0\rangle$ with covariance matrix $\Gamma_0$ and annihilation operators $\hat{a}_i$, we reach any Gaussian state vector $|J\rangle$ by a Bogoliubov transformation*

$$\hat{b}_i = \alpha_{ij}\hat{a}_j + \beta_{ij}\hat{a}_j^\dagger, \tag{133}$$

*such that $\hat{b}_i|J\rangle = 0$. We compute $\Gamma$ from $\alpha$ and $\beta$ via*

$$M \overset{q,p}{\equiv} \left( \begin{array}{c|c} \operatorname{Re}\alpha + \operatorname{Re}\beta & \operatorname{Im}\beta - \operatorname{Im}\alpha \\ \hline \operatorname{Im}\alpha + \operatorname{Im}\beta & \operatorname{Re}\alpha - \operatorname{Re}\beta \end{array} \right) \overset{a,a^\dagger}{\equiv} \left( \begin{array}{c|c} \alpha & \beta \\ \hline \beta^* & \alpha^* \end{array} \right), \tag{134}$$

*and then evaluating $\Gamma = M\Gamma_0 M^\intercal$. Vice versa, for a given $\Gamma$, there are many choices of $\alpha$ and $\beta$. They can be computed from* (134) *by setting $M = Tu$, where $T^2 = \Delta = \Gamma\Gamma_0^{-1}$ and choosing an arbitrary $u \in \mathrm{U}(N)$, such as $u = \mathbb{1}$.*

### 3.8 Thermal states

Every mixed Gaussian state can be written as a thermal state $\rho = e^{-\beta\hat{H}}/Z$, where $\hat{H}$ is a quadratic Hamiltonian with a unique ground state and $Z = \operatorname{Tr}(e^{-\beta\hat{H}})$. Without loss of generality, we can assume $\beta = 1$ and $Z = 1$ by redefining $\hat{H}$. With this choice, $\hat{H}$ is also known as the *modular Hamiltonian*. A general quadratic Hamiltonian can be written as

$$\hat{H} = \begin{cases} c_0 + q_{ab}\hat{\xi}^a\hat{\xi}^b & \textbf{(bosons)} \\ c_0 + \mathrm{i}\, q_{ab}\hat{\xi}^a\hat{\xi}^b & \textbf{(fermions)} \end{cases}, \tag{135}$$

where $q_{ab}$ is symmetric for bosons and anti-symmetric for fermions. Note that there is no factor of $\frac{1}{2}$ as in (35), which will simplify later conventions. Because $\hat{H}$ has a unique ground state, it follows that there exists a basis of creation and annihilation operators with number operators $\hat{n}_i$, such that

$$\hat{H} = \begin{cases} c_0 + \sum_i \omega_i(\hat{n}_i \pm \frac{1}{2}) & \textbf{(bosons)} \\ c_0 + \sum_i \omega_i(\hat{n}_i \pm \frac{1}{2}) & \textbf{(fermions)} \end{cases}, \tag{136}$$

where $\omega_i > 0$ and $\hat{n}_i$ is the respective bosonic or fermionic number operator. In this specific basis, the density operator $\rho$ decomposes into a tensor product over single modes, from which we can derive the respective standard forms of $J^a{}_b$, $G^{ab}$, $\Omega^{ab}$ and $q_{ab}$ listed in table 3.

Table 3: We list the standard forms of $J$, $G$, $\Omega$, $q$ and $c_0$ for a mixed Gaussian state $\rho = \exp(-c_0 - q_{ab}\hat{\xi}^a\hat{\xi}^b)$.

|  | Bosons | Fermions |
|---|---|---|
| $\rho$ | $\bigotimes\limits_{i=1}^{N_A}\left(\dfrac{e^{-2\hat{n}_i \ln \coth r_i}}{\cosh r_i \sinh r_i}\right)$ | $\bigotimes\limits_{i=1}^{N_A}\left(\cos r_i \sin r_i e^{-2\hat{n}_i \ln \tan r_i}\right)$ |
| $J \overset{q,p}{\equiv}$ | $\bigoplus\limits_{i=1}^{N_A}\begin{pmatrix} 0 & \cosh 2r_i \\ -\cosh 2r_i & 0 \end{pmatrix}$ | $\bigoplus\limits_{i=1}^{N_A}\begin{pmatrix} 0 & \cos 2r_i \\ -\cos 2r_i & 0 \end{pmatrix}$ |
| $J \overset{a,a^\dagger}{\equiv}$ | $\bigoplus\limits_{i=1}^{N_A}\begin{pmatrix} -i\cosh 2r_i & 0 \\ 0 & i\cosh 2r_i \end{pmatrix}$ | $\bigoplus\limits_{i=1}^{N_A}\begin{pmatrix} -i\cos 2r_i & 0 \\ 0 & i\cos 2r_i \end{pmatrix}$ |
| $G \overset{q,p}{\equiv}$ | $\bigoplus\limits_{i=1}^{N_A}\begin{pmatrix} \cosh 2r_i & 0 \\ 0 & \cosh 2r_i \end{pmatrix}$ | $\bigoplus\limits_{i=1}^{N_A}\begin{pmatrix} 1 & 0 \\ 0 & 1 \end{pmatrix}$ |
| $G \overset{a,a^\dagger}{\equiv}$ | $\bigoplus\limits_{i=1}^{N_A}\begin{pmatrix} 0 & \cosh 2r_i \\ \cosh 2r_i & 0 \end{pmatrix}$ | $\bigoplus\limits_{i=1}^{N_A}\begin{pmatrix} 0 & 1 \\ 1 & 0 \end{pmatrix}$ |
| $\Omega \overset{q,p}{\equiv}$ | $\bigoplus\limits_{i=1}^{N_A}\begin{pmatrix} 0 & 1 \\ -1 & 0 \end{pmatrix}$ | $\bigoplus\limits_{i=1}^{N_A}\begin{pmatrix} 0 & \cos 2r_i \\ -\cos 2r_i & 0 \end{pmatrix}$ |
| $\Omega \overset{a,a^\dagger}{\equiv}$ | $\bigoplus\limits_{i=1}^{N_A}\begin{pmatrix} 0 & -i \\ i & 0 \end{pmatrix}$ | $\bigoplus\limits_{i=1}^{N_A}\begin{pmatrix} 0 & -i\cos 2r_i \\ i\cos 2r_i & 0 \end{pmatrix}$ |
| $q \overset{q,p}{\equiv}$ | $\bigoplus\limits_{i=1}^{N_A}\begin{pmatrix} \ln\coth r_i & 0 \\ 0 & \ln\coth r_i \end{pmatrix}$ | $\bigoplus\limits_{i=1}^{N_A}\begin{pmatrix} 0 & \ln\tan r_i \\ -\ln\tan r_i & 0 \end{pmatrix}$ |
| $q \overset{a,a^\dagger}{\equiv}$ | $\bigoplus\limits_{i=1}^{N_A}\begin{pmatrix} 0 & \ln\coth r_i \\ \ln\coth r_i & 0 \end{pmatrix}$ | $\bigoplus\limits_{i=1}^{N_A}\begin{pmatrix} 0 & i\ln\tan r_i \\ -i\ln\tan r_i & 0 \end{pmatrix}$ |
| $c_0$ | $\sum\limits_{i=1}^{N_A}\log(\cosh r_i \sinh r_i)$ | $-\sum\limits_{i=1}^{N_A}\log(\cos r_i \sin r_i)$ |

**Formula 8** (Thermal states). *For a mixed Gaussian state $\rho$ with covariance matrix $\Gamma$, we can always write $\rho = e^{-\hat{H}}$ with $\hat{H}$ from (135) and*

$$q_{ab} = \begin{cases} -i\,\omega_{ac}\,\text{arccoth}\,(iJ)^c{}_b & \textbf{(bosons)} \\ -i\,g_{ac}\,\text{arctanh}\,(iJ)^c{}_b & \textbf{(fermions)} \end{cases}, \tag{137}$$

$$c_0 = \begin{cases} \tfrac{1}{4}\log\det\left(\tfrac{\mathbb{1}+J^2}{4}\right) & \textbf{(bosons)} \\ -\tfrac{1}{4}\log\det\left(\tfrac{\mathbb{1}+J^2}{4}\right) & \textbf{(fermions)} \end{cases}, \tag{138}$$

*where $q_{ab}$ and $c_0$ diverge for $J^2 = -\mathbb{1}$ in such a way that the limit of $\rho$ describes the projector $\rho = |J\rangle\langle J|$. These relations can be easily inverted to compute $J$ and $\Gamma$ in terms of $q_{ab}$.*

## 3.9 Wave functions

Most physicists encounter Gaussian states for the first time when studying the quantum harmonic oscillator. The ground state is a Gaussian state with Gaussian wave function $q \mapsto \psi(q)$,

where $q \in Q$ is a vector in position space $Q$. In this section, we show how every pure bosonic Gaussian state can be represented as Gaussian wave function, either as pure wave function $q \mapsto \psi(q)$ or as mixed wave function $(q, \bar{q}) \mapsto \rho(q, \bar{q})$, and how to convert between wave functions and covariance matrices.

In order to write down a wave function, one needs to make a choice by splitting the classical phase space $V$ into the direct sum $V = Q \oplus P$ with $\dim Q = \dim P = N$, such that the symplectic form vanishes on $Q, P \subset V$. More precisely, we find the block form

$$\Omega^{ab} = \left( \begin{array}{c|c} 0 & \Omega^{\alpha\dot{\beta}} \\ \hline \Omega^{\dot{\alpha}\beta} & 0 \end{array} \right) \text{ and } \omega_{ab} = \left( \begin{array}{c|c} 0 & \omega_{\alpha\dot{\beta}} \\ \hline \omega_{\dot{\alpha}\beta} & 0 \end{array} \right), \tag{139}$$

where we have $q \in Q$ and $p \in P$. The phase space decomposition $V = Q \oplus P$ induces a dual decomposition $V^* = Q^* \oplus P^*$. The off-diagonal blocks in $\Omega$ and $\omega$ induce isomorphism $Q \simeq P^*$ and $Q^* \simeq P$.[14]

### 3.9.1 Pure states

We write the most general pure Gaussian state as

$$\psi(q) = \left( \det \tfrac{A}{\pi} \right)^{1/4} \exp\left( -\frac{1}{2} q^\alpha (A_{\alpha\beta} + iB_{\alpha\beta}) q^\beta \right). \tag{140}$$

Note that the determinant of the bilinear form $A$ implies that the wave function is not a scalar function, but rather a scalar density of weight $1/2$, *i.e.*, if we change our coordinates $q \to \tilde{q} = Cq$ for some $C \in \mathbb{R}$, we have $\psi(q) \to \tilde{\psi}(\tilde{q}) = C^{N/2} \psi(C^{-1}\tilde{q})$, such that $\int |\psi(q)|^2 d^N q = \int |\tilde{\psi}(\tilde{q})|^2 d^N \tilde{q}$. This ensures that the square modulus of the wave function can be integrated over $Q$ to give probabilities. We decompose the bosonic covariance matrix $G^{ab}$ and the symplectic form $\Omega^{ab}$ based on our decomposition of the phase space $V = Q \oplus P$, such that

$$G^{ab} = \langle \psi | (\hat{\xi}^a \hat{\xi}^b + \hat{\xi}^b \hat{\xi}^a) | \psi \rangle = \left( \begin{array}{c|c} G^{\alpha\beta} & G^{\alpha\dot{\beta}} \\ \hline G^{\dot{\alpha}\beta} & G^{\dot{\alpha}\dot{\beta}} \end{array} \right). \tag{141}$$

Note that the only requirement for the respective decomposition $V = Q \oplus P$ is that the restrictions $\Omega^{\alpha\beta}$ and $\Omega^{\dot{\alpha}\dot{\beta}}$ vanish.

**Formula 9** (Pure state wave function). *Given a bosonic Gaussian state vector $|G\rangle$ and a phase space decomposition $V = Q \oplus P$, we can convert between the covariance matrix decomposed in the blocks (141) and the wave function representation from (140) containing the bilinear forms $A_{\alpha\beta}$ and $B_{\alpha\beta}$ using*

$$\begin{aligned} G^{\alpha\beta} &= (A^{-1})^{\alpha\beta}, \\ G^{\dot{\alpha}\dot{\beta}} &= -\Omega^{\dot{\alpha}\gamma}(A + BA^{-1}B)_{\gamma\delta}\Omega^{\delta\dot{\beta}}, \\ G^{\alpha\dot{\beta}} &= -(A^{-1})^{\alpha\gamma}B_{\gamma\delta}\Omega^{\delta\dot{\beta}}, \\ G^{\dot{\alpha}\beta} &= \Omega^{\dot{\alpha}\gamma}B_{\gamma\delta}(A^{-1})^{\gamma\beta}. \end{aligned} \tag{142}$$

*Vice versa, we can solve these equations for $A$ and $B$ in terms of $G$ to find*

$$A_{\alpha\beta} = G^{-1}_{\alpha\beta} \quad \text{and} \quad B_{\alpha\beta} = G^{-1}_{\alpha\gamma}G^{\gamma\dot{\delta}}\omega_{\dot{\delta}\beta}. \tag{143}$$

Note that $G^{-1}_{\alpha\beta}$ is the inverse of the $N \times N$ block $G^{\alpha\beta}$ satisfying $G^{\alpha\beta}G^{-1}_{\beta\gamma} = \delta^\alpha_{\ \gamma}$ over $Q \subset V$ which should not be confused with the full $2N \times 2N$ inverse $g_{ab}$ of $G^{ab}$ with $G^{ab}g_{bc} = \delta^a_{\ c}$ over $V$.

---

[14]This isomorphism means that we can identify the position vector $q^\alpha$ with dual momentum $q^\alpha \omega_{\alpha\dot{\beta}}$ and similar.

### 3.9.2 Mixed states

Similarly, we can also write out the most general mixed Gaussian state in the position representation as

$$\rho(q,\bar{q}) = Z \exp\left(-\frac{1}{2}\begin{pmatrix}q\\\bar{q}\end{pmatrix}^{\mathsf{T}}\begin{pmatrix}A+\mathrm{i}B & C+\mathrm{i}D\\C-\mathrm{i}D & A-\mathrm{i}B\end{pmatrix}\begin{pmatrix}q\\\bar{q}\end{pmatrix}\right),\tag{144}$$

where $q = (q_1,\ldots,q_N)$, $\bar{q} = (\bar{q}_1,\ldots,\bar{q}_N)$ and the normalization is given by

$$Z = \left(\det\frac{A+C}{\pi}\right)^{1/2}.\tag{145}$$

Again, the wave function representation of the mixed state $\rho$ is density of weight $1/2$. As before, we would like to relate the bilinear forms $A$, $B$, $C$ and $D$ in terms of the covariance matrix

$$G^{ab} = \mathrm{Tr}[\rho(\hat{\xi}^a\hat{\xi}^b + \hat{\xi}^b\hat{\xi}^a)] = \left(\begin{array}{c|c}G^{\alpha\beta} & G^{\alpha\dot{\beta}}\\\hline G^{\dot{\alpha}\beta} & G^{\dot{\alpha}\dot{\beta}}\end{array}\right).\tag{146}$$

We find the following relations.

**Formula 10** (Mixed state wave function)**.** *The different blocks of the covariance matrix $G^{ab}$ are related to the matrices $A, B, C, D$ via*

$$\begin{aligned}G^{\alpha\beta} &= \left((A+C)^{-1}\right)^{\alpha\beta},\\G^{\dot{\alpha}\dot{\beta}} &= -\Omega^{\dot{\alpha}\gamma}\left(A-C+(B+D)(A+C)^{-1}(B-D)\right)_{\gamma\delta}\Omega^{\delta\dot{\beta}},\\G^{\alpha\dot{\beta}} &= -\left((A+C)^{-1}\right)^{\alpha\gamma}(B-D)_{\gamma\delta}\Omega^{\delta\dot{\beta}},\\G^{\dot{\alpha}\beta} &= \Omega^{\dot{\alpha}\gamma}(B+D)_{\gamma\delta}\left((A+C)^{-1}\right)^{\delta\beta},\end{aligned}\tag{147}$$

*which can be inverted to give*

$$\begin{aligned}A_{\alpha\beta} &= \frac{1}{2}\left(G^{-1}_{\alpha\beta} - \omega_{\alpha\dot{\gamma}}\left(G^{\dot{\gamma}\dot{\delta}} - G^{\dot{\gamma}\epsilon}G^{-1}_{\epsilon\zeta}G^{\zeta\dot{\delta}}\right)\omega_{\dot{\delta}\beta}\right),\\B_{\alpha\beta} &= -\frac{1}{2}\left(G^{-1}_{\alpha\gamma}G^{\gamma\dot{\delta}}\omega_{\dot{\delta}\beta} - \omega_{\alpha\dot{\gamma}}G^{\dot{\gamma}\delta}G^{-1}_{\delta\beta}\right),\\C_{\alpha\beta} &= \frac{1}{2}\left(G^{-1}_{\alpha\beta} + \omega_{\alpha\dot{\gamma}}\left(G^{\dot{\gamma}\dot{\delta}} - G^{\dot{\gamma}\epsilon}G^{-1}_{\epsilon\zeta}G^{\zeta\dot{\delta}}\right)\omega_{\dot{\delta}\beta}\right),\\D_{\alpha\beta} &= -\frac{1}{2}\left(G^{-1}_{\alpha\gamma}G^{\gamma\dot{\delta}}\omega_{\dot{\delta}\beta} + \omega_{\alpha\dot{\gamma}}G^{\dot{\gamma}\delta}G^{-1}_{\delta\beta}\right).\end{aligned}\tag{148}$$

*In our standard basis, we will have $\Omega^{\alpha\dot{\beta}} \stackrel{q,p}{\equiv} \mathbb{1}$, $\Omega^{\dot{\alpha}\beta} \stackrel{q,p}{\equiv} -\mathbb{1}$, $\omega_{\alpha\dot{\beta}} \stackrel{q,p}{\equiv} -\mathbb{1}$ and $\omega_{\alpha\dot{\beta}} \stackrel{q,p}{\equiv} \mathbb{1}$, which simplifies above expressions further*

Note here the signs. Notice also that formula (147) reduces to formula (142) if the state is pure, *i.e.*, $C = D = 0$.

## 4 Optimization algorithm

Having reviewed the parametrization of Gaussian states using complex structures and having related this formalism to the other most commonly used parametrizations in the literature, we now return to our initial goal of efficient local optimization over the class of Gaussian states. Finding the minimal value (or maximum) of a function $f$ on some large manifold $\mathcal{M}$

is in general a hard problem and the primary goal in the field of mathematical optimization. One distinguishes between global and local optimization, *i.e.*, if one is able to find the global minimum or if one may get stuck in a local one. In this section, we present a schematic overview of our approach to efficient *local* optimization over the class of pure Gaussian states, based on the geometric considerations of section 2.4. We also allude to the flexibility of our optimization algorithm in finding *global* minima and avoiding the pitfalls of poor convergence. We use a rudimentary gradient descent implementation [30], but exploit the natural geometry of Gaussian states and exploit the Lie group structure of $\mathcal{G}$.

## 4.1 Gradient descent on matrix manifolds

Given a function $f : \mathcal{M} \to \mathbb{R}$ on some manifold $\mathcal{M}$, we can find its minimum from some starting point using gradient descent. At any point $x \in \mathcal{M}$ in the manifold, the range of possible directions of motion can be expanded in a basis of the vectors in the tangent space to $\mathcal{M}$ at $x$, denoted $\mathcal{T}_x \mathcal{M}$. Gradient descent is one of the most basic methods of finding a minimum by moving iteratively in directions which locally decrease the function. This means picking out suitable vectors in $\mathcal{T}_x \mathcal{M}$ directed along those directions which minimise the function value. Specifically, these are the components of the gradient descent vector field on the manifold, which is given by

$$\mathcal{F}^\mu = -G^{\mu\nu} \frac{\partial f}{\partial x^\nu}, \tag{149}$$

*i.e.*, it associates with each point $x \in \mathcal{M}$ the directional derivative of $f$. The inverse metric $G^{\mu\nu}$ is included in this definition to remove the sensitivity of the gradient to the choice of local basis $x^\mu$.

The analytical solution to the gradient descent problem is the integral curve associated with the vector field (149). In a numerical realization of gradient descent, we approximate this continuous curve by sufficiently small discrete incremental steps. This notion of moving a certain distance along one of the tangent vectors in $\mathcal{T}_x \mathcal{M}$ while remaining on $\mathcal{M}$, which is trivial when $\mathcal{M}$ is flat, is realized for the general non-flat case by a so-called retraction map. This is a map $R : \mathcal{T} \mathcal{M} \to \mathcal{M}$, with restrictions to the domains $\mathcal{T}_x \mathcal{M}$ given by the maps

$$R_x : \quad T_x \mathcal{M} \to \mathcal{M}, \tag{150}$$

which is required to satisfy

$$R_x(0) = x \quad \text{and} \quad dR_x(0) = \text{id}_{\mathcal{T}_x \mathcal{M}}, \tag{151}$$

where $\text{id}_{\mathcal{T}_x \mathcal{M}}$ denotes the identity mapping on the tangent space. It is important to note that the retraction map is not unique and that the most convenient choice of a retraction map will be that which minimizes the computational effort while remaining a sufficiently accurate approximation to the continuous integral curve.[15]

If we optimize over a Lie group $\mathcal{G}$, a natural choice for the retraction map is the exponential map, which, for a matrix Lie group, is simply given by the matrix exponential,

$$e^K = \sum_{n=0}^{\infty} \frac{K^n}{n!}, \tag{153}$$

---

[15]In the flat case $\mathcal{M} = \mathbb{R}^n$, the natural choice of the retraction map is

$$R_x(u) = x + u, \tag{152}$$

since in this case the manifold and its tangent space are globally isomorphic. This is the familiar notion of moving forward by some step $u$ from a point $x$.

since for Lie groups tangent vectors and Lie algebra elements $K$ are equivalent. When optimizing over a Lie group $\mathcal{G}$, we divide the integral curve into continuous segments, curves $\gamma(t) = M e^{tK}, t \in [0,1]$ connecting subsequent points $M$ and $M' = M e^{K}$ in the group manifold. Here, $K$ denotes the tangent vector to the group manifold at $M$ corresponding with motion to $M'$ and this motion is realized by the exponential (retraction) map.

In practice, computing the full power series in (153) is expensive and we would prefer a more computationally viable approximation to the exponential. In principle, we can consider small Lie algebra elements $K$ under the norm $||K||^2 = \mathrm{Tr}(KK^{\intercal})$ and then perform a reasonable truncation of the power series. The issue here is that a power series approximation of the exponential will not lie in the desired Lie group *i.e.*, it cannot serve its purpose as a retraction map. There is, however, another approximation to the matrix exponential, which does fulfill this criterion: For algebra elements $K$, we have [31]

$$e^{\epsilon K} \sim \frac{(\mathbb{1} + \frac{\epsilon}{2} K)}{(\mathbb{1} - \frac{\epsilon}{2} K)} \quad \text{as} \quad \epsilon \to 0, \tag{154}$$

which will always map into the associated Lie group. It is important to note that evidently, if one of the eigenvalues of $\frac{\epsilon}{2} K$ is 1, then the expression in (154) cannot be inverted. We avoid this by choosing $\epsilon$ sufficiently small. Evaluating the RHS of (154) is much more efficient than computing the exponential of the LHS, as the computation of the inverse $(\mathbb{1} - \frac{\epsilon}{2} K)^{-1}$ and its multiplication with $(\mathbb{1} + \frac{\epsilon}{2} K)$ can be performed with a single method with the complexity of matrix multiplication.[16]

## 4.2 Optimization on the Gaussian state manifold

As discussed in Section 2, the Gaussian state manifold is equipped with a Riemannian metric $g_{\mu\nu}$ with inverse $G^{\mu\nu}$ and therefore the vector field (149) can be naturally defined for both the bosonic and fermionic state manifolds (1) and (2). In general, the inverse metric $G^{\mu\nu}$ needs to be re-evaluated at every point of the manifold, but as suggested previously, by moving into one of the standard basis choices in (21), the matrix representation of the inverse metric is constant. This is the first of the properties of Gaussian states which allows for a particularly computationally efficient implementation of gradient descent optimization over this class of states. Section 2.5 outlines in some detail the parametrization of the Gaussian state manifold in terms of transformations $M$ of some reference state complex structure $J_0$. Based on this parametrization, when optimizing over the manifold $\mathcal{M}$ of all pure Gaussian states we may equally say that we are optimizing over the matrix groups (33) quotiened by the redundancies associated $\mathrm{U}(N)$, which form manifolds of dimensions

$$\dim \mathcal{M}_b = N(2N+1) - N^2 = N(N+1), \tag{155}$$

$$\dim \mathcal{M}_f = N(2N-1) - N^2 = N(N-1). \tag{156}$$

This means that in practice, the vector field $\mathcal{F}^{\mu}$ is computed not with respect to the local basis of a tangent space to $\mathcal{M}$ at a state $J_M$, but rather with respect to an orthonormal basis $\Xi_{\mu}$ of the Lie algebra $\mathfrak{g}$ or, more precisely, of the subspace $\mathfrak{h}'_{\perp} \subset \mathfrak{g}$ introduced in (72) which generates non-zero variations in the complex structure. This idea is what leads to the expression for the variation of a state in terms of a Lie algebra element in (69).

We can relate the gradient vector $\mathcal{F}^{\mu}$ to the associated Lie algebra element $K$ as

$$K = \mathcal{F}^{\mu} \Xi_{\mu} \tag{157}$$

---

[16]Exact matrix inversion $X = A^{-1}$ can be performed by solving the linear system $AX = \mathbb{1}$, which is as fast as matrix multiplication. Computing $AB^{-1}$ is just as efficient, as we now merely solve $XB = A$.

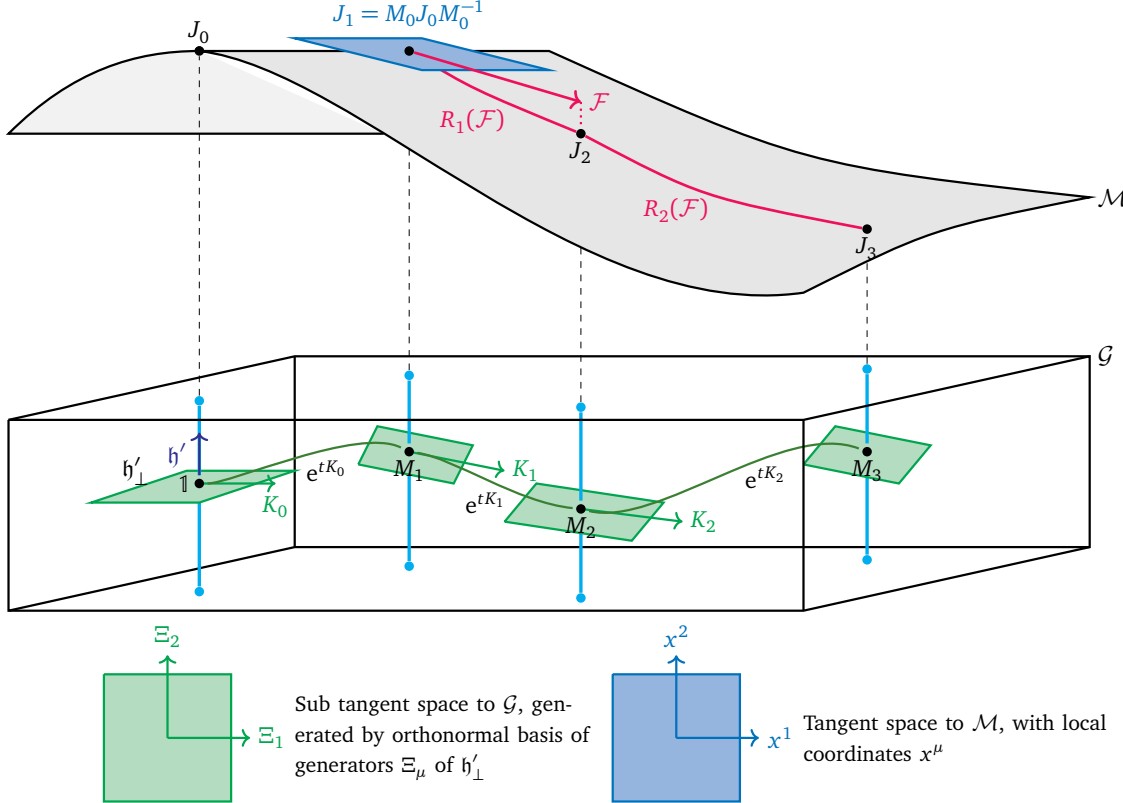

Figure 2: *Gradient descent on the Gaussian state manifold.* Visualization of the gradient descent geometry for a two-dimensional state manifold. The state manifold $\mathcal{M}$ is parametrized in terms of complex structures $J_n$ which in turn are parametrized by the transformations $M_n$ in the Lie group $\mathcal{G}$ as given in (44), with respect to some reference state $J_0$. The blue surface indicates a tangent space to $\mathcal{M}$ at state $J_1$. The vector field $\mathcal{F}$ in this tangent space is also included, as well as the retraction map $R(\mathcal{F})$ which is shown as a projection of the vector field onto $\mathcal{M}$ to visualize the notion of moving in the direction of a tangent vector but in the manifold. The associated group $\mathcal{G}$ is also shown and crucially it should be noted that the tangent spaces at each point in the group are aligned with the manifold tangent spaces (to highlight the isomorphism between the two) but do not reproduce the smooth manifold $\mathcal{M}$, since the equivalent of the curve traced out on $\mathcal{M}$ by successive retraction maps simply connects matrix elements which lie along the (light blue) fibers which represent the stabilizers $\mathfrak{h}'$ as indicated illustratively at the identity. In $\mathcal{G}$, the gradient vector field at points $M_n$ is denoted by $K_n$ as defined in (157). The lines connecting points $M_n$ and $M_{n+1}$ in the group are defined by $e^{sK_n}$ with $s \in [0,1]$ as written in (160).

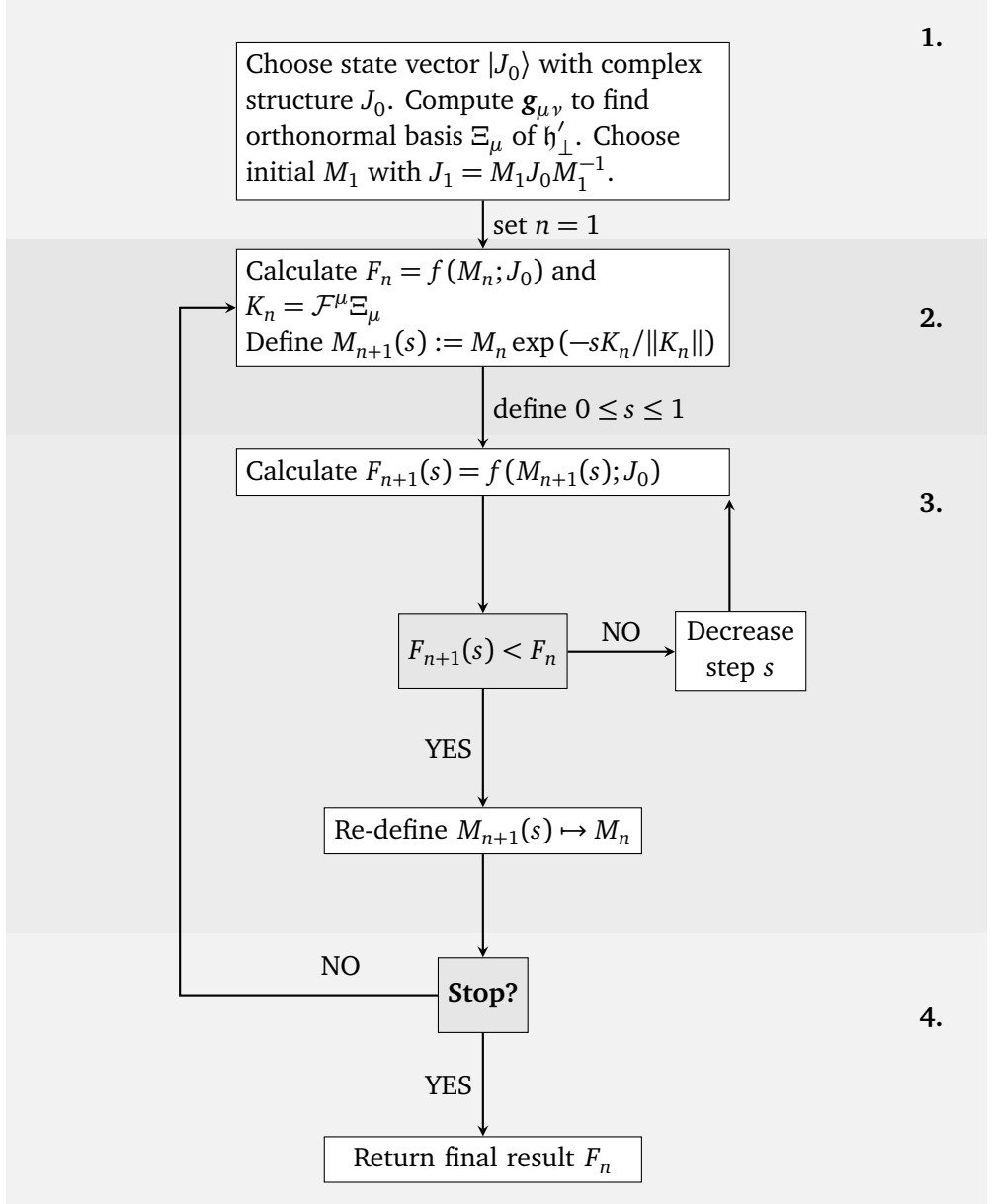

Figure 3: *Graphical representation of the algorithm.* We show a step-by-step explanation of the optimization algorithm described in the main text. Gray shading indicates a decision box and the color-coded sections correspond with those distinguished in the main text. The "Stop?" decision box indicates the implementation of a stop criterion.

using local basis $\Xi_\mu$ of $\mathfrak{h}'_\perp$. A second key point arises from the left-invariance of the Riemannian metric on the Gaussian state manifold: Since this leads to the preservation of the orthonormality of any choice of $\Xi_\mu$ under transformations in the group, we do not need to compute a new orthonormal basis at different points in the manifold. Instead, we can choose $\Xi_\mu$ to be the generators of the Lie group, which form the natural orthonormal basis of the tangent space to the identity. This leads to significant computational speedups, particularly when optimizing over high-dimensional manifolds. The setup for gradient descent on the Gaussian state manifold is visualized in fig. 2.

## 4.3 Performing gradient descent

Since we work in a parametrization of the state manifold solely in terms of the transformations of a reference state, a computational implementation of the algorithm should be able to evaluate the target function $f$ for any state vector $|J_M\rangle$ with only $J_0$ and $M$ as arguments. Here, we write this as $(M, J_0) \mapsto f(M, J_0)$. To define local derivatives, we introduce a local coordinate system $x^\mu$ around a point $M \in \mathcal{G}$, such that $f(x) = f(Me^{x^\mu \Xi_\mu}, J_0)$ leading to

$$\frac{\partial f}{\partial x^\mu} = \frac{\partial}{\partial t}\bigg|_{t=0} f\left(Me^{x^\mu \Xi_\mu}, J_0\right), \tag{158}$$

which allows us to define the vector field $\mathcal{F}^\mu$ according to (149) with respect to the basis $\Xi_\mu$ of $\mathfrak{h}'_\perp$. We re-emphasize here that (158) lets us naturally express the gradient in terms of the variation of the group element only, *i.e.*, at no point are we required to move from the group $\mathcal{G}$ to the state manifold $\mathcal{M}$. This approach is shown for various examples in Section 5.

We now provide a step-by step explanation of the realization of an iterative gradient descent minimization algorithm based on the considerations above. The steps are summarized graphically in fig. 3, which complements fig. 2.

**1. Initialization.** Our algorithm is initialized on a Gaussian state vector $|J_0\rangle$ with covariance matrix $\Gamma_0$ and complex structure $J_0$, such that the action of the subgroup $\mathcal{G}' \subset \mathcal{G}$ generates the state manifold under consideration. We then construct an orthonormal basis $\Xi_\mu$ for $\mathfrak{h}'_\perp$, which are both defined with respect to $\Gamma_0$. For this, we compute the metric $\mathbf{G}^{\mu\nu}$ explicitly. We evaluate $\mathbf{g}_{\mu\nu}$ in an arbitrary basis $\Xi_\mu$, so we can orthogonalize it, such that both $\mathbf{g}_{\mu\nu}$ and $\mathbf{G}^{\mu\nu}$ are equal to the identity. The metric $\mathbf{g}_{\mu\nu}$ is efficiently computed as

$$\mathbf{g}_{\mu\nu} = \frac{1}{4}(\Xi_\mu \Xi_\nu + \Xi_\mu \Gamma_0 \Xi_\nu^{\mathsf{T}} \Gamma_0), \tag{159}$$

where we use (74) with respect to the reference state vector $|J_0\rangle = |0\rangle$. By construction, we identify the tangent spaces at all group elements $M_n$ with the ones at $M_0 = \mathbb{1}$. While $|J_0\rangle$ is usually chosen in some standard form, we can still initialize the algorithm on some $M_1$ based on the problem at hand.

**2. Gradient computation.** We now perform successive steps in the group as

$$M_{n+1} = M_n \exp\left(-s\frac{K_n}{||K_n||}\right), \quad 0 \le s \le 1, \tag{160}$$

where $K_n = \mathcal{F}^\mu \Xi_\mu$ with $\mathcal{F}^\mu$ calculated at the point $M_n$ and $s$ chosen such that $f(M_{n+1}; J_0) < f(M_n; J_0)$.

**3. Sub-routine to determine step-size.** To choose an appropriate step-size $s$, we use a sub-routine at each iteration which should be chosen so as to balance the efficiency gained by needing fewer steps to reach the minimum and the extra computational effort of executing the subroutine. In the examples discussed in later section, we found that the very rudimentary approach of iteratively halving the step size was sufficient to ensure good convergence. However, a simple line search methods like a quasi-Newton routine can also be used.

**4. Stop condition.** This iterative motion is repeated until some pre-determined stopping criterion (*e.g.*, a tolerance on the gradient norm or the difference between subsequent function values) is reached.

## 4.4 Practical considerations

While the focus of this work is not on elaborate numerical methods for implementing the algorithm described above, we mention here for the sake of completeness some additional

considerations regarding the practical implementation the algorithm. These have been added in our implementation of the algorithm to varying degrees to enhance its efficiency.

**Constrained optimization.** Our approach lends itself intuitively to constrained optimization, since we can choose to restrict our optimization to a smaller range of states, being some subspace of $\mathcal{G}$, by truncating the Lie algebra basis. We show an example of this is in the section on Complexity of Purification, where those algebra elements which do not generate non-zero variations of the complex structure can be explicitly cut out of the basis.

**Extension to global optimization.** There is ultimately no fail-safe way to locate global minima using gradient descent. However, we can increase the probability of convergence to the global minimum by performing gradient descent from a number of sufficiently far separated starting points in the manifold and choosing the lowest of the local minima from a large enough sample size. Here "sufficiently far" refers to sprinkling the manifold evenly in the region of interest. For fermions, this can be achieved by randomly generating matrices in $SO(2N, \mathbb{R})$ with respect to the Haar measure. This does not work for bosons, as the group $Sp(2N, \mathbb{R})$ is non-compact, but one could try a Gaussian measure instead that is concentrated in the region where the function $f$ is expected to have minima.

**Identifying suitable starting points.** In choosing different starting points, it may be possible, given some analytical intuition about the physical system at hand, to identify starting points in the manifold from which the risk of landing in a local minimum is particularly low. Additionally, there may be some starting points in the manifold from which gradient descent will converge the fastest (*i.e.*, from which it will take a much lower number of iterations to reach the minimum).

**Parallel optimization.** Where the most suitable starting points cannot be found analytically, we must resort to numerical methods: If, rather than minimizing successively from different starting points, we choose to perform the optimizations simultaneously, we can discriminate between trajectories which promise to converge more or less quickly to the minimum. In our algorithm, we implement this feature, and after each set of 5 iterations, only the 10% of trajectories with the lowest function value and the highest gradient, respectively, are pursued further. While this does generally speaking greatly reduce the total number of iterations required, there is of course a trade-off between this improvement and the computational effort of an initially large number of trajectories.

## 4.5  The `GaussianOptimization.m` **package**

To complement the theory outlined in this paper, we supply the public `GaussianOptimization.m` Mathematica package with a simple implementation of the optimization algorithm discussed in the previous sections. The package revolves around the function `GOOptimize`, which performs the gradient descent optimization from some initial complex structure and transformation. The input arguments to this function are divided into three categories:

**Problem-specific.** These are the arguments related to the specific optimization problem at hand, the scalar function and its derivative with respect to some Lie algebra element, expressed in terms of the initial complex structure and an arbitrary transformation, in the spirit of Table 4.

**System-specific.** These relate to the geometry of the optimization problem. Fundamentally, this includes the symplectic or orthogonal basis (which can be generated using the built-in functions `GOSpBasis` and `GOOBasis`), but also the corresponding metric (generated by `GOMetricSp` and `GOMetricO`). It also includes the initial complex structure and the (list of) initial transformations.

Table 4: *Function and gradient parametrization.* We list the quantities discussed in this section as scalar functions of the complex structure $J_M = M J_0 M^{-1}$ at $M$ in the state manifold. We also list the associated gradient functions, parametrized by the infinitesimal changes in the complex structure $\delta J_M(K)$, as given in (69). The expressions for the CoP are defined in terms of $\Delta = -J_M J_{\mathrm{T}}$, introduced as the *relative covariance matrix* in ref. [20], and $\delta\Delta = \delta J_M(K) J_{\mathrm{R}}^{-1}$.

|  | (A) Approximate ground states | (B) Entanglement of purification | (C) Complexity of purification |
|---|---|---|---|
| Bosonic $f$ | $E = \langle \hat{H} \rangle$ | $S_{AA'} = \frac{1}{2}\operatorname{Tr} D \log D^2$ | $C = \sqrt{\frac{1}{8}\operatorname{Tr}\log^2(\Delta)}$ |
| Bosonic $df$ | $dE = \frac{dE}{d\Gamma^{ab}}(K\Gamma + \Gamma K^{\intercal})^{ab}$ | $dS_{AA'} = \frac{1}{2}\operatorname{Tr} dD \log D^2$ | $dC = 2\operatorname{Tr}\log(\Delta)\Delta^{-1}\delta\Delta$ |
| Fermionic $f$ | $E = \langle \hat{H} \rangle$ | $S_{AA'} = -\frac{1}{2}\operatorname{Tr} D \log D$ | $C = \sqrt{\frac{i}{8}\operatorname{Tr}\log^2(\Delta)}$ |
| Fermionic $df$ | $dE = \frac{dE}{d\Gamma^{ab}}(K\Gamma + \Gamma K^{\intercal})^{ab}$ | $dS_{AA'} = -\operatorname{Tr} dD \log D$ | $dC = -2\operatorname{Tr}\log(\Delta)\Delta^{-1}\delta\Delta$ |

**Procedure-specific.** These are the parameters related to the numerical implementation of the algorithm, including stopping criteria based on step limits and tolerances on the function value and gradient.

The `GaussianOptimization.m` package is designed to be user-friendly and all functions come with comprehensive documentation. It is accompanied by an example notebook which includes a systematically organized overview over the functions included in the package, as well as implementations of the applications discussed in the next section. The functions and function gradients for these applications are also implemented as part of the package.

# 5 Applications

In this section, we show how our optimization algorithm may be used in several relevant physical contexts: approximating the ground state of Hamiltonians and computing the entanglement and complexity of purification for fermionic and bosonic systems. We indicate how to parametrize the function to be extremized in terms of the complex structure and how to compute the associated local derivatives (158). As discussed in the previous section, this is essential to unlocking the full computational efficiency of the algorithm. We also provide suggestions regarding convenient starting points and parametrizations. In the examples of this section, the gradient of the function $f$ could be obtained analytically in terms of the complex structure $J$ using the chain rule and properties of matrix calculus. However, this may not be possible in general, *e.g.*, if a function $f$ is the result of some numerical algorithm, it may not be possible to compute its derivative analytically. In this case, automatic differentiation (AD) procedures [32], which provide a numerical algorithm to compute the gradient without the need of an analytical derivative and also without the drawbacks of a purely numerical derivative, present a computationally feasible alternative.

## 5.1 Approximate ground states

The energy function $E = \langle \hat{H} \rangle$ is one of the most prominent functions on families of pure quantum states that should be minimized. This is particularly relevant in the context of finding variational ground states, *i.e.*, finding states within a given variational family of ansatz ground states that approximate the true ground state most accurately with respect to some merit function, which typically is the energy expectation value $E = \langle \hat{H} \rangle$.

Pure Gaussian states and certain submanifolds are known to be very suitable variational families to approximate ground states of bosonic and fermionic Hamiltonians with local inter-

actions. The approximation typically improves with the dimension of the system, *i.e.*, Gaussian states often only capture qualitative features in one spatial dimension, but improve when moving to two and and three dimensions, as mean field descriptions become more accurate. Gaussian states are also heavily used as trial states in mathematical physics to find upper bounds to the energy of quantum gases, *e.g.*, when studying the dilute limit of Bose gases [5]. There exists a range of different tools to find *the best Gaussian state*, *i.e.*, the Gaussian state with the lowest energy expectation value $E$. A prominent example is the Hartree-Fock method, which is typically applied to fermions, but can also be used to approximate bosonic ground states. Another established method is imaginary time evolution, where the geometric flow of $e^{-\tau \hat{H}}$ is approximated on the given manifold.

From a purely numerical perspective, many optimization methods are suitable to find the minimum of a function on conveniently parametrized family. However, in the context of Gaussian states many standard parametrization (using squeezing parameters or quadratic Hamiltonians acting on a reference state) tend to converge unreliably or get stuck in local minima. Taking the natural Riemannian geometry of Gaussian states (induced by the Fubini-Study metric) into account can significantly improve the convergence of such numerical methods. In fact, one can show that gradient descent with respect to this natural geometry coincides with projected imaginary time evolution [15], which is known to have favorable convergence properties. Both, the group-theoretic parametrization and the resulting straight-forward gradient descent algorithm are therefore perfectly suitable to find approximate ground states within the Gaussian state families.

Finding the minimum of energy function $E = \langle \hat{H} \rangle$ requires us to evaluate $E$ and its derivative $dE$ efficiently. For this, we assume that $\hat{H}$ can be written as finite series

$$\hat{H} = h_0 + (h_1)_a \hat{\xi}^a + \cdots + (t_n)_{a_1 \ldots a_n} \hat{\xi}^{a_1} \ldots \hat{\xi}^{a_n}, \tag{161}$$

whose expectation can be evaluated using Wick's theorem. For a Gaussian state $|J\rangle = |J, 0\rangle$ with $n$-point correlation function $C_n^{a_1 \cdots a_n} = \langle \hat{\xi}^{a_1} \ldots \hat{\xi}^{a_n} \rangle$, Wick's theorem states the following.

(a) Odd correlation functions vanish, *i.e.*, $C_{2n+1} = 0$.

(b) Even correlation functions are given by the sum over all two-contractions

$$C_{2n}^{a_1 \cdots a_{2n}} = \sum_{\sigma} \frac{|\sigma|}{n!} C_2^{a_{\sigma(1)} a_{\sigma(2)}} \cdots C_2^{a_{\sigma(2n-1)} a_{\sigma(2n)}}, \tag{162}$$

where $C_2^{ab} = \frac{1}{2}(G^{ab} + i \Omega^{ab})$ has been introduced in (19) and the permutations $\sigma$ satisfy $\sigma(2i-1) < \sigma(2i)$ and $|\sigma| = 1$ for bosons and $|\sigma| = \text{sgn}(\sigma)$ for fermions.

We can always use canonical commutation or anti-commutation relations to ensure that $(t_i)_{a_1 \ldots a_i}$ is totally symmetric (bosons) or anti-symmetric (fermions), in which case Wick's theorem only leads to all contractions with $\frac{1}{2} \Gamma^{ab}$, *i.e.*, either way, we find $E = E(\Gamma)$ as polynomial in the entries of $\Gamma$. A Lie algebra element $K$ perturbs $\Gamma$ at linear order (tangent vector) as $\delta \Gamma = K\Gamma + \Gamma K^{\mathsf{T}}$, such that

$$dE = \frac{\partial E}{\partial \Gamma^{ab}} \delta \Gamma^{ab} = \frac{\partial E}{\partial \Gamma^{ab}} (K\Gamma + \Gamma K^{\mathsf{T}})^{ab}. \tag{163}$$

This allows us a straight-forward implementation of gradient descent on the manifold of all pure Gaussian states (or appropriate submanifolds) based on the algorithm discussed in section 4.

As an interesting observation, let us mention that the described approach can also be used to approximate real time evolution on the manifold of pure Gaussian states. Due to the fact that

the manifold of pure Gaussian states is a Kähler manifold the commonly used variational principles (Lagrangian, McLachlan, Dirac-Frankel) agree [15] and can be implemented as Hamiltonian equations of motion. Our gradient descent algorithm implements the vector field

$$\mathcal{F}^{\mu} = -G^{\mu\nu}\frac{\partial E}{\partial x^{\nu}}, \tag{164}$$

which must be replaced by the Hamiltonian time evolution

$$\mathcal{X}^{\mu} = -\Omega^{\mu\nu}\frac{\partial E}{\partial x^{\nu}}, \tag{165}$$

*i.e.*, we only need to adjust our algorithm in step **2. Gradient computation**, where we replace $K_n = \mathcal{F}^{\mu}\Xi_{\mu}$ by $K_n = \mathcal{X}^{\mu}\Xi_{\mu}$. Here, $\Omega^{\mu\nu}$ is the inverse of $\omega_{\mu\nu}$ computed from (75). Just as in the case of $G^{\mu\nu}$ our group-theoretic parametrization (left invariance) of the Gaussian state manifold ensures that we only need to evaluate $\Omega^{\mu\nu}$ once and can use the same matrix for subsequent steps in the algorithm. Note, however, that there are important differences between imaginary time evolution (gradient descent) and real time evolution. For imaginary time evolution, the step size is only used to ensure that the energy decreases, while for real time evolution we need to keep track of it to know the current time parameter. Moreover, for imaginary time evolution, we just need to make sure that the energy function decreases with each step, while other errors due to the finite step size are not a problem. For real time evolution, we will always make small errors due to the finite step size and can only try to decrease it by enforcing relevant conservation laws. In particular, the energy function should stay exactly constant, so we can try to use the step size to keep the accumulated error under control.

We can also use (163) in combination with (164) and (165) to derive the real and imaginary evolution equations of the covariance matrix Γ, namely [15, 33]

$$\begin{aligned}
\frac{d}{dt}\Gamma &= -4(G\frac{\partial E}{\partial\Gamma}G + \Omega\frac{\partial E}{\partial\Gamma}\Omega), \quad \textbf{(real)} \\
\frac{d}{d\tau}\Gamma &= -4(G\frac{\partial E}{\partial\Gamma}G + \Omega\frac{\partial E}{\partial\Gamma}\Omega). \quad \textbf{(imaginary)}
\end{aligned} \tag{166}$$

For bosonic systems, it is natural to also allow for a non-zero displacement vector $z^a = \langle\hat{\xi}^a\rangle$, which can be seamlessly integrated in the presented formalism, as discussed in refs. [15, 33].

In summary, our group-theoretic parametrization of Gaussian states and the resulting optimization algorithm are suitable to find approximate ground states and to perform projected real time evolution on the manifold of pure Gaussian state. In practical applications, we can typically reduce the dimension of the manifold by implementing certain symmetries, *e.g.*, translational symmetry, from scratch by reducing the number of Lie algebra generators $\Xi_{\mu}$ accordingly and choosing an initial state respecting the chosen symmetry.

## 5.2 Gaussian entanglement of purification (EoP)

The *entanglement of purification* (EoP), first introduced in ref. [34], quantifies the degree of entanglement between subsystems in a composite quantum system. As such, it serves as a valuable correlation measure and has recently become of interest in quantum many body systems [35]. Suppose we are given a mixed state in some Hilbert space $\mathcal{H} = \mathcal{H}_A \otimes \mathcal{H}_B$ which can be described by a density operator $\rho_{AB}$. We now define a new Hilbert space,

$$\mathcal{H}' = \mathcal{H}_A \otimes \mathcal{H}_B \otimes \mathcal{H}_{A'} \otimes \mathcal{H}_{B'}, \tag{167}$$

choosing the ancillary $\mathcal{H}_{A'} \otimes \mathcal{H}_{B'}$ in such a way that there exists a purification $|\psi\rangle \in \mathcal{H}'$ such that

$$\rho_{AB} = \text{Tr}_{\mathcal{H}_{A'} \otimes \mathcal{H}_{B'}} |\psi\rangle \langle\psi|. \tag{168}$$

Of course, this purification is not unique, and the EoP is defined in terms of the *von Neumann entropy* $S(\rho) = -\operatorname{Tr}(\rho \log \rho)$, as

$$E_P := \inf_{|J\rangle \in \mathcal{H}'} S(\operatorname{Tr}_{BB'} |J\rangle \langle J|), \tag{169}$$

*i.e.*, the minimum of the entanglement entropy between subsystems $A \oplus A'$ and $B \oplus B'$. Accordingly, determining the EoP in general requires an optimization over the full Hilbert space $\mathcal{H}'$, which is a computationally intensive task that quickly becomes unfeasible.

A much more reasonable problem is to focus instead on the *Gaussian EoP*, obtained by assuming that both the initial mixed state and the purification are Gaussian. The optimization over all purifications then reduces to the familiar problem of optimization on the sub-manifold of Gaussian states composed from purifications of $\rho_{AB}$. The properties of these states have been discussed in some detail in section 2.6 – in fact, the only difference to note here is that in the context of EoP, we label the original subsystem by $A \oplus B$ rather than $A$ and the ancillary by $A' \oplus B'$ rather than $A'$.

We recall from the previous discussion on Gaussian purifications that the manifold of purifications can be parametrised in terms of complex structures $J$ with restrictions to the subsystems given by the restricted complex structures $J_{AB}, J_{A'B'}, J_{A'A}, J_{BB'}$. In Refs. [18, 36], an expression based on this parametrization was derived for the Gaussian entanglement entropy, first defined in [37] for bosons and [38] for fermions. The expression reads

$$S_{AA'}(|J\rangle) = \begin{cases} \operatorname{Tr}\left( \frac{\mathbb{1}_A + iJ_{AA'}}{2} \log \left| \frac{\mathbb{1}_A + iJ_{AA'}}{2} \right| \right) & \textbf{(bosons)} \\ -\operatorname{Tr}\left( \frac{\mathbb{1}_A + iJ_{AA'}}{2} \log \frac{\mathbb{1}_A + iJ_{AA'}}{2} \right) & \textbf{(fermions)} \end{cases}, \tag{170}$$

and once again makes use of the complex structure formalism to provide a unified expression for both bosons and fermions. This expression can be framed more concisely by defining $D = \frac{1}{2}(\mathbb{1} + iJ_{AA'})$ as

$$S_{AA'} = \begin{cases} \frac{1}{2} \operatorname{Tr}\left( D \log D^2 \right) & \textbf{(bosons)} \\ -\operatorname{Tr}\left( D \log D \right) & \textbf{(fermions)} \end{cases}, \tag{171}$$

where we used $\log |D| = \frac{1}{2} \log D^2$, as $D$ has real eigenvalues. The derivative of the entanglement entropy can be obtained by a straightforward application of the product rule and the cyclicity of the trace as

$$dS_{AA'} = \begin{cases} \frac{1}{2} \operatorname{Tr}\left( dD \log D^2 \right) & \textbf{(bosons)} \\ -\operatorname{Tr}\left( dD \log D \right) & \textbf{(fermions)} \end{cases}, \tag{172}$$

where we have defined $dD = \frac{i}{2} \delta J_{AA'}$. Note that we have $\operatorname{Tr}(dD) = 0$ due to fact that $\delta J$ is anti-symmetric in a basis where $G$ proportional to the identity. These results are summarized in Table 4.

Equipped with a manifold of pure Gaussian states and a scalar function and its derivative defined on this manifold in terms of the complex structure, we are now in a position to employ our optimization algorithm to efficiently compute the Gaussian EoP.

In practice, we begin with a matrix representation of the mixed state reduced complex structure $J_{AB}$ in a basis $\hat{\xi} = (\hat{\xi}_A, \hat{\xi}_B)$ which decomposes over the two subsystems $A$ and $B$. We denote the transformation by $T$ which relates $J_{AB}$ to its mixed state standard form $J_{\text{sta}}^{\text{m}}$ defined in (79), so that

$$J_{AB} = T J_{\text{sta}}^{\text{m}} T^{-1}. \tag{173}$$

The transformation is obtained from the eigenvectors of $J_{AB}$ as discussed in section 2.6. We can now construct an initial purification of the form in (79). In doing so, we are free to choose any $\dim(A'B') \geq \dim(AB)$, and we speak of a *minimal purification* when the number of modes in $AB$ is the same as that in $A'B'$.

The convenience of our choice of basis as $\hat{\xi}' = (\hat{\xi}_A, \hat{\xi}_B, \hat{\xi}_{A'}, \hat{\xi}_{B'})$ now becomes evident: In this basis, the complex structure of the purified state will take the block form

$$J = \left( \begin{array}{c|c} J_{AB} & J_{AB,A'B'} \\ \hline J_{A'B',AB} & J_{A'B'} \end{array} \right), \tag{174}$$

where the blocks on the main diagonal are the restricted complex structures defining the mixed states $\rho_{AB}$ and $\rho_{A'B'}$. It should be noted that, in contrast, the off-diagonal blocks do not represent complex structures as they map from $A \oplus B$ to $A' \oplus B'$ or vice versa. While $J_{AB}$ is fixed to preserve the restriction to the original subsystem, varying $J_{A'B'}$ and the off-diagonal blocks in a compatible way corresponds to different purifications of $\rho_{AB}$. The state manifold of interest is therefore parametrized by the transformations $M_{A'B'}$ acting on the reduced complex structure $J_{A'B'}$, which act on the full complex structure as $M = \mathbb{1}_{AB} \oplus M_{A'B'}$.

We initialize the optimization algorithm at the initial purification, which is in the standard form and therefore the very first transformation must be of the form

$$M_0 = T \oplus \widetilde{M}_0, \tag{175}$$

where $M_0$ denotes an arbitrary starting point in the variational manifold and the leading block in the transformation returns $J_{AB}^{\text{st}}$ to its initial form $J_{AB}$. This ensures that the restriction of $J_1 = M_0 J_0 M_0^{-1}$ to $A \oplus B$ returns the initial reduced complex structure $J_{AB}$. The optimization can then proceed in the way outlined in the previous section, with steps

$$M_n = \mathbb{1}_{AB} \oplus \widetilde{M}_n \tag{176}$$

to ensure that the optimization procedure leaves $J_{AB}$ unchanged.

A comprehensive study of Gaussian entanglement of purification in free quantum field theories based on our methods can be found in [17].

## 5.3 Gaussian complexity of purification (CoP)

In ref. [39], it has first been suggested that a notion of (circuit) complexity might provide fresh insights and might meaningfully complement notions of entanglement in holography. Motivated by the subsequent interest in *holographic complexity* as well as a preceding geometric interpretation of complexity in quantum circuits [40], significant attention has been dedicated to extending notions of complexity to quantum field theories [41, 42]. Since the thermal and ground states of free quantum fields are Gaussian, a framework for the complexity of Gaussian states has been developed in refs. [20, 21], which we draw on here.

A particular area of recent interest has been the study of *complexity of purification* (CoP) as a correlation measure in composite quantum systems based on the notion of complexity rather than entanglement [43]. In this context, a typical problem would be the following: We are given a mixed state in some Hilbert space $\mathcal{H}_A$, which is characterized by a density matrix $\rho_A$. We now define a new Hilbert space,

$$\mathcal{H}' = \mathcal{H}_A \otimes \mathcal{H}_{A'}, \tag{177}$$

choosing the ancillary system $A'$ in such a way there exists a purification $|\psi_T\rangle \in \mathcal{H}'$ such that

$$\rho_A = \text{Tr}_{A'} |\psi_T\rangle \langle \psi_T|. \tag{178}$$

We refer to this purification as the target state. The CoP is defined as the minimum of a complexity function $C$ with respect to some reference state $\psi_\text{R}$ over all purifications of the initial state *i.e.*,

$$C_P = \min_{|\psi_\text{T}\rangle \in \mathcal{H}'} \mathcal{C}(|\psi_\text{T}\rangle, |\psi_\text{R}\rangle). \tag{179}$$

There are several distinct proposals for the complexity function $C(|\psi_\text{T}\rangle, |\psi_\text{R}\rangle)$ in the literature. In the context of this work, we will once again focus only on *Gaussian CoP* by making the assumption that both the reference and target states are Gaussian in nature. For bosonic and fermionic Gaussian states, there exists a consensus definition[17] associated with the geodesic distance between reference and target states, whose analytical expressions have been derived in ref. [21] for bosons and in ref. [20] for fermions.

The most concise formulation of this complexity function unsurprisingly involves the relative complex structure, introduced in (45), of the target and reference states,

$$\Delta = -J_\text{T} J_\text{R}, \tag{180}$$

which captures all the information between the two. In terms of $\Delta$, the complexity is then defined as

$$\mathcal{C} = \sqrt{\frac{|\text{Tr}\log^2(\Delta)|}{8}}, \tag{181}$$

although for the purposes of a numerical optimization, the square root is irrelevant and can be neglected.

Given this parametrization of the complexity in terms of complex structures, we may obtain the CoP by optimization on the manifold of Gaussian purifications of the initial $J_A$, as in the previous section. However, we may also choose a more computationally efficient approach, by noting a somewhat subtle point regarding the complexity function. By the cyclicity of the trace, any transformation $J_T \mapsto M J_T M^{-1}$ will change the complexity (181) in a way equivalent to the transformation $J_R \mapsto M^{-1} J_R M$. This means that we can choose to optimize over the manifold of pure reference states rather than target states. This seems arbitrary until we note that we may assume without loss of generality that the reference state is a product state between $A$ and $A'$ *i.e.*, that the matrix representation of $J_\text{R}$ in our basis is simply

$$J_\text{R} = [J_\text{R}]_A \oplus [J_\text{R}]_{A'}, \tag{182}$$

where the subscripts $A$ and $A'$ denote the restrictions to either subsystem.

A comprehensive study of Gaussian complexity of purification in free quantum field theories based on our methods can be found in [17].

# 6 Optimality of Gaussian EoP

This section focuses on a specific application defined and reviewed in the previous section 5.2, namely entanglement of purification. We combine our numerical results from our numerical algorithm with several analytical arguments to support the conjecture that for mixed Gaussian states only Gaussian purifications are required to compute the entanglement of purification.

---

[17]Note, however, that even for Gaussian states, there also exist alternative $p$-norm definitions [44].

## 6.1 Conjectures on optimality

We will present numerical and analytical arguments for the validity of the following two conjectures.

**Conjecture 1** (Gaussian optimality conjecture). *Given a mixed Gaussian state $\rho$ of a bosonic or fermionic system and a system decomposition $V = A \oplus B$ with $N_A$ and $N_B$ degrees of freedom, respectively, it is sufficient to optimize over all Gaussian purifications to compute the entanglement of purification (minimal entanglement entropy $S_{AA'}$ over all purifications in $\mathcal{H}_A \otimes \mathcal{H}_B \otimes \mathcal{H}_{A'} \otimes \mathcal{H}_{B'}$), i.e., the global minimum of $S_{AA'}$ is reached on the submanifold of Gaussian states.*

**Conjecture 2** (Minimum purification conjecture). *When minimizing the $S_{AA'}$ over all Gaussian purifications, the minimum is reached when choosing the numbers of degrees of freedom of the purifying systems $A'$ and $B'$ to be given by the respective numbers of degrees of freedom in $A$ and $B$, i.e., $N_{A'} = N_A$ and $N_{B'} = N_B$.*

At first sight, this conjecture may appear rather ambitious, considering that we assume that the optimization over the generally exponentially small family of Gaussian purification (compared to all non-Gaussian purifications) is sufficient and that the number of purifying degrees of freedom just need to match the ones of the original system (in contrast to the bounds for finite dimensional non-Gaussian systems from ref. [34]). However, for researchers familiar with typical properties of Gaussian states, our conjectures will likely appear much more realistic, considering that Gaussian states provide in many settings one of the simplest non-trivial realizations of quantum information concepts. This appears in the context analytical formulas for the entanglement entropy and other correlations measures (such as the logarithmic negativity).

Let us emphasize that we have formulated two distinct conjectures that only together provide us with clear instructions on how the full entanglement of purification can be computed numerically from the Gaussian optimization algorithm presented in the previous section. Conjecture 1 ensures that for mixed Gaussian states, we only need to consider Gaussian purifications, but to actually run the algorithm, we need to choose both the total number $N$ of degrees of freedom to purify as well as how we split these purifying degrees of freedom into the auxiliary subsystems $A'$ and $B'$.

## 6.2 Numerical evidence

We provide numerical support for conjectures 1 and 2 based on two paradigmatic models. For bosons, we consider the Klein-Gordon scalar field with mass $m$, discretized on a one-dimensional periodic lattice with $N$ sites, equipped with the Hamiltonian

$$\hat{H} = \frac{\delta}{2} \sum_{i=1}^{N} \left( \hat{\pi}_i^2 + \frac{m^2}{\delta^2} \hat{\varphi}_i^2 + \frac{1}{\delta^4} (\hat{\varphi}_i - \hat{\varphi}_{i+1})^2 \right), \tag{183}$$

where $\delta > 0$ represents the lattice spacing. For fermions, we consider the transverse field Ising model

$$\hat{H} = -\sum_{i=1}^{N} (2J\, \hat{S}_i^{\mathrm{x}} \hat{S}_{i+1}^{\mathrm{x}} + h\, \hat{S}_i^{\mathrm{z}}), \tag{184}$$

in the critical limit $J = h$. Here, $S_i^{\mathrm{x}}$ and $S_i^{\mathrm{z}}$ represent the local spin-1/2 x- and z-component operators on the $i$-th site (the conventions match [45–47]).

Providing categorical numerical evidence for the first conjecture proves a substantial challenge, since it requires an optimization over the entire Hilbert space, which is the daunting problem that our Gaussian approach is trying to circumvent.

Table 5: *Numerical evidence for conjecture 1.* We present the numerically computed non-Gaussian EoP to 7 s.f. for disjoint intervals of width $N_A = N_B = 1$ (which we purify with $N_{A'} = N_{B'} = 1$) at a distance of $d$ sites in the fermionic critical transverse Ising model, on a circle with $N = 100$, with $J = h = 1$. We contrast this with the Gaussian EoP result.

| $d$ | Non-Gaussian | Gaussian |
|----|----|----|
| 10 | 0.00306129 | 0.00306101 |
| 30 | 0.00038316 | 0.00038291 |
| 50 | 0.00000166 | 0.00000151 |
| 70 | 0.00046825 | 0.00046801 |
| 90 | 0.00434489 | 0.00434461 |

Table 6: *Numerical evidence for conjecture 2.* We present the numerically computed Gaussian EoP to 9 s.f. for disjoint intervals of width $N_A$ and $N_B$ at a distance of $d$ sites in the Klein-Gordon model (top) and critical transverse Ising model (bottom), on a circle with $N = 100$ sites. For the Klein-Gordon model, we set the mass to $m/\delta = 0.1$, and for the Ising model, we set $J = h = 1$. The optimal purification, highlighted in color, is evidently obtained for equal numbers of degrees of freedom in the original subsystems and the corresponding subsystems of the ancillary, *i.e.*, $N_A = N_{A'}$ and $N_B = N_{B'}$.

| $N_A + N_B$ | $1+1$ | $1+2$ | | $1+3$ | | | $2+2$ | | |
|---|---|---|---|---|---|---|---|---|---|
| $N_{A'} + N_{B'}$ | $1+1$ | $1+2$ | $2+1$ | $1+3$ | $2+2$ | $3+1$ | $1+3$ | $2+2$ | $3+1$ |
| Klein-Gordon field | | | | | | | | | |
| $d = 10$ | 0.01861871 | 0.02073452 | 0.11482986 | 0.02187326 | 0.02371109 | 0.17471765 | 0.11708904 | 0.02307170 | 0.11708905 |
| $d = 30$ | 0.00022978 | 0.00026256 | 0.09482403 | 0.00028175 | 0.00223536 | 0.15435431 | 0.09486090 | 0.00029999 | 0.09486090 |
| $d = 50$ | 0.00001590 | 0.00002022 | 0.09458539 | 0.00002412 | 0.00197901 | 0.15410719 | 0.09459102 | 0.00002588 | 0.09459102 |
| $d = 70$ | 0.00034749 | 0.00048793 | 0.09504583 | 0.00064375 | 0.00259556 | 0.15470108 | 0.09523927 | 0.00068457 | 0.09523927 |
| $d = 90$ | 0.03052751 | 0.04357474 | 0.13688870 | 0.05929784 | 0.06093414 | 0.20883147 | 0.15464517 | 0.06226844 | 0.15464518 |
| Critical transverse field Ising model | | | | | | | | | |
| $d = 10$ | 0.00288040 | 0.00639951 | 0.06677170 | 0.00933387 | 0.01324964 | 0.11126531 | 0.07202181 | 0.01234729 | 0.07202181 |
| $d = 30$ | 0.00057335 | 0.00135921 | 0.06301066 | 0.00210074 | 0.00604134 | 0.10643203 | 0.06426257 | 0.00276998 | 0.06426256 |
| $d = 50$ | 0.00040596 | 0.00098339 | 0.06273091 | 0.00155212 | 0.00549403 | 0.10606776 | 0.06367226 | 0.00204620 | 0.06367227 |
| $d = 70$ | 0.00062304 | 0.00153900 | 0.06314448 | 0.00247813 | 0.00641779 | 0.10668275 | 0.06466840 | 0.00326765 | 0.06466840 |
| $d = 90$ | 0.00408126 | 0.01078810 | 0.07007032 | 0.01883801 | 0.02269729 | 0.11772862 | 0.08218632 | 0.02506887 | 0.08218632 |

In the fermionic case, the finite-dimensional Hilbert space allows us to adapt our approach to gradient descent using Lie groups and algebras to this problem by optimizing over the (compact) group of unitary transformations $\mathrm{U}(2^N)$ of an $N$-mode density operator. On the manifold of transformations $U \in \mathrm{U}(2^N)$ with respect to some reference state $\rho_0$, parametrizing the non-Gaussian purifications according to $\rho_U = U \rho_0 U^\dagger$, we can define the entanglement entropy function as

$$S_{AA'} = -\mathrm{Tr}\left(\rho_{AA'} \log \rho_{AA'}\right), \tag{185}$$

with $\rho_{AA'} = \mathrm{Tr}_{BB'}\left(U \rho_0 U^{-1}\right)$. In line with the previous discussion, we can also define the derivative of $S_{AA'}$ as

$$dS_{AA'} = -\mathrm{Tr}\left(\delta \rho_{AA'} \log \rho_{AA'}\right), \tag{186}$$

with $\delta \rho_{AA'} = \mathrm{Tr}_{BB'}\left(U[\rho_0, \hat{K}]U^{-1}\right)$ for $\hat{K} \in \mathfrak{u}(2N)$.

It should be noted that computing the partial trace for a fermionic density operator in this context is non-trivial. In practice, we construct the initial purified density operator in

the convenient basis $\hat{\xi} = (\hat{\xi}_A, \hat{\xi}_B, \hat{\xi}_{A'}, \hat{\xi}_{B'})$. Tracing out the subsystem $BB'$ therefore involves a permutation of the degrees of freedom, in the sense $\rho_{ABA'B'} \mapsto \rho_{AA'BB'}$. While such a re-ordering is trivial for commuting bosonic degrees of freedom (or spin degrees of freedom), the permutation of fermionic creation operators do anti-commute, so the computation of the partial trace to find $\rho_{AA'}$ will lead to extra sign flips due to the required permutations. This is subtle, but well-understood in various contexts [48–51] and already taken into account when we computed the entanglement entropy of fermionic Gaussian states in (170).

**Evidence for conjecture 1.** This approach to non-Gaussian optimization proves efficient at small system sizes, however, the computational effort grows exponentially in the number of degrees of freedom in the system and it soon becomes unfeasible. Table 5 shows the non-Gaussian EoP for the fermionic (critical) Ising model, within the numerically accessible regime. Evidently, this data supports the conjecture that the optimal purification of a mixed Gaussian state is Gaussian.

**Evidence for conjecture 2.** We can tackle the second conjecture in a more comprehensive way, since it only requires us to perform Gaussian optimization. Table 6 shows the numerical Gaussian EoP for a variety of dimensions, for both the bosonic and fermionic cases. Evidently, here we also see good agreement between the numerical results and our expectations based on the conjecture.

## 6.3 Analytical bounds

As alluded in the previous section, it is highly plausible to conjecture that the purification for which the entanglement entropy is minimized belongs to the class of Gaussian states (conjecture 1). We have some further analytical evidence for this: After all, the map from quantum states on $\mathcal{H}'$ to ones on $\mathcal{H}_A \otimes \mathcal{H}_{A'}$ performing a partial trace over the complement of $\mathcal{H}_A \otimes \mathcal{H}_{A'}$ can be seen as a constrained Gaussian channel, reflecting the constraint that the inputs must be such that the reductions to $\mathcal{H}_A \otimes \mathcal{H}_B$ are precisely the given Gaussian states $\rho_{AB}$. Captured in this way, the entanglement of purification can be seen as a solution to a *minimum output entropy* problem of a Gaussian quantum channel [52,53], a problem in which the von-Neumann entropy of the output of a quantum channel is minimized under varying the input of the channel. At least for Gaussian bosonic systems this question has been settled under rather general conditions [52,53] (albeit not under the specific constraints considered here). This connection will be made more precise elsewhere. Not referring to this conjecture, the *Gaussian entanglement of purification* (only allowing for Gaussian purifications), constitutes an upper bound for $E_P$.

That said, the quality of approximation can be bounded by a *lower bound* to $E_P$ that can be computed [34]. This is the *entanglement of formation* (EoF) $E_F$ of $\rho_{AB}$ [54], satisfying

$$E_F(\rho_{AB}) \leq E_P(\rho_{AB}), \tag{187}$$

and being defined as the infimum

$$E_F(\rho_{AB}) := \inf \sum_j p_j S(\mathrm{Tr}_B |\psi_j\rangle \langle \psi_j|), \tag{188}$$

with

$$\sum_j p_j |\psi_j\rangle \langle \psi_j| = \rho_{AB}. \tag{189}$$

The entanglement of formation can be in instances computed and also conveniently bounded [55]. The easiest such lower bounds, valid for arbitrary as well as for Gaussian states, for which it is extremal both in the bosonic and fermionic [56] setting, is the *hashing bound*

$$S(\mathrm{Tr}_B \rho_{AB}) - S(\rho_{AB}) \leq E_F(\rho_{AB}) \leq E_P(\rho_{AB}). \tag{190}$$

Another insight helpful in the numerical optimization of the Gaussian entanglement of purification is a bound to the number of auxiliary modes constituting systems $A'B'$ that is required without restricting generality. Naively, one might expect that one needed a squared number of bosonic or fermionic modes in the purification. In fact, it is easy to see that one can restrict systems $A'B'$ to be composed of as many modes $N_{A'}$ and $N_{B'}$ as $A$ and $B$ consist of, *i.e.*, $N_A$ and $N_B$.

Given a quantum state $\rho_{AB}$, it will be associated with some $J_{AB}$. As discussed in section 2.6, we can always find a basis, such that $J_{AB}$ takes the standard form of a mixed state given by (79). Note, however, that this will be in general with respect to a basis that mixes the degrees of freedom of $A$ and $B$. If we start with a basis $\hat{\xi}_1 = (\hat{\xi}_A, \hat{\xi}_B)$, there exists a group transformation $T_{AB} \in \mathcal{G}_{AB}$, such that

$$J_{AB} \equiv \begin{pmatrix} J_A & J_{A,B} \\ J_{B,A} & J_B \end{pmatrix} = T_{AB} J_{\mathrm{sta}}^{\mathrm{m}} T_{AB}^{-1}, \tag{191}$$

where the mixed state standard form was defined in (79). We can use this $T_{AB}$, which combines $A$ and $B$ to construct a purification $|J\rangle_{ABA'B'}$, in which $A'$ and $B'$ are correlated in the same way. For this, we complete the basis $\hat{\xi} = (\hat{\xi}_A, \hat{\xi}_B)$ from (191) to $\hat{\xi}' = (\hat{\xi}_A, \hat{\xi}_B, \hat{\xi}_{A'}, \hat{\xi}_{B'})$ and choose in this basis

$$J \equiv T J_{\mathrm{sta}}^{\mathrm{p}} T^{-1} \quad \text{with} \quad T = T_{AB} \oplus T_{A'B'}, \tag{192}$$

*i.e.*, we use the same transformation $T_{AB}$ to combine $\hat{\xi}_A$ and $\hat{\xi}_B$ as we use to mix $\hat{\xi}_{A'}$ and $\hat{\xi}_{B'}$. Here, we have the purified standard form $J_{\mathrm{sta}}^{\mathrm{p}}$ from (83). From our numerical studies, we know that this choice is generally not the optimal one, but it provides a meaningful starting point for our optimization algorithm.

We can also move on to arrive at analytical upper bounds, however. For this purpose, we block-diagonalize the submatrices $[J]_A$ and $[J]_B$ individually (rather than $[J]_{AB}$ as a whole as in (191)), *i.e.*, we write

$$J \equiv M \tilde{J} M^{-1} \quad \text{with} \quad M = M_A \oplus M_B \oplus \mathbb{1}_{A'B'}, \tag{193}$$

with $M_A \in \mathcal{G}_A$ and $M_B \in \mathcal{G}_B$, so that

$$\tilde{J}_{AB} = \left( \begin{array}{ccc|c} \tilde{c}_1^A \mathbb{A}_2 & \cdots & 0 & \\ \vdots & \ddots & \vdots & X \\ 0 & \cdots & \tilde{c}_{N_A}^A \mathbb{A}_2 & \\ \hline & & & \tilde{c}_1^B \mathbb{A}_2 & \cdots & 0 \\ & -X^{\intercal} & & \vdots & \ddots & \vdots \\ & & & 0 & \cdots & \tilde{c}_{N_B}^B \mathbb{A}_2 \end{array} \right), \tag{194}$$

where $\mathbb{A}_2$ has been introduced in (84) and $X$ is some $2N_A \times 2N_B$ rectangular matrix and $\tilde{c}_i^A$ and $\tilde{c}_i^B$ are real numbers in $[0, \infty)$ for bosons and $[0, 1]$ for fermions. As $\tilde{J}$ is just originating from a basis transformation of the purified $J$ and $J_{\mathrm{sta}}^{\mathrm{p}}$, we have

$$\tilde{J}^2 = -\mathbb{1}. \tag{195}$$

One can now arrive at analytical upper bounds, acknowledging the following insight. The von-Neumann entropy of $AA'$ can be computed from the reduction $[\tilde{J}]_{AA'}$ via (170). This way, the von-Neumann entropy formula can directly be computed on the level of $[\tilde{J}]_{AA}$. Let $P$ be the *pinching* that projects the matrix $\tilde{J}$ into the $2 \times 2$ block diagonal form both in the main

block and the off diagonal block of $\tilde{J}$. Since $i\tilde{J}$ is Hermitian (with respect to the inner product of $g$), such a pinching will render the resulting matrix $P(i\tilde{J})$ more mixed than $i\tilde{J}$ in the sense of *majorization* [57], *i.e.*, if the non-increasingly ordered eigenvalues of $i\tilde{J}_{AA'}$ are $\pm i\tilde{\lambda}_i$ and the ones of $P(i\tilde{J}_{AA'})$ are $\pm i\tilde{\lambda}'_i$, we will have

$$\sum_{i=1}^{j} \tilde{\lambda}'_i \leq \sum_{i=1}^{j} \tilde{\lambda}_i \quad \text{and} \quad \sum_{i=1}^{N_A+N_{A'}} \tilde{\lambda}'_i = \sum_{i=1}^{N_A+N_{A'}} \tilde{\lambda}_i, \tag{196}$$

for all $j$ in the first equation. Since the function $S_{AA'}(\tilde{J})$ as a function of $\tilde{J}$ is Schur-concave both for bosons and fermions, we have

$$S_{AA'}(P(i\tilde{J})) \geq S_{AA'}(i\tilde{J}), \tag{197}$$

again for both bosons and fermions. In other words, the pinched matrix gives rise to an upper bound to the von-Neumann entropy of the involved quantum states and hence also an upper bound to the entanglement of purification. That said, now the eigenvalues entering the expression can be read off directly, giving rise to an explicit formula of an upper bound of the entanglement of purification. This mindset can be used to avoid costly numerical optimization and to study systems in the thermodynamic limit, while still arriving at reasonable bounds.

## 6.4 Proof of local optimality

Some further analytical evidence in support of conjecture 1 is provided by the fact that the entanglement entropy $S_{AA'}$ is locally optimal for a Gaussian purification, *i.e.*, we will prove that after finding the optimal Gaussian purification $|J\rangle_{ABA'B'}$ with minimal $S_{AA'}$ any infinitesimal non-Gaussian change of $|J\rangle_{ABA'B'}$ will not lower $S_{AA'}$. For simplicity of notation, we write $|J\rangle$ for the purification $|J\rangle_{ABA'B'}$ on $ABA'B'$. We consider the mixed Gaussian state $\rho_{AB}$. Let us define $|J\rangle$ as the optimal Gaussian purification, *i.e.*, a Gaussian state vector such that

$$\rho_{AB} = \text{Tr}_{A'B'} |J\rangle \langle J| \tag{198}$$

and such that the entanglement entropy $S_{AA'}(|J\rangle \langle J|) = S(\rho_{AA'})$ is minimal among all Gaussian states. In practice, we would choose here $N_A = N_{A'}$ and $N_B = N_{B'}$ as suggested by conjecture 2, but this is not important for the argument.

As discussed in section 3.8, we can write the mixed Gaussian $\rho_{AA'} = \exp(-\hat{H}_{AA'})/Z$ with $\hat{H}_{AA'} = q_{ab}\hat{\xi}^a\hat{\xi}^b$ and $Z = e^{c_0}$ based on formula 8. If we now perturb our optimal purification in a non-Gaussian way, *i.e.*, by applying a unitary

$$|\psi_\epsilon\rangle = (\mathbb{1}_A \otimes \mathbb{1}_B \otimes U_{A'B'}(\epsilon))|J\rangle, \tag{199}$$

with $U_{A'B'}(0) = \mathbb{1}_{A'} \otimes \mathbb{1}_{B'}$, the first law of entanglement entropy [58–60] states that the linear change of $\delta S_{AA'}$ around $\epsilon = 0$ is given by

$$\delta S_{AA'} = \frac{d}{d\epsilon} \langle \psi_\epsilon | \hat{H}_{AA'} | \psi_\epsilon \rangle \big|_{\epsilon=0}. \tag{200}$$

However, we note that $\hat{H}_{AA'}$ is a quadratic Hamiltonian, which implies that the first order change of the entanglement entropy will only feel the change of the two-point function of $|\psi_\epsilon\rangle$. This means that at linear order, we can replace the change of $|\psi_\epsilon\rangle$ by a Gaussian change of the state. However, by assumption the state vector $|J\rangle$ has been the optimal Gaussian purification, such that any Gaussian perturbation will always increase the entanglement entropy $S_{AA'}$. Moreover, as the Gaussian purification $|J\rangle$ has been assumed to be optimal among all

Gaussian purifications, the variation $\delta S_{AA'}$ will vanish at linear order. In summary, even if we allow for non-Gaussian perturbations of $|J\rangle$, we will have

$$\delta S_{AA'} = 0 \,. \tag{201}$$

However, this does not exclude the possibility of a finite transformation $U_\epsilon$ to lower the entanglement entropy, but constitutes a first step towards proving that the Gaussian purification is optimal.

## 7 Discussion

We have presented a geometric approach to optimize over arbitrary differentiable functions on the manifolds of pure bosonic or fermionic Gaussian states. Our method is based on the well-known gradient descent algorithm, but exploits the natural action of a Lie group onto these manifolds to move between different Gaussian states. This way, we can efficiently perform gradient descent with respect to the Fubini-Study metric associated to the manifold of Gaussian states. In the context of variational families, it is an important question if a given manifold satisfies the so-called Kähler property [15], but for the purpose of gradient descent on Gaussian manifolds this property is not important and we show explicitly how our approach can be applied to suitable Gaussian submanifolds (generated by subgroups of the symplectic or orthogonal group).

For the most part of this manuscript, we used a new formalism for the treatment of Gaussian states that largely unifies the bosonic and fermionic case and emphasizes their similarities. This formalism is based on the geometric Kähler structures consisting of a metric $G$, a symplectic form $\Omega$ and a complex structure $J$ on the classical phase space $V$ of the theory, as reviewed in section 2. In order to carefully distinguish if a matrix represents a linear map (such as $J$), a bilinear form (such as $G$ and $\Omega$) or a dual bilinear form (such as $g$ and $\omega$), we used the index position of a respective matrix entry (such as $J^a{}_b$ vs. $G^{ab}$). As there are many equivalent ways to describe and parametrize Gaussian states, we provided a comprehensive dictionary in section 3 to allow for a seamless conversion between different formalisms. This dictionary may also be of use to other applications involving Gaussian states.

We have further implemented our optimization algorithm numerically to study three applications that are relevant for condensed matter physics, quantum information and high energy theory, namely finding approximate ground states, computing the Gaussian entanglement of purification (EoP) and finally calculating the so-called Gaussian complexity of purification (CoP). For each of these applications, we have reviewed the key ingredients of our optimization procedure, namely an analytical expression for the function and its gradient in terms of the complex structure $J$ parametrizing our Gaussian state family.

In section 6, we have combined numerical and analytical insights to support a conjecture on the optimality of Gaussian entanglement of purification, *i.e.*, we have claimed that for a mixed Gaussian state it is sufficient to optimize entanglement of purification only over Gaussian states. This claim has been supported by numerical evidence from small fermionic systems, where we can also perform the full optimization over all purifications and find that it agrees with one over only Gaussian purifications. Moreover, we have shown analytically that the Gaussian entanglement of purification is locally optimal even in the larger set of non-Gaussian optimizations. Finally, our conjecture also makes a statement about the required number of degrees of freedom (and their distribution) in the purifying subsystem. This is supported by our numerics as well.

The key reason why we do not need to re-evaluate the Fubini-Study metric at each step of our optimization algorithm lies in the fact that our optimization manifold (Gaussian states or

suitable submanifolds) are generated by the Lie group $\mathcal{G}$ (Sp$(2N, \mathbb{R})$ for bosons, O$(2N, \mathbb{R})$ for fermions) or a suitable subgroup $\mathcal{G}'$. As we have a unitary representation $\mathcal{U}(M)$ of group elements $M \in \mathcal{G}$, the Hilbert space inner product is preserved under the left-action of this group. It therefore suffices to choose an orthonormal basis of Lie algebra elements at one point (at a given reference state vector $|J_0\rangle$ in the manifold) and this basis will stay orthonormal when moving to other states via the group action $\mathcal{U}(M)|J_0\rangle = |MJ_0M^{-1}\rangle$. Another advantage is that we naturally ensure to not overparametrize, *i.e.*, we can remove those Lie algebra elements that do not change the reference state vector $|J_0\rangle$ which ensures via the natural group action that we also do not have redundant directions at other states. All of these desirable properties also apply to other families of pure states, as long as they are generated from some unitary representation of a Lie group. A prominent example of such families are the so-called group theoretic coherent states introduced by Gilmore [61, 62] and Perelomov [63, 64]. The only difference to the Gaussian case is that we may not have equally simple analytical formulas for the functions we would like to optimize, such as expectation values (Wick's theorem) or entanglement entropies. Of course, our method also applies to the family of all pure states (projective Hilbert space) and in fact, we already used an appropriately adjusted version of our algorithm when we computed the full non-Gaussian entanglement of purification in section 6.2 for small fermionic systems (large fermionic or general bosonic systems are not feasible due to the large or infinite dimension of the associated Hilbert space). In practice, we find that our algorithm significantly outperforms approaches in which the optimization space does not take the Lie algebra symmetries into account. For example, we find an order-of-magnitude speedup of entanglement of purification calculations relative to previous methods used by one of the authors [35], even though this previous method relied on a limited-memory Broyden–Fletcher–Goldfarb–Shannon (L-BFGS) implementation usually considered superior to the gradient descent method used here. This highlights the potential of using our approach to achieve even faster Gaussian state optimization relying on more involved optimization step functions.

# Acknowledgements

We thank Eugenio Bianchi, Hugo Camargo, Ignacio Cirac, Tommaso Guaita, Michal Heller, Tadashi Takayanagi and Tao Shi for inspiring discussions. BW is supported in part by the Heinrich Böll Foundation undergraduate scholarship scheme and the Imperial College President's Undergraduate Scholarship and gratefully acknowledges the hospitality from MPQ. AJ has been supported by the FQXi as well as the Perimeter Institute for Theoretical Physics. Research at Perimeter Institute is supported by the Government of Canada through the Department of Innovation, Science, and Economic Development, and by the Province of Ontario through the Ministry of Research and Innovation. JE has been supported by the DFG (CRC 183, project B01, FOR 2724, EI 519/14-1). This work has also received funding from the European Union's Horizon 2020 research and innovation programme under grant agreement No. 817482 (PASQuanS). LH acknowledges support by VILLUM FONDEN via the QMATH center of excellence (grant No. 10059).

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
