# Peer review of "Local optimization on pure Gaussian state manifolds"

_SciPost Physics, doi:SciPost Phys. 10, 066 (2021)_

## Round 2 · Referee Report · Michael Kaicher (Referee 1) · 2020-12-8

Strengths

When discussing fermionic and bosonic states, one typically does so distinctly, see e.g. [1]. This paper shows how both types of Gaussian states may be talked about in a unified way by means of Kaehler structures. A strength of this paper is that is summarizes various parametrizations of Gaussian states and shows how to convert between them and doing so in a unified way for bosonic and fermionic Gaussian states.

It tackles one of the most important problems in the study of many-body quantum systems, namely extremizing an arbitrary function (such as the energy expectation value) within the family of Gaussian states using an efficient algorithm based on gradient descent.

It gives a very thorough introduction and supports its claims (a conjecture concerning the entanglement of purification) using both analytical and numerical arguments.

It includes toy examples (for bosonic and fermionic systems, respectively) in order to make the introduction part more accessible to the reader.

It provides a Mathematica package ("GaussianOptimization.m") as well as a example notebook that includes the functions used in the numerical example section, and can be accessed by the reader.

It very clearly highlights whenever a certain representation is used.

References:
[1] Shi, T., Demler, E., & Cirac, J. I. (2018). Variational study of fermionic and bosonic systems with non-Gaussian states: Theory and applications. Annals of Physics, 390, 245-302

Weaknesses

It is often times hard to understand which parts of the paper present a new result to the field, as much of the content is basically already contained in e.g. [2]. In particular, I believe, this holds true for one of the main parts in the paper: finding the extremum of an arbitrary function within the family of Gaussian states without the need to evaluate the symplectic form or metric at each iteration step. It is not clear to me how this is different from (or if it even claims to be different from) Chapter 5.2.5 in [2]. This might be totally my fault, but I seem to miss the key difference between what was presented in [2] opposed to what is presented here and what the key differences are. This is basically my main and only point of critique, and I believe that clarifying how (or if) this gradient descent approach differs from previous approaches (such as [1] or [2]) would help the reader a lot.

In Chapter VI.C, analytical lower and upper bounds are derived in the form of the Entanglement of Formation (EoF) and the pinched matrix (Eq.196), respectively. I wonder if it is possible to check for the two studied finite examples, if the numerical values displayed in Table V and VI lie within these bounds? This is less of a critique and more curiosity, as it seems even though analytical bounds where found, they were not studied with regards to the two systems at hand, if I am not mistaken. Is this because the bounds only hold in a certain limit?

As this work is of importance in particular for theoretical physicists that focus on quantum simulation and who may not have such a strong mathematical physics background, I believe it would add a lot to the readability of the paper if an appendix would be added, that show the minor steps in the derivation of various equations, or at least always make sure to have the citation where this basic computation may be found at hand. Instead of an appendix, more detailed footnotes would maybe also already do the trick.

References:
[2]: Geometry of variational methods: dynamics of closed quantum systems,
Lucas Hackl, Tommaso Guaita, Tao Shi, Jutho Haegeman, Eugene Demler, J. Ignacio Cirac, SciPost Phys. 9, 048 (2020)

Report

In this paper, a unified approach to the study of Gaussian states for bosonic and fermionic systems is given.

After providing a review of Gaussian states and introducing a unified notation for both bosonic and fermionic systems in Chapter II, the paper gives a comprehensive summary of the various representations of Gaussian states and how to switch among them in Chapter III, summarized in Table II.

In Chapter IV, the paper shows how to extremize an arbitrary function (such as the energy expectation value) on the family of fermionic and bosonic Gaussian states based on a gradient descent approach in a way which does not require having to evaluate the inverse metric in each iteration (this is a typically quite computationally costly step, see e.g. [1]).

In Chapter V, the authors provide three prominent examples of analytical functions which one wishes to find the global/local extremum, namely the energy expectation value (whose respective Gaussian state gives an approximate ground state to a given problem Hamiltonian), the Gaussian EoP and Complexity of Purification (CoP), which is a correlation measure in composite many-body systems.

This paper provides numerical and analytical support for two conjectures in Chapter VI, namely the Gaussian optimality (conjecture 1)-, and minimum purification conjecture (conjecture 2 - both defined on page 26 of the arXiv manuscript), which (if both hold true) state that Gaussian purifications "are sufficient to compute the Entanglement of Purification (EoP) of arbitrary mixed Gaussian states". It studies two example quantum mechanical systems in order to check if these two conjectures hold: The Klein-Gordon scalar field for bosons and the transverse field Ising model for fermions.

The results for conjecture 1 are summarized in Table V and they show the numerically computed values of the EoP for non-Gaussian- and Gaussian states for the fermionic transverse field Ising model. As for all studied system sizes, the EoP values are smaller for Gaussian states than for non-Gaussian states, using Eq.168 it follows that the conjecture holds that optimal purification of a mixed Gaussian state is Gaussian.

Evidence that conjecture 2 holds for the studied bosonic and fermionic systems is given in Table VI. While it is argued why the number of degrees of freedom of the purifying systems $A'$ and $B'$ must be identical to the number of degrees of freedom in $A$ and $B$ in the main text of VI.C (in other words, why $N_{A}+N_{B} = N_{A'}+N_{B'}$), Table VI validates conjecture 2 by showing that the minimal EoP is obtained for $N_A=N_{A'}$ and $N_B=N_{B'}$.

The results of the paper are discussed and summarizes in Chapter VII.

Requested changes

1- Eq.(10) for bosons: This should probably read $\left[\hat \xi^a,\hat \xi^b\right]=i\Omega^{ab}$

  1. Eq.(14)-(15): The sentences "$\rho$ is pure a Gaussian state" and "$\rho$ is pure Gaussian" are a little confusing - it is supposed to be the same, right?

  2. Page 4, end of the first paragraph: Did you mean to write: $|\psi\rangle=|J,z=0\rangle=|J\rangle$?

  3. Page 6, end of last paragraph: cxample = example

  4. Page 7, between Eq.(50)-(51) "also satisfying $[K,J_0]$" - I believe there is something missing, because this is just the commutator, not a relation/equation.

  5. Page 8, text in Example 6: "... he tangent" = the tangent

  6. Page 12: Before Eq.(91) "defined is defined" = "is defined"

  7. Page 14: before Sec. D: "One can show that ... for all $\xi$" - please provide a Reference here, or at least a footnote.

  8. Page 15: sentence leading to Eq. (103): "We have the ... matrix $\Gamma$. Using the ..." - the sentences should be put together.

  9. Page 15: Eq.(112): for fermions, there should be $\frac{1}{2}g_{ac}{K^c_{}}b\left(\hat \xi^a+\hat \xi^b_++\hat \xi^a_-\hat \xi^b_-\right)$ instead of $\frac{1}{2}\omega_{ac}{K^c_{}}b\left(\hat \xi^a+\hat \xi^b_++\hat \xi^a_-\hat \xi^b_-\right)$

  10. Page 15: Eq.(109) and (107): Is $M=T$?

  11. Page 16: Between Eq.(117)-(118): Is definition $L=\tanh(K)$ for both fermions and bosons?

  12. Page 16: Before Eq.(125), define $\alpha,\dot\alpha,\beta,\dot\beta$.

  13. Page 19: Reference for Eq.(153), e.g. [3]

  14. Page 20: Reference at the end of the "... the product of two matrices."

  15. Page 21: FIG. 3: Define ONB within caption or figure!

  16. Page 22: In "Extension to global optimization": you have the sentence "...sufficiently far separated starting points..." - sufficiently separated in comparison to what? What is the criteria here?

  17. Page 26: After Eq.(126): "new subsystem $A'$" - not $B$.

  18. Page 28: Reference after sentence "... under rather general conditions (...) [Ref!]"

  19. Page 28: After Eq.(190): "We can use this $M_{AB}$... - did you mean to write $T_{AB}$? Or is this just the same as in my point 11.?

  20. Page 29: After Eq.(193): It's easier to give the Equation reference for the double slashed $A_2$ here, then making the reader search for it.

References: [3]: Kraus, C. V. (2009). A quantum information perspective of fermionic quantum many-body systems (Doctoral dissertation, Technische Universität München).

  • validity: high
  • significance: high
  • originality: ok
  • clarity: good
  • formatting: excellent
  • grammar: excellent

Author:  Lucas Hackl  on 2021-01-20  [id 1164]

(in reply to Report 1 by Michael Kaicher on 2020-12-08)

We thank the referee for the careful review and implemented the suggested changes. Let us respond to the main points raised by the referee:

“It is often times hard to understand which parts of the paper present a new result to the field, as much of the content is basically already contained in e.g. [2]. In particular, I believe, this holds true for one of the main parts in the paper: finding the extremum of an arbitrary function within the family of Gaussian states without the need to evaluate the symplectic form or metric at each iteration step. It is not clear to me how this is different from (or if it even claims to be different from) Chapter 5.2.5 in [2]. This might be totally my fault, but I seem to miss the key difference between what was presented in [2] opposed to what is presented here and what the key differences are.” We believe that the main novelty of the present manuscript is the in-depth presentation and explicit construction of the optimization algorithm for pure Gaussian states and certain submanifolds (in particular, those generated by a subgroup of the full symplectic/orthogonal group). While the idea for the presented algorithm was sketched in [2], it was not worked out explicitly for pure Gaussian states (or any submanifolds generated by subgroups) and only real/imaginary time evolution is discussed. In contrast, the present manuscript introduces our algorithm from scratch and constructs explicitly the representation of states by their complex structure together with the respective orthonormal basis of Lie algebra elements. Moreover, we show explicitly how the derivative of a general function f(J) can be computed analytically as a trace of powers in J (with explicit formulas for EoP and CoP). In summary, the main difference between the one-page sketch in [2] and the present manuscript is that we worked out the idea explicitly and in full detail for pure Gaussian state manifolds.

“This is basically my main and only point of critique, and I believe that clarifying how (or if) this gradient descent approach differs from previous approaches (such as [1] or [2]) would help the reader a lot.” We thank the referee for this suggestion and implemented this by explicitly comparing the results of the present manuscript with other approaches in the literature (such as [1] and [2]). The key difference to [1] is that we give an explicit group-theory inspired parametrization of the Gaussian state manifold (namely, using group elements M rather than covariance matrices). Moreover, [1] only discusses these equations in the context of imaginary time evolution (optimizing the expectation value of some operator), while we show explicitly how the same strategy can be used for arbitrary functions (such as EoP and CoP), which cannot be understood as imaginary time evolution.

“As this work is of importance in particular for theoretical physicists that focus on quantum simulation and who may not have such a strong mathematical physics background, I believe it would add a lot to the readability of the paper if an appendix would be added, that show the minor steps in the derivation of various equations, or at least always make sure to have the citation where this basic computation may be found at hand. Instead of an appendix, more detailed footnotes would maybe also already do the trick.” We greatly appreciate this suggestion and carefully went through the whole manuscript to add additional comments and references. Most importantly, we explicitly referred to the reference [4], which contains longer derivations and relevant propositions with proofs.

Requested changes 1. Corrected. 2. Corrected. 3. Corrected. 4. Corrected. 5. Corrected. 6. Corrected. 7. Corrected. 8. Corrected. 9. Corrected. 10. Corrected. 11. No, M is more general than T because T is the exponential of a K_+, so for every M we find a corresponding T, such that TJT^{-1}=MJM^{-1}, i.e., they are related by the U(N) redundancy for parameterizing Gaussian states. 12. Yes, L=tanh(K) applies to both bosons and fermions. The difference is that the eigenvalues -/+ix of K are imaginary for fermions, so that we get tanh(ix)=i tan(x). 13. Corrected. 14. Corrected. 15. We explained how this is achieved. 16. Corrected. 17. We added some further explanation 18. Corrected. 19. Corrected. 20. Corrected. 21. Corrected.

--- References --- [1] Shi, T., Demler, E., & Cirac, J. I. (2018). Variational study of fermionic and bosonic systems with non-Gaussian states: Theory and applications. Annals of Physics, 390, 245-302 [2] Geometry of variational methods: dynamics of closed quantum systems, Lucas Hackl, Tommaso Guaita, Tao Shi, Jutho Haegeman, Eugene Demler, J. Ignacio Cirac, SciPost Phys. 9, 048 (2020) [3] Kraus, C. V. (2009). A quantum information perspective of fermionic quantum many-body systems (Doctoral dissertation, Technische Universität München). [4] Bosonic and fermionic Gaussian states from linear complex structures, Lucas Hackl, Eugenio Bianchi, arXiv:2010.15518.

---

## Round 2 · Referee Report · Anonymous (Referee 2) · 2021-1-13

Strengths

1- Gaussian states are an important family to optimize over and the authors present an optimized way to do so 2- unified treatment for fermionic and bosonic Gaussian, which additionally gives insight into the analogies between these closely related families of states/operations 3- a useful and comprehensive overview of ways to parameterize pure Gaussian states 4- highlights the mathematical structures exploited

Weaknesses

1- sometimes a bit too condensed for my taste (it would help to remind the reader of the properties used in certain steps of the derivations; examples below) 2- some more explanations or targeted references regarding mathematical machinery (Lie algebra, tangent spaces, ) would be welcome 3- the authors explain how their approach exploits mathematical structures to obtain a more efficient optimization algorithm, but doesn't mention compared to which other algorithms nor the expected speed-up. 4- does not provide the code implementing the algorithm

Report

The authors present a thorough discussion of an optimization approach in the manifold of Gaussian states. They take extra care to exploit the analogies between fermionic and bosonic Gaussian states to present a their results in a unified treatment for both. I think that both the results and the presentation are valuable and recommend publication.

To set the stage for their optimization algorithm, the authors carefully analyze the group / Lie algebra structure of the Gaussian unitaries that can be used to parameterize the variations of the states. This allows them to discard transformations that leave a given state invariant and focus of those that change it in the optimization procedure. They also present a overview of different currently used notations / parameterizations of Gaussian states and how to convert between them. While this is exploited in the algorithm, I think it is also valuable independent of that application.

The authors outline the individual steps of their optimization procedure (characterizing states by the symplectic/orthogonal transformation to obtain it from a reference state; choosing a basis adapted to the point around which they optimize such that the matrix representation of the metric doesn't change) supported by graphical and diagrammatical figures. They also give some hints on extending the work to constrained or global optimization. Then they demonstrate and employ their method to three interesting examples: finding the ground state energy (and doing approximate real time evolution), and the computation of Gaussian entanglement of purification and Gaussian complexity of purification. In the final section, two conjectures about the optimality of Gaussian purifications for EoP are supported with numerical and analytical evidence.

The present work is substantial and well-written and in my opinion clearly suitable for publication in the SciPost Physics: it is written clearly, and introduces the subject treated well in abstract and introduction. The analytical derivations can be reproduced and the relevant literature is cited. Code to reproduce the numerical results is not provided (I would welcome this, of course, but it's not clear whether SciPost acceptance criteria require it).
In my view, the optimization approach presented here has potential to be useful in many contexts where optimization of manifolds of Gaussian states are of interest. In addition, it exposes an unified treatment of bosons and fermions which may be useful also outside of the optimization context and allow a common treatment of both types of Gaussian states in other areas.

Requested changes

1- I find the mathematical argument in some places very compressed and and occasionally a few explaining words (or reminder of properties used) would be helpful to the reader (eg., the derivations of eq.(54) or eqs.(73,74) or the properties of J that allow to go from (169) to (170), which Thm. is used to obtain (195)...).

2- A subsection discusses the Mathematica package (even "we provide [...]"), but no indicator whether and under what conditions it might be available and if so how to obtain it.

3- eq.(10): commutator brackets and expectation-value brackets seems to be missing and there's a spurious "t"

4- p3, 1st col, after eq.(6): aren't the CAR {qi,qj}=\delta_{ij}={pi,pj} while [qi,pj}=0?

5- p3, 2nd col, footnote 2 remarks on fermionic (but non-Gaussian) states with non-zero first moments. My understanding is that all such states are non-physical as they violate the parity superselection rule and would allow for signaling. A reference for the non-Gaussian, but physical state would be welcome.

6- p4, 1st col: the invertibility of Omega is (for fermions) only assured for pure states, maybe that's worth mentioning explicitly before or after eq.(13). It's not immediately obvious from the foregoing that (14) characterizes pure Gaussian states; maybe a reference would help the reader not familiar with this characterization.

7- p4, 2nd col: "the eigenvectors \hat{\xi}^a_\pm of J" is there a "\ket{J}" missing as \hat{\xi}^a_\pm are operators, not vectors? But maybe I misunderstand something here, as further down "\hat{\xi}^a_\pm" seems again be referred to as a vector (spanning the eigenspace of J). Please explain.

8- p4, 2nd col: In eq.(18) expectation-value brackets are missing in the LHS of the 2nd column. I seem to get a "-i\Omega" in eq.(19), maybe the authors can confirm that the sign error is on my side...

9- p6, eq.(42): why is it U(N) and not U(J) (or U(Gamma)?) The set depends on the state and the dimension (which N fixes) would also implicitly fixed by the dimensionality of an argument J?

10- p7, 2nd col. I'm not sure that the classification of symmetric spaces (CI, DIII) is immediately known by all readers. Maybe a reference would be in order.

11- p7, 2col: I guess \rho_\Gamma = |J><Gamma| instead of |Gamma><Gamma| is a typo; if not, please explain. (A similar combination of |J> and |Gamma| appears in the first row of Table II.)

12- p8, it reads as if eqs.(55) follow from eqs.(53,54), but as g(A,B), but I'm missing where a definition of the real linear forms g, omega is given. Or is (55) that definition?

13- p8, 1st col: eq.(58): why u(N), not u(J)?

14- p10, 1st col: shouldn't J_A be called a "restricted complex structure"?

15- p10, eq.(77): should the subscripts "1" (of J and T) be "A" instead?

16- p15, eq.(106): indices on LHS?

17- p15, 2nd col, before eq.(110): L is undefined here, I suppose it refers to L=tanh(K) as defined subsequently? Or to Table II?

18- p16, 2nd col: what is meant by "the splitting from eq.(124)" (right after that eq.)?

19- p16, 2nd col: maybe I missed where the indec notation used in eq.(125) was introduced, if not maybe a brief explanation how to relate (a,b) and the alpha, \dot{alpha},.. could be added. eq.(127): the RHS seems to equal gamma^* (Re gamma - i Im gamma...) eq.(130): check, the subscripts in the imaginary parts are wrong

20- p17, 1st col: after eq.(135) some statement about n_i appears to be missing.

21- p18, scalar density vs. scalar function?

22- p.18, after eq.(147): shouldn't the last of the four expressions in the standard basis be for omega rather than Omega?

23- p20, caption Fig. 2: should the "vector field \tilkde{X}" refer to {\cal F} instead? R(F) should also be mentioned in the caption.

24- p21, Fig. 3: can't it happen that there is no s for which F_{n+1} < F_n? (should )

25- p25, eq.(170): why is there a factor 1/2 in the eq. for fermions? (169) seems to say S_{AA'} = -tr(D log D) does eq.(171) use tr(delta J)=0 and why does this hold?

26- p26, 1st col, after eq.(176): "ancillary system B" should probably be "A'"; and in (177) what is traced out is probably A', not A.

27- p29, after eq.(193): refer to eq.(83) for definition of A_2

28- p29, 1st col.: "One finds for the reduction of the von Neumann entropy via (169)." sounds strange or incomplete. (is the meaning: One finds the vN entropy of the reduction via (169)?)

29- minor typos: * p4, 1st col, after eq.(13): "under consideration" (no plural) * p4, 2nd col: eq.(16) are 2N equations * p7, 1st col: "reach a standard form" (nor "forms") * p8, 2nd col: before eq(68): "that it is compatible" -> "that are compatible" * p11, before eq.(85): "find" -> "to find" * p14, 1st line "We" -> "we" * p26, 2nd col, last paragraph: "purifications" (s missing) * p30, 2nd col: "our manifolds" (s missing)

  • validity: high
  • significance: high
  • originality: high
  • clarity: good
  • formatting: excellent
  • grammar: good

Author:  Lucas Hackl  on 2021-01-20  [id 1165]

(in reply to Report 2 on 2021-01-13)

We thank the referee for the careful review and implemented the suggested changes. Let us particularly respond to the identified weaknesses of the manuscript:

“sometimes a bit too condensed for my taste (it would help to remind the reader of the properties used in certain steps of the derivations; examples below)” We added further comments to various derivations and also cited the manuscript [1] that appeared in the meantime, which contains more detailed derivations and propositions with proofs in the context of the unified formalism of Gaussian states that we are using.

“some more explanations or targeted references regarding mathematical machinery (Lie algebra, tangent spaces, ) would be welcome” We added further references/citations to relevant literature.

“the authors explain how their approach exploits mathematical structures to obtain a more efficient optimization algorithm, but doesn't mention compared to which other algorithms nor the expected speed-up.” The speedup of the algorithm depends on the precise implementation. Our approach simplifies the optimization landscape by removing redundant search directions and ``flattening’’ it by using a metric that is constant throughout optimization. This generally increases stability and convergence speed in different implementations; in practice, our gradient descent method for entanglement of purification (see Sec. 5B) is roughly ten times faster than a previous method for the same problem without this implementation used by one of the authors in arXiv:1902.02369, despite it relying on an L-BFGS method that is usually considered faster than gradient descent. We have added a comment on speedups at the end of the discussion section.

“does not provide the code implementing the algorithm” The Mathematica package and notebook were included in our arXiv submission. We would be happy to also provide it in some other form if SciPost gives such an option.

Requested changes: 1. We included additional comments and references. 2. We added a sentence to explain that the package is included in the arXiv submission. 3. Corrected. 4. Corrected. 5. We agree that the parity superselection rule forbids such quantum states in genuine fermionic systems. However, we were also thinking of using the fermionic formalism to describe other physical systems, such as spin systems using the Jordan-Wigner transformation. In this case, z^a may not be zero for relevant physical states described within the fermionic formalism. The simplest example would be a single Qubit whose Hilbert space can be described as the Fock space of a single fermionic degree of freedom, where any superposition of spin up and spin down will have non-zero z^a. We clarified this in the text. 6. We added a comment and a reference. 7. We added a comment. \xi is a vector whose components are operators. \xi^a_\pm are eigenvectors of J in the sense of the footnote we added. 8. We checked the equations and they should be correct. There are no expectation value brackets needed as the canonical commutation and anti-commutation relations are exact. We do not find a factor of -i for our conventions. 9. Yes, U(N) and also the algebra u(N) depend on J and one could indicate this by an index. We added a comment, but decided to not consistently write an additional index to not overload notation. 10. We added a reference. 11. Corrected. 12. Corrected. 13. See above at 9. 14. Corrected. 15. Corrected. 16. Corrected. 17. We removed the reference to L at this point. 18. We were referring to the block decomposition and rephrased accordingly. 19. We added an additional explanation of the notation in (125) and corrected the signs of the imaginary parts in (127) and (130). 20. Corrected. 21. We added an explanation on how a scalar density changes under a coordinate transformation. 22. Corrected. 23. Corrected. 24. If the gradient norm is non-zero (i.e., we don’t have the stop condition satisfied), we know that for sufficiently small s we must have F_{n+1}<F_n, which follows from the definition of the gradient. 25. There should not be a factor 1/2 and we corrected this. We have tr(D)=0 due to the fact that J is anti-symmetric in a basis where G is proportional to the identity. We added a comment. 26. Corrected. 27. Corrected. 28. Corrected. 29. Corrected.

--- References --- [1] Bosonic and fermionic Gaussian states from Kähler structures, Lucas Hackl, Eugenio Bianchi, arXiv:2010.15518 .

---

## Round 3 · Referee Report · Anonymous (Referee 2) · 2021-2-26

Report

The authors have adequately addressed all concerns and questions I had. Optimization over Gaussian states is important in many areas of physics and the method proposed and explained in this work, which exploits the mathematical structure of this set, is likely to provide important advances in efficiency and stability and has a broad range of applicability. The authors have made available a well-documented Mathematica-package, implementing their approach.
In addition, the manuscript provides a valuable summary of properties and representations of (pure) Gaussian states and carefully shows how to treat bosonic and fermionic states in a single formalism and it my well become a useful reference for these matters.
In summary, I consider this a high-quality contribution and gladly recommend publication.

---

## Round 3 · Referee Report · Michael Kaicher (Referee 1) · 2021-2-26

Strengths

When discussing fermionic and bosonic states, one typically does so distinctly, see e.g. [1]. This paper shows how both types of Gaussian states may be talked about in a unified way by means of Kaehler structures. A strength of this paper is that is summarizes various parametrizations of Gaussian states and shows how to convert between them and doing so in a unified way for bosonic and fermionic Gaussian states.

It tackles one of the most important problems in the study of many-body quantum systems, namely extremizing an arbitrary function (such as the energy expectation value) within the family of Gaussian states using an efficient algorithm based on gradient descent.

It gives a very thorough introduction and supports its claims (a conjecture concerning the entanglement of purification) using both analytical and numerical arguments.

It includes toy examples (for bosonic and fermionic systems, respectively) in order to make the introduction part more accessible to the reader.

It provides a Mathematica package ("GaussianOptimization.m") as well as a example notebook that includes the functions used in the numerical example section, and can be accessed by the reader.

It very clearly highlights whenever a certain representation is used.

References:
[1] Shi, T., Demler, E., & Cirac, J. I. (2018). Variational study of fermionic and bosonic systems with non-Gaussian states: Theory and applications. Annals of Physics, 390, 245-302

Weaknesses

The weaknesses were corrected in the resubmitted version.

Report

In this paper, a unified approach to the study of Gaussian states for bosonic and fermionic systems is given.

After providing a review of Gaussian states and introducing a unified notation for both bosonic and fermionic systems in Chapter II, the paper gives a comprehensive summary of the various representations of Gaussian states and how to switch among them in Chapter III, summarized in Table II.

In Chapter IV, the paper shows how to extremize an arbitrary function (such as the energy expectation value) on the family of fermionic and bosonic Gaussian states based on a gradient descent approach in a way which does not require having to evaluate the inverse metric in each iteration (this is a typically quite computationally costly step, see e.g. [1]).

In Chapter V, the authors provide three prominent examples of analytical functions which one wishes to find the global/local extremum, namely the energy expectation value (whose respective Gaussian state gives an approximate ground state to a given problem Hamiltonian), the Gaussian EoP and Complexity of Purification (CoP), which is a correlation measure in composite many-body systems.

This paper provides numerical and analytical support for two conjectures in Chapter VI, namely the Gaussian optimality (conjecture 1)-, and minimum purification conjecture (conjecture 2 - both defined on page 26 of the arXiv manuscript), which (if both hold true) state that Gaussian purifications "are sufficient to compute the Entanglement of Purification (EoP) of arbitrary mixed Gaussian states". It studies two example quantum mechanical systems in order to check if these two conjectures hold: The Klein-Gordon scalar field for bosons and the transverse field Ising model for fermions.

The results for conjecture 1 are summarized in Table V and they show the numerically computed values of the EoP for non-Gaussian- and Gaussian states for the fermionic transverse field Ising model. As for all studied system sizes, the EoP values are smaller for Gaussian states than for non-Gaussian states, using Eq.168 it follows that the conjecture holds that optimal purification of a mixed Gaussian state is Gaussian.

Evidence that conjecture 2 holds for the studied bosonic and fermionic systems is given in Table VI. While it is argued why the number of degrees of freedom of the purifying systems $A′$
and $B′$ must be identical to the number of degrees of freedom in $A$ and $B$ in the main text of VI.C (in other words, why $N_A+N_B=N_{A′}+N_{B′}$), Table VI validates conjecture 2 by showing that the minimal EoP is obtained for $N_A=N_{A′}$ and $N_B=N_{B′}$.

The results of the paper are discussed and summarizes in Chapter VII.

Requested changes

none

---

## Round 3 · Author Response

We implemented the changes requested by the referees.

---

## Round 3 · List of Changes

A detailed list of changes is provided in our reply to the referees' lists of requested changes.

---

## Editorial Decision

published